# Conserved regulatory logic at accessible and inaccessible chromatin during the acute inflammatory response in mammals

Azad Alizada [1,2,11], Nadiya Khyzha[3,4,11], Liangxi Wang[1,2], Lina Antounians[1,2], Xiaoting Chen[5], Melvin Khor[3,4], Minggao Liang[1,2], Kumaragurubaran Rathnakumar [1,4], Matthew T. Weirauch [5,6,7,8], Alejandra Medina-Rivera[1,9], Jason E. Fish [3,4,10 ✉] & Michael D. Wilson [1,2 ✉]

The regulatory elements controlling gene expression during acute inflammation are not fully elucidated. Here we report the identification of a set of NF-κB-bound elements and common chromatin landscapes underlying the acute inflammatory response across cell-types and mammalian species. Using primary vascular endothelial cells (human/mouse/bovine) treated with the pro—inflammatory cytokine, Tumor Necrosis Factor-α, we identify extensive (~30%) conserved orthologous binding of NF-κB to accessible, as well as nucleosome-occluded chromatin. Regions with the highest NF-κB occupancy pre-stimulation show dramatic increases in NF-κB binding and chromatin accessibility post-stimulation. These 'pre-bound' regions are typically conserved (~56%), contain multiple NF-κB motifs, are utilized by diverse cell types, and overlap rare non-coding mutations and common genetic variation associated with both inflammatory and cardiovascular phenotypes. Genetic ablation of conserved, 'pre-bound' NF-κB regions within the super-enhancer associated with the chemokine-encoding *CCL2* gene and elsewhere supports the functional relevance of these elements.

[1] Hospital for Sick Children, Genetics and Genome Biology, Toronto, Canada. [2] Department of Molecular Genetics, University of Toronto, Toronto, Canada. [3] Department of Laboratory Medicine and Pathobiology, University of Toronto, Toronto, Canada. [4] University Health Network, Toronto General Hospital Research Institute, Toronto, Canada. [5] Center for Autoimmune Genomics and Etiology, Cincinnati Children's Hospital, Cincinnati, OH, USA. [6] Division of Biomedical Informatics, Cincinnati Children's Hospital, Cincinnati, OH, USA. [7] Department of Pediatrics, University of Cincinnati College of Medicine, Cincinnati, OH, USA. [8] Division of Developmental Biology, Cincinnati Children's Hospital, Cincinnati, OH, USA. [9] Laboratorio Internacional de Investigación sobre el Genoma Humano, Universidad Nacional Autónoma de México, Juriquilla, Mexico. [10] University Health Network, Peter Munk Cardiac Centre, Toronto, Canada. [11] These authors contributed equally: Azad Alizada, Nadiya Khyzha. ✉email: jason.fish@utoronto.ca; michael.wilson@sickkids.ca

Inflammation has evolved as an adaptive physiological response to infection and tissue injury[1]. However, an exacerbated inflammatory response is a hallmark of many chronic diseases such as autoimmune and cardiovascular diseases[1–3]. Acute inflammation is controlled by a diverse set of gene-regulatory processes that occur in multiple cell types including endothelial cells (ECs), lymphocytes, macrophages, fibroblasts, and adipocytes. In response to cytokines such as tumor necrosis factor alpha (TNFα), pro-inflammatory gene regulation is mediated through the nuclear factor kappa-light-chain-enhancer of activated B cells (NF-κB), a conserved transcription factor (TF) complex that translocates into the nucleus and binds to genomic elements to regulate a variety of biological events, notably host defense, inflammation, stress responses, differentiation, and apoptosis[2,4–8].

In vertebrates, the NF-κB complex is a dimer composed of subunits from a family of five proteins, all of which have DNA binding capacity imparted by the REL homology domain[9]. The predominant dimer driving the classical NF-κB response consists of RELA (p65) and NFKB1 (p50)[5]. In response to TNFα, RELA/NFKB1 heterodimers rapidly translocate into the nucleus and bind to chromatin. NF-κB can then regulate target gene expression through synergistic interactions with other TFs and transcriptional co-activators/repressors, which are dictated by the cell type[10,11]. A remarkable feature of the NF-κB response is the dynamic feedback regulation of the pathway. This involves immediate expression of *NFKBIA*, which encodes the inhibitor IκBα. Induction of IκBα terminates the response and results in the shuttling of inactive NF-κB back into the cytoplasm, thus resetting it for subsequent activation[12]. These oscillations of NF-κB signaling from latency to response followed by resolution are conserved from *Drosophila* to human[13], making NF-κB a paradigmatic rapid response factor[2].

Human disease-causing mutations directly affecting NF-κB and its regulatory proteins have been uncovered for several inflammation-related diseases such as atherosclerosis, inflammatory bowel disease, systemic lupus erythematosus, and rheumatoid arthritis[2,3]. Genetic variations within regions of the genome bound by NF-κB are also of great interest as this variation is a plausible mechanism for interindividual variation in acute and chronic inflammatory responses[14]. Numerous studies have characterized the repertoire of NF-κB binding regions in a variety of mouse and human cell types[14–23]. While these studies have identified ~$2 \times 10^6$ NF-κB binding motifs (canonical or half-sites), there are typically only tens of thousands of robust NF-κB-bound regions in cells. Thus a major challenge remaining is to establish which of the binding regions drive the expression of the hundreds of known NF-κB target genes[24].

To dissect the function of genomic NF-κB binding regions, it is essential to determine the chromatin context of binding events[25]. Studies assessing chromatin accessibility (e.g., DNase-seq and assay for transposase accessible chromatin [ATAC-seq]) as well as genome-wide mapping of TFs and post-translational modifications of histones such as H3K27ac have revealed distinct modes of NF-κB binding[15,16,22,26,27]. Perhaps the best understood mode by which NF-κB interacts with the genome is through binding to regions that are already accessible due to the prior binding of other TFs[15,23,27,28]. These binding regions have been associated with the immediate induction of inflammatory genes, as well as the repression of cell-identity genes[20]. A second mode of NF-κB binding involves delayed access to chromatin through collaborative action of either lineage determining TFs[22] or signal-dependent TFs such as IRF3 in B-cell lines[29,30], which drive chromatin remodeling[22,25,26]. Genes associated with these sites are induced with delayed kinetics, many of which are involved in shutting off the inflammatory response[25]. Although still

considered controversial, several lines of evidence support a third mode in which NF-κB binds to nucleosome-occupied DNA[31–35]; however its importance with respect to the NF-κB response is unclear.

Comparative epigenomics involving cross-species comparisons of chromatin features (e.g., TF binding, histone modifications, accessible chromatin) can uncover functional regulatory elements. Interspecies comparison of TF binding in ES cells[36], liver[37], adipocytes[38], and immortalized cell lines[39] all reveal a relatively small subset of conserved orthologous TF binding (~10–20%), that would not be readily revealed by measuring DNA constraint alone. The functional relevance of conserved orthologous TF binding has been shown in the liver, where conserved master regulator TF binding coincides precisely with disease-causing regulatory mutations[40,41]. Although downstream processes resulting from the inflammatory response can differ substantially between species[42], acute gene expression patterns in fibroblasts and mononuclear phagocytes have revealed the conserved nature of NF-κB pathway induction[43]. Thus, carefully controlled cross-species comparison of acute inflammation may yield biological insights and reveal principles of mammalian gene regulation.

To identify conserved, and hence likely functional, NF-κB–chromatin interactions, we performed a comparative epigenomic study of the acute NF-κB response in primary aortic ECs isolated from human, cow, and mouse. We ascertained RELA binding along with chromatin accessibility and select histone modifications before and during the TNFα-induced acute inflammatory response. Approximately 30% of all human RELA-bound regions (~20,000) were conserved with at least one other species, and a significant fraction of these shared a conserved mode of binding to chromatin, including binding to inaccessible chromatin. Notably, a small number of regions (~1000) with measurable RELA binding prior to TNFα stimulation were among the most highly induced regions 45 minutes (min) after stimulation. Most of these "prebound" regions were shared across species, utilized by multiple cell types, and were components of inflammation-induced super-enhancers. These prebound RELA regions were enriched near NF-κB target genes and overlapped human disease-associated mutations. Genomic deletions of these prebound RELA elements by CRISPR/Cas9 genome editing in ECs further supported their functional importance before and after induction with TNFα. Overall, our cross-species, cross-cell-type study of NF-κB–chromatin interactions reveals mechanistic insights into the NF-κB response and provides a valuable resource of NF-κB-bound elements that likely play a central role in controlling mammalian inflammatory responses and represent genomic hotspots for noncoding variants associated with inflammatory phenotypes and diseases.

## Results

**Conserved orthologous NF-κB–chromatin interactions occur near target genes.** Knowing where and how NF-κB interacts with chromatin to control gene expression is requisite for understanding NF-κB gene-regulatory networks across time, cell type, and disease state[15,17,30]. To identify the immediate NF-κB binding events that occur during an acute inflammatory response in mammals, we first mapped RELA binding using ChIP-seq, and accessible chromatin using ATAC-seq. These experiments were performed in primary human aortic ECs (HAECs), mouse aortic ECs (MAECs), and bovine aortic ECs (BAECs) under basal and TNFα-stimulated conditions (Fig. 1a and Supplementary Dataset 1). The rationale for comparing human, mouse, and cow is that: (a) these species belong to different mammalian orders (presumed common ancestor was ~85 million years ago[44]) and

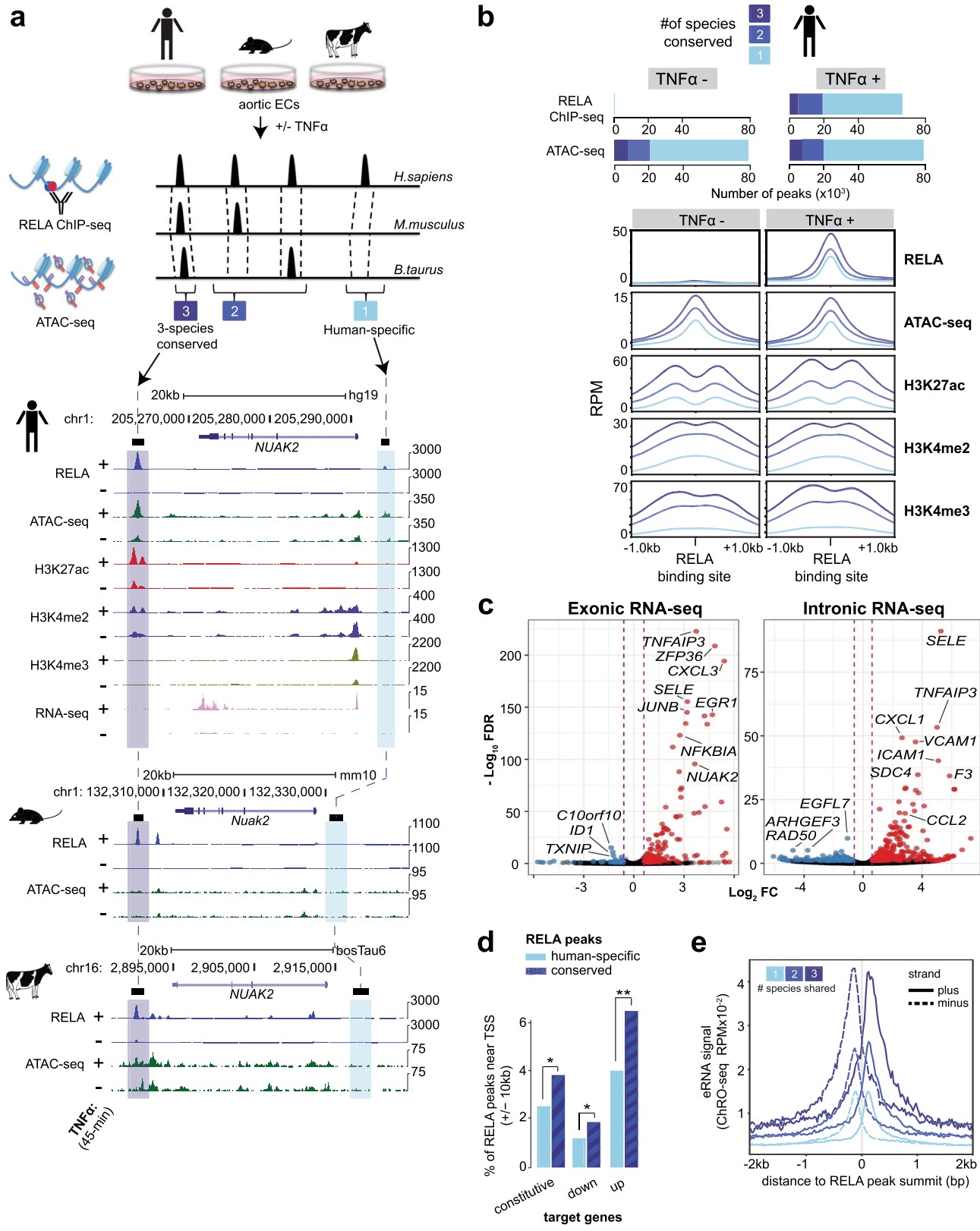

thus provide three evolutionary perspectives of the NF-κB response; and (b) in addition to the obvious importance of mouse models, BAECs are a robust, important, and well-established vascular EC model[45].

ECs from each species were treated with species-matched recombinant TNFα protein. Since we found RELA to be

consistently enriched in the nucleus from 15 min up to 45 min after TNFα treatment (Supplementary Fig. 1a), we processed samples 45 min post-treatment to facilitate the observation of immediate epigenomic and transcriptional changes induced by RELA. ChIP-seq was performed with an antibody generated against the conserved C-terminus of RELA. We detected 66,148

**Fig. 1 Conserved RELA bound regions have strong enhancer features and are enriched near target genes. a** Workflow of the comparative genomic analyses performed with ChIP-seq and ATAC-seq in TNFα stimulated (45 min) primary aortic endothelial cells (ECs) isolated from human, mouse, and bovine aortas. The *NUAK2* locus is used as a representative genomic region depicting 3-species conserved (dark blue) and human-specific (light blue) RELA peaks. **b** Stacked bar graphs (top) showing the number of 3-species conserved (dark blue), 2-species conserved (blue), and human-specific (light blue) RELA ChIP-seq and ATAC-seq peaks in human aortic ECs (HAECs), before and after TNFα stimulation. Profile plots (bottom) showing ChIP-seq and ATAC-seq signals in reads per million (RPM). The signals are centered on RELA peak summits in HAECs. The plot lines depict mean RPM ± SEM. **c** Volcano plots showing TNFα upregulated (red, $\log_2 FC > 0.6$, FDR < 0.1) and downregulated (blue, $\log_2 FC < -0.6$, FDR < 0.1) genes derived from intronic and exonic reads of total RNA-seq of HAECs (45-min TNFα vs. basal). Representative NF-κB target genes are labeled. FC fold change; FDR false discovery rate. **d** Percentages of conserved (combined 3-species and 2-species) and human-specific RELA peaks that reside within +/−10 kb of transcriptional start sites (TSS) of TNFα upregulated, TNFα downregulated, and constitutively expressed HAEC genes (p value: *<$4.0 \times 10^{-4}$, p value: **<$1.0 \times 10^{-18}$, two-sided Fisher's exact test with Bonferroni correction for multiple testing). Source data and exact p values are provided in the Source Data file. **e** Profile plot showing ChRO-seq signals at distal RELA-bound regions (mean RPM × $10^{-2}$; >3 kb from the TSS). The signals are centered on RELA peak summits in HAECs, as indicated by color.

human, 31,602 mouse, and 90,570 bovine RELA peaks 45 min post TNFα (Supplementary Dataset 1). The RELA peaks in mouse and bovine cells were recapitulated using an independent antibody (Supplementary Fig. 1b, c). Using a 1-bp peak overlap in the ENSEMBL-Enredo-Pecan-Orteus (EPO) multiple sequence alignment (MSA) as the criteria for conserved orthologous binding, we found that 30% of human RELA peaks were shared with either mouse or bovine cells, and 8% were conserved in all three species (Fig. 1b). The proportion of conserved orthologous binding with at least one other mammal was similar when studied from the mouse (33%) or bovine (21%) perspectives (Supplementary Fig. 1d). The vast majority of the conserved RELA peaks remained (~98%) when we increased the minimum RELA peak overlap within the MSA from 1 bp to 20, 50, or 100 bp (Supplementary Fig. 1e).

To gain insight into the epigenomic features of RELA peaks in HAECs, before and after TNFα stimulation, we profiled H3K27ac, which is indicative of active promoters and enhancers[46–48]; H3K4me2, which marks active or latent promoters or enhancers[46,47,49]; and H3K4me3, which is enriched at active promoters[47,50]. In the presence of TNFα stimulation, the conserved (2- and 3-species) RELA peak summits had higher ChIP-seq (RELA, H3K27ac, H3K4me2, and H3K4me3) and ATAC-seq signals (normalized read counts, $p < 1 \times 10^{-15}$) when compared to species-specific RELA peaks (Fig. 1b and Supplementary Fig. 1d). These observations are exemplified near the known NF-κB target gene, NUAK family kinase 2 (*NUAK2*)[51] (Fig. 1a). Conserved RELA peaks shared in all three species had higher ChIP-seq and ATAC-seq signals than peaks shared in only two species ($p < 0.0004$, Fig. 1b and Supplementary Fig. 1d). Approximately 35% of the RELA peaks conserved in three species were found to reside within promoter regions (<1 kb to transcription start site (TSS), $p < 2.2 \times 10^{-16}$) compared to ~10% of the human-specific RELA peaks (Supplementary Fig. 1f, g). Many conserved distal intergenic NF-κB binding elements were also detected (~21%, ~4000 HAEC peaks; Supplementary Fig. 1f, g), consistent with the notion that both proximal and distal regulatory elements participate in the pro-inflammatory NF-κB response[16].

To associate RELA peaks with human target gene expression, we performed total RNA-seq on HAECs under basal and 45-min TNFα-stimulated conditions. Using the iRNA-seq approach[52], we separately assessed exonic and intronic reads as a proxy for transcriptional activity (Fig. 1c and Supplementary Dataset 1). We identified 537 differentially expressed genes (exonic and intronic analyses combined; false discovery rate (FDR) < 0.1, |log₂ FC| > 0.6) of which 385 were upregulated and 162 were downregulated 45-min after TNFα induction (Fig. 1c). The conserved RELA peaks were significantly enriched near TNFα-upregulated genes relative to the human-specific peaks (>1.5-fold difference, p

< $9.0 \times 10^{-19}$, +/−10-kb TSS; Fig. 1d). Taken together, conserved NF-κB peak regions showed higher levels of active chromatin marks and robust RELA binding, and a stronger association with TNFα-target genes than the non-conserved NF-κB peaks.

Enhancer activity can be ascertained by global RNA polymerase run-on assays, which reveal the bi-directional transcription of enhancer RNAs (eRNAs)[22]. To test if conserved RELA binding is associated with enhancer activity during acute inflammation we performed chromatin run-on assay (ChRO-seq)[53] before and 45 min after TNFα treatment in TeloHAEC cells, which are a commercially available HAEC line with normal karyotype that recapitulates EC biology[54]. Supporting their reliability as a HAEC model, we found that genome-wide RELA binding and H3K27ac enrichment in TeloHAECs were similar to what we observed for the TNFα-induced RELA response in HAECs (Supplementary Fig. 1h). We found ChRO-seq signals to be the highest at the 3-species conserved RELA peaks followed by 2-species and human-specific RELA peaks (Fig. 1e). Thus, from the perspective of NF-κB target genes and NF-κB bound regions, conserved orthologous RELA peaks highlight a functionally relevant set of inflammatory enhancers.

**NF-κB–chromatin modes are a robust feature of the acute inflammatory response.** NF-κB binding to the genome occurs via different modes of chromatin interaction[25]. To put NF-κB binding into a chromatin context, we ascertained chromatin accessibility at RELA peaks before and 45 min after TNFα stimulation in each species. This allowed us to broadly classify RELA binding into four chromatin accessibility modes: open constitutively (Mode O), open after TNFα stimulation (Mode OA), closed constitutively (Mode C), and closed after TNFα stimulation (Mode CA) (Fig. 2a, b and Supplementary Dataset 2). As expected, the predominant mode of RELA binding across all species was Mode O (57%). Unexpectedly, a substantial proportion of RELA binding (31%) occurred at constitutively closed (Mode C) regions (Fig. 2a and Supplementary Dataset 2). A further 9% occurred at Mode OA regions, and 3% at Mode CA regions. In addition to the four chromatin accessibility-based categories, we identified 571 regions that were prebound with RELA prior to TNFα stimulation (Mode P), 99% of which were Mode O regions (Fig. 2a, b and Supplementary Dataset 2). Conserved orthologous binding was most prevalent at Mode P (55%; approximately twofold, $p < 1.0 \times 10^{-48}$) and Mode O regions (36% ~1.3-fold, $p < 1.0 \times 10^{-50}$), which supports the relevance of our chromatin accessibility-based NF-κB classification scheme (Fig. 2a and Supplementary Dataset 2).

We next used the NucleoATAC method to determine nucleosome occupancy at RELA-bound regions[55] and examined changes in nucleosome occupancies within each of our five NF-κB modes before and after TNFα stimulation. Supporting our

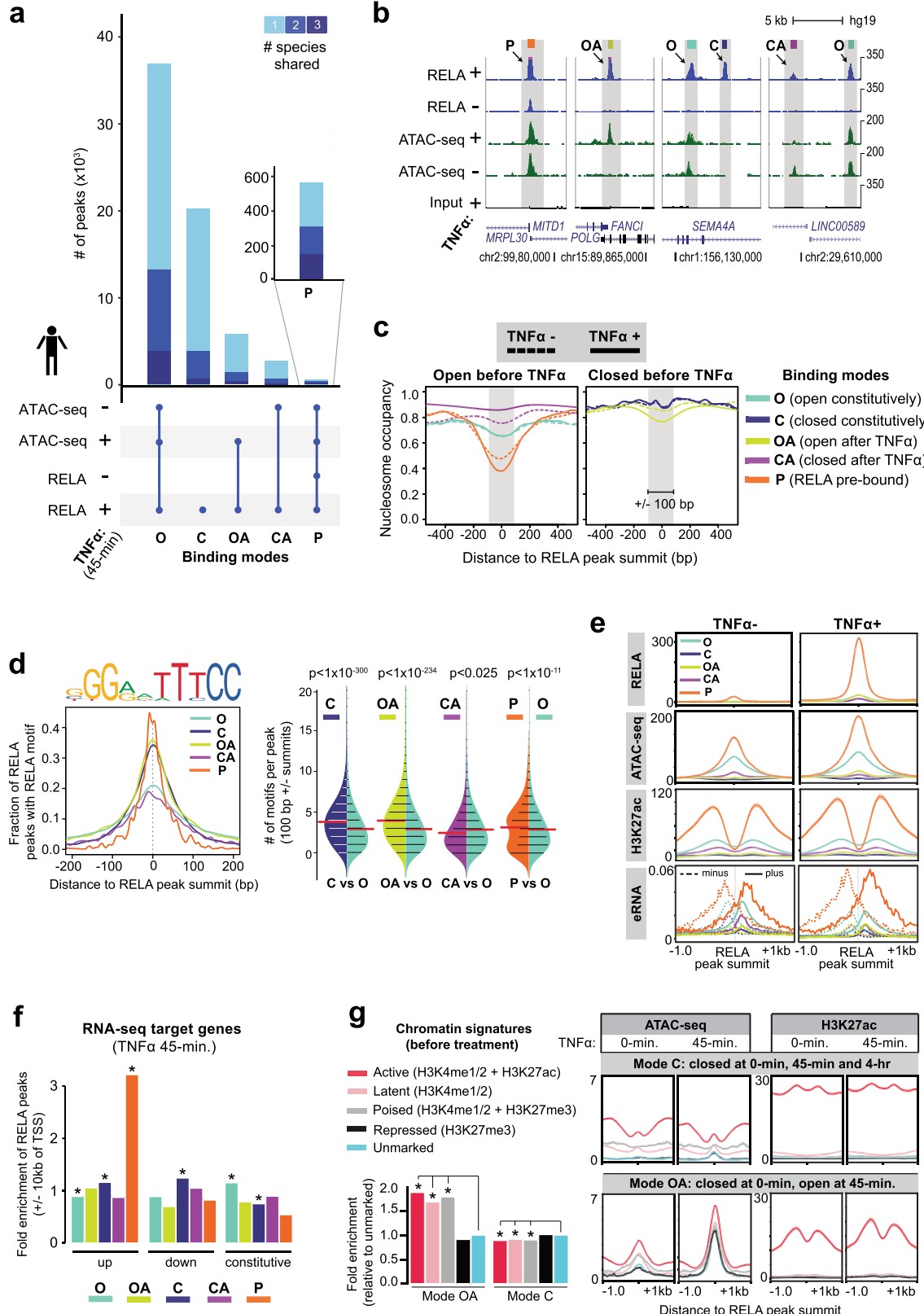

ATAC-seq peak-based categorizations, the highest nucleosome occupancies prior to TNFα treatment were observed at the inaccessible Mode C (0.91) and OA (0.86) regions (Fig. 2c). While TNFα treatment had no obvious effect on nucleosome occupancy in Mode C regions, nucleosome occupancy was significantly decreased (by 9%) in Mode OA regions. Mode P regions had the

lowest average nucleosome occupancy score prior to TNFα treatment (0.48, +/−100 bp of summits, $p < 1.0 \times 10^{-13}$), which was further significantly decreased by 15% after TNFα treatment ($p < 1.0 \times 10^{-70}$). Mode O and CA also had low average nucleosome occupancy scores (0.66 and 0.76, respectively); this was unchanged by TNFα treatment for Mode O, whereas TNFα

**Fig. 2 Genomic regions prebound by RELA under basal conditions show the highest activity following TNFα stimulation. a** Stacked bar chart (top) indicating overlaps of RELA peaks with ATAC-seq peaks in unstimulated (−) and 45-min TNFα-stimulated (+) HAECs. Fractions representing 3-species conserved (dark blue), 2-species conserved (blue), and human-specific (light blue) RELA peaks are shown within each overlap. **b** Representative genomic regions (bottom) illustrating each mode of RELA binding. Mode O (open), RELA binding to accessible chromatin; Mode C (closed), to inaccessible chromatin; Mode OA (open after), to accessible chromatin after TNFα; Mode CA (closed after), to inaccessible chromatin after TNFα; and Mode P (prebound), binding prior to TNFα. Y-axis indicates reads per million (RPM). **c** TNFα-induced changes in nucleosome occupancy for each RELA binding mode from overlaps in **a**. Occupancy scores are indicated for basal (dashed line) and 45-min TNFα-stimulated (solid line) conditions. **d** Density plots (left) showing the fraction of RELA peaks harboring canonical RELA motif (JASPAR - MA0107.1) as a function of distance from RELA peak summit for each mode. Bean plots (right) comparing the number of canonical RELA motifs per peak (+/− 100 bp of RELA peak summits) between modes. p values were calculated using two-sided Mann–Whitney U test with Bonferroni correction. **e** Profile plots showing ChIP-seq (H3K27ac and RELA), ATAC-seq, and ChRO-seq (eRNA) signals (RPM). Signals are centered on RELA peak summits. Plot lines indicate mean RPM ± SEM. **f** Bar plots indicating fold enrichments of RELA binding modes within +/−10 kb of TSS of TNFα upregulated, downregulated, and constitutively expressed HAEC genes from RNA-seq analysis (p value: *<9.0 × 10$^{-3}$ two-sided Fisher's exact test with Bonferroni correction). **g** Bar charts showing fold enrichments for Mode OA peaks within chromatin marked with histone modifications as shown in Supplementary Fig. 2k (p value: *<2.5 × 10$^{-04}$, Chi-squared test with Yates's correction for continuity of independence, Bonferroni correction). Profile plots showing ATAC-seq and H3K27ac ChIP-seq signals centered on Mode C and OA subtypes classified in Supplementary Fig. 2k. Plot lines indicate mean RPM ± SEM. Source Data file is provided for Fig. 2f, g.

treatment increased it by 10% at Mode CA ($p < 1.0 \times 10^{-18}$) (Fig. 2c). Further supporting our finding that NF-κB can bind in the absence of obvious chromatin accessibility, ATAC-seq signal comparisons between Mode C regions and Mode O regions of similar RELA ChIP-seq signal revealed that Mode C regions have lower ATAC-seq signals independent of RELA binding strength (Supplementary Fig. 2a).

We next asked whether Mode C regions could be detected using different biochemical assays for assessing chromatin accessibility (DNase-seq) and nucleosome positioning (MNase-seq) in ECs and other tissues. Indeed, using human umbilical vein EC (HUVEC) DNAse-seq data[56], we found that Mode C regions lacked DNAse-seq signal in contrast to Mode O regions (Supplementary Fig. 2d). Similarly, looking at Mode C regions using HUVEC MNase-seq data[18] revealed high nucleosome occupancy at Mode C peak summits whereas Mode O peaks showed depletion of nucleosomes at peaks summits (Supplementary Fig. 2b). We could also readily identify Mode C regions using paired RELA and DNase-seq data obtained from human adipocytes[20] (Supplementary Fig. 2e) as well as ChIP-seq using a different NF-kB subunit NFKB1 (p50) and ATAC-seq data obtained from stimulated T cells[57] (Supplementary Fig. 2f). Together these results indicate that NF-κB binding at relatively nucleosome-occluded regions is a robust feature of the acute inflammatory response.

**Genomic features of NF-κB binding at accessible and nucleosome-occluded regions**. The DNA motif preference of NF-κB is of direct relevance to its mechanism of action and has been intensively studied[24]. Structural analyses of NF-κB binding demonstrate that it occurs within the context of the canonical pseudo-symmetric 5′-GGGRNWYYCC-3′ motif[58] as well as NF-κB half-sites[59], of which there are more than $2 \times 10^{6}$ sites in the human genome[24]. We found that Mode P regions showed significant enrichments for the canonical RELA motif (Supplementary Fig. 2h, k). Previous work has also demonstrated that RELA is extensively recruited to pre-existing lineage-determining TF-bound regions that do not necessarily contain the canonical RELA motif[15,17,22,28]. Accordingly, we found that the most significant de novo motif enrichments at accessible chromatin regions (Mode O) were for ETS family factors (MEME e value = $4.9 \times 10^{-162}$) (Supplementary Fig. 2h). ETS factors such as ERG play key roles in vascular ECs and have been shown to coincide with RELA binding in ECs[15,54]. Notably, RELA binding to inaccessible chromatin regions (Mode C and OA) also showed strong enrichments for the canonical RELA motif (Supplementary Fig. 2h, k).

The presence of multiple NF-κB motifs is a well-established feature of inflammatory enhancers[60]. Mode P regions contained significantly more RELA motifs compared to all RELA-bound accessible chromatin regions (Mode O) (Fig. 2d), and include the exemplary *NFKBIA* enhancer previously used to model how multiple canonical NF-κB motifs control NF-κB binding at key target genes[60]. We also observed multiple canonical motifs at inaccessible chromatin regions that we classified as having high nucleosome occupancy (Mode C and OA regions). Specifically, Mode C and OA regions had ~4 motifs per peak (+/−100 bp of the RELA peak summit), which is significantly more than what is observed at the accessible Mode O regions ($p < 1.0 \times 10^{-8}$; Fig. 2d).

To further identify distinguishing features of the RELA binding modes, we looked at epigenomic and transcriptional changes induced by TNFα treatment. Relative to all other modes, the prebound Mode P regions showed by far the highest normalized read counts for active H3K27ac marks (greater than twofold, $p < 1.0 \times 10^{-44}$), RELA (greater than ninefold, $p < 2.0 \times 10^{-60}$), ATAC-seq (greater than twofold, $p < 1.0 \times 10^{-48}$), and eRNA (ChRO-seq) before and after TNFα stimulation (Fig. 2e). Similar RELA and H3K27ac ChIP-seq and ATAC-seq results were observed for mouse and cow (Supplementary Fig. 2i). Mode P regions also displayed the highest enrichment near TNFα-upregulated genes (approximately threefold, $p < 1.0 \times 10^{-8}$, +/−10 kb of TSSs) (Fig. 2f), 40% of which (n = 226) were distal intergenic (Supplementary Fig. 2j). In contrast, Mode O regions did not undergo drastic changes in H3K27ac and ATAC-seq signal post-inflammation and were enriched near expressed genes that did not change in response to TNFα stimulation (1.2-fold, $p < 5.0 \times 10^{-7}$, +/−10 kb of TSSs; Fig. 2f). This observation suggests that many of the observed RELA binding events at accessible chromatin (Mode O) reflect opportunistic or even potentially antagonistic binding to established EC-related enhancers.

We observed that NF-κB binding events at regions of high nucleosome occupancy (Mode C and OA regions), occur in genomic regions enriched for both "active" (H3K4me1/2 and H3K27ac) and "repressive" (H3K27me3) histone modifications (Fig. 2g). In the case of the late accessible Mode OA regions, we observe an increase in H3K27ac signal after TNFα stimulation (Fig. 2g and Supplementary Fig. 2k, l). There was a substantial number of Mode C regions (n = 16,204) that were also inaccessible in an independent HAEC dataset where both RELA and ATAC-seq were obtained after 4 h of TNFα treatment[15]. Thirteen percent of these Mode C regions (n = 2166) were conserved orthologous RELA binding events that also fell within

active, H3K27ac-enriched chromatin (Supplementary Fig. 2k). Consistent with their evolutionary conservation, these persistently inaccessible and H3K27ac-marked Mode C regions were significantly enriched near target genes (approximately twofold, $p < 1.0 \times 10^{-15}$, +/−10 kb of TSSs; Supplementary Fig. 2k) and had higher average signal for H3K27ac than the $\sim 4 \times 10^4$ ATAC-seq peaks that lacked RELA binding (Supplementary Fig. 2g). Relative to RELA binding to open chromatin (Mode O) or RELA binding to chromatin that opens after TNFα stimulation (Mode OA), these persistently inaccessible Mode C regions did not occur in well-defined, nucleosome depleted "valleys" of the expected "peak-valley-peak" enrichment profiles of H3K27ac ChIP-seq at RELA peaks (Fig. 2g and Supplementary Fig. 2l). We propose that the presence of multiple canonical motifs presented to NF-κB in nucleosome-occluded—yet permissive—chromatin context provides a plausible, thermodynamic explanation for the existence of Mode C regions.

**NF-κB–chromatin modes are conserved**. All NF-κB modes showed evidence of DNA constraint at the RELA ChIP-seq peak summit with Mode P peaks having the highest constraint (Supplementary Fig. 2m). However, chromatin context affects NF-κB binding[25], and whether the chromatin context of NF-κB binding is also conserved is not known. We postulated that identifying conserved NF-κB-bound regions that also preserved their mode of binding in the other species would be a relevant criteria to identify functional regulatory regions. To do this we ascertained which NF-κB modes were "preserved" (i.e., conserved mode of NF-κB binding) in another species using comparisons of RELA ChIP-seq and ATAC-seq data from mouse and cow (Fig. 3a). Indeed we found that conserved RELA-bound regions were also likely to preserve their mode of binding in one or more species: 47% of Mode P ($p < 1.0 \times 10^{-31}$, $n = 150$), 71% of Mode O ($p < 1.0 \times 10^{-108}$, $n = 9641$), 10% of Mode OA ($p < 1.0 \times 10^{-7}$, $n = 149$), 15% of Mode CA ($p < 4.0 \times 10^{-3}$, $n = 103$), and 63% of Mode C ($p < 1.0 \times 10^{-140}$, $n = 2467$) (Fig. 3b and Supplementary Dataset 2). Generally, the features that distinguished each mode were enhanced when the mode was preserved between species: Mode P showed higher ATAC-seq and RELA and H3K27ac ChIP-seq signals when preserved (normalized read counts, $p < 2.0 \times 10^{-16}$; Fig. 3c); Mode C and Mode OA regions showed more RELA motifs at peak summits when preserved ($p < 3.0 \times 10^{-4}$, Fig. 3d). Strikingly, the preserved Mode P regions were fivefold enriched near TNFα-upregulated genes ($p < 1.0 \times 10^{-6}$, +/−10 kb of TSSs; Fig. 3e). The preserved Mode P and OA regions were enriched for inflammation-related terms such as "immune system process" while the preserved Mode O regions were associated with EC-related functions (e.g., "positive regulation of angiogenesis" and "response to laminar fluid shear stress") (Fig. 3f and Supplementary Dataset 3). Both preserved Mode C and OA regions showed enrichment for genes involved in cytokine production (Fig. 3f and Supplementary Dataset 3). The preserved Mode C regions also gave significant enrichments for "vasculature development" genes, and included binding near EC-related genes including *EGFL7*, *FLT1*, and *NOS3* (Fig. 3f and Supplementary Dataset 3). Overall, the preservation of NF-κB binding modes across species highlights the importance that chromatin context plays during NF-κB gene regulation and provides additional evidence that NF-κB interactions at regions with high nucleosome occupancy is a robust feature of acute inflammatory responses.

**Conserved NF-κB binding events are often utilized by multiple cell types**. NF-κB has both pan-cell-type and cell-type-specific roles[3,61,62]. To determine the genomic features of NF-κB binding that is shared between different cell types, we compared the

HAEC RELA peaks to cell lines where RELA binding was previously ascertained: lymphoblastoid cell line (LCL)[14], adipocyte[20], and HUVEC data[16] (Fig. 4a and Supplementary Fig. 3a). As would be expected based on the literature, we observed pan-cell RELA binding near the essential NF-κB inhibitory gene *NFKBIA*[12], EC-specific RELA binding near the endothelial gene *EGFL7*[63], LCL-specific RELA binding near the B-cell expressed CD23 antigen encoded by *FCER2*[64], and adipocyte-specific RELA binding near the lipid sensor GPR120, encoded by *FFAR4*[65] (Fig. 4a). Compared to the overall 30% conserved orthologous RELA binding we observed, 58% of the four-cell-type-shared, and 44% of the three-cell-type-shared RELA regions were conserved in one or more species (Fig. 4b and Supplementary Fig. 3b). Consistent with being functionally relevant for inflammation, pan-cell RELA regions were mostly found near TNFα-upregulated genes (+/−10 kb of TSSs; approximately sixfold enrichment, $p < 1.0 \times 10^{-40}$; Fig. 4c) and were enriched for the canonical NF-κB motif (MEME $e$ value $= 1.9 \times 10^{-798}$; Fig. 4c). Indeed pan-cell RELA regions associated with immune response genes including many of the essential NF-κB signaling components (e.g., *NFKBIA*, *NFKBIB*, *IKBKG*, *TNFAIP3*, *IKBKE*; Fig. 4c, Supplementary Datasets 3 and 4). The four-cell-type-shared RELA peaks (pan-cell; $n = 782$) had the strongest RELA binding signal in each of the four-cell types (ChIP-seq reads, $p < 2.0 \times 10^{-05}$; Supplementary Fig. 3c) and were enriched for Mode P and Mode OA regions ($\sim 17$-fold, $p < 1.0 \times 10^{-47}$ and approximately twofold, $p < 1.0 \times 10^{-21}$, respectively, Supplementary Fig. 3d). In comparison to pan-cell type RELA binding, EC-specific binding events were enriched for Mode O regions ($n = 5683$, Fig. 4c and Supplementary Fig. 3d) and for ETS-like motifs (MEME $e$ value $= 1.6 \times 10^{-459}$, Fig. 4c and Supplementary Dataset 4) and AP-1 motifs (MEME $e$ value $= 9.7 \times 10^{-341}$, Supplementary Dataset 4) and were associated with EC pathways (i.e., angiogenesis; Genomic Regions Enrichment of Annotations Tool (GREAT) FDR $q$ value $< 6.5 \times 10^{-70}$, Fig. 4c and Supplementary Dataset 3). However, unlike Mode O regions, EC-specific Mode OA ($n = 561$) and Mode C regions ($n = 764$) were enriched for RELA motifs and TNFα-upregulated genes (Fig. 4c). These EC-specific Mode OA regions were enriched for pathways related to cytokine signaling and leukocyte migration and were found at inflammatory enhancers near EC-related genes such as *ICAM1* and *SELE* (Supplementary Fig. 3d, e).

The genome-wide chromatin accessibility and binding dynamics of RELA and the key endothelial ETS family member ERG were recently investigated in HAECs using ChIP-seq[15]. We noticed that pan-cell RELA peaks (both conserved and human-specific) show an increase in RELA binding, ATAC-seq, and H3K27ac signals at 45 min (our data) and 4 h[15] after TNFα stimulation (Supplementary Fig. 3f). In contrast, EC-specific RELA peaks did not show any further increase in RELA signal beyond 45 min and had a reduction in ATAC-seq signal and H3K27ac at 4 h relative to untreated HAECs (Supplementary Fig. 3f). Such antagonistic effects were more pronounced at conserved RELA-bound regions (Supplementary Fig. 3f). Taking into consideration the pan-cell type or EC-specific NF-κB peak classifications, we observed a decrease in ERG binding (9%, $p < 1.0 \times 10^{-4}$) at EC-specific regions, and an increase in ERG at pan-cell NF-κB regions (37%, $p < 1.0 \times 10^{-16}$, Fig. 4d). This data support both competitive (e.g., ERG near basal EC genes) and collaborative (e.g., ERG near pro-inflammatory EC genes) models of NF-κB-mediated gene regulation (Fig. 4e).

**Conserved NF-κB bound regions are a prominent component of inflammatory super-enhancers**. Clusters of strong enhancers, also known as super-enhancers (SEs), are prominent features of

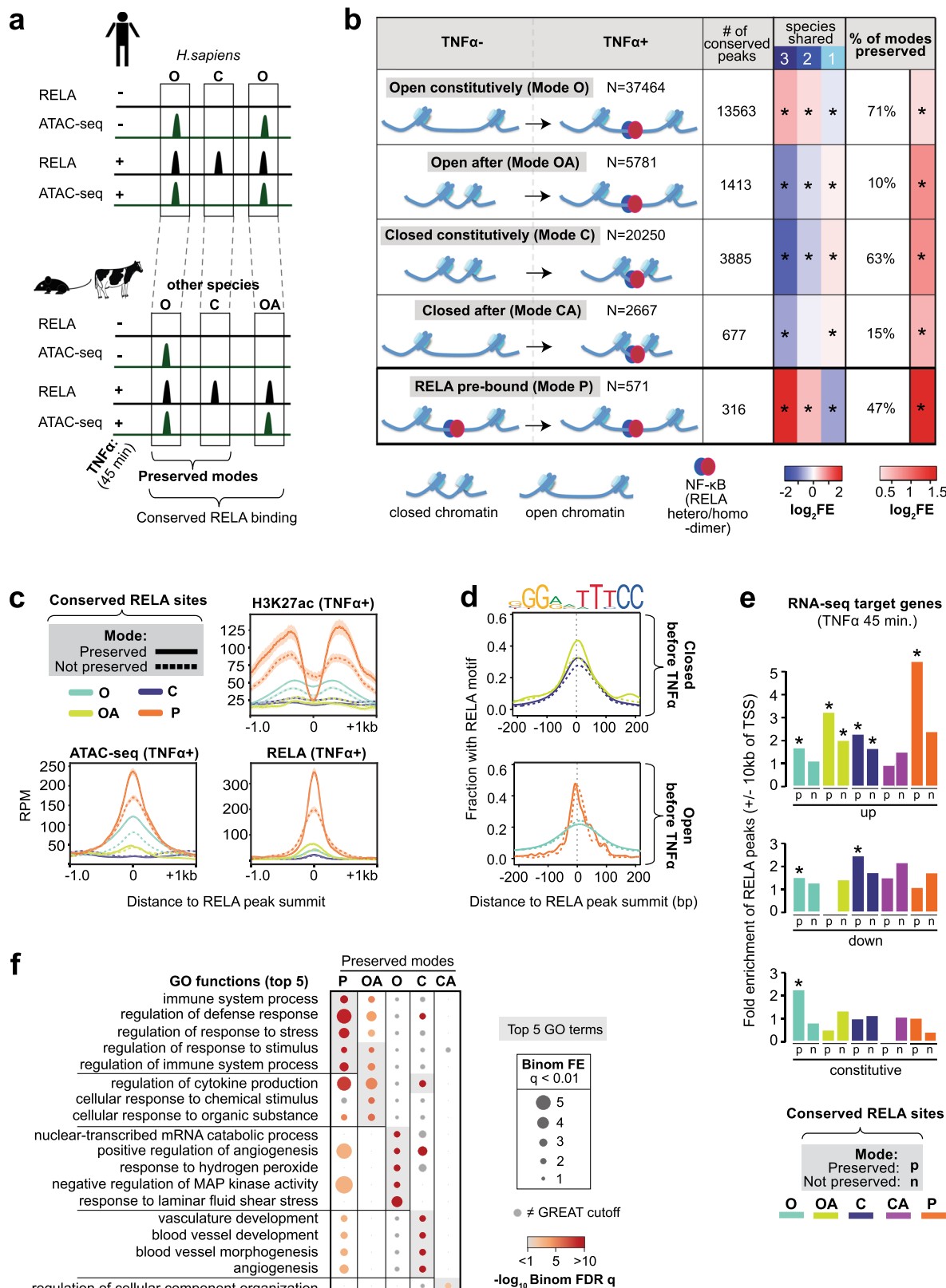

cell-type specific, inflammatory and disease-associated genes[66,67]. Inflammatory SEs have been shown to recruit transcriptional machinery at the expense of cell-identity genes[16,20] in a process referred to as SE-mediated cofactor squelching[20]. To examine the evolutionary conservation and epigenomic features of TNFα-

induced SEs, we used the ROSE algorithm to identify 'RELA SEs' from our RELA ChIP-seq data in TNFα-stimulated HAECs, MAECs, and BAECs (Fig. 5a). Supporting the functional relevance of human RELA SEs, they were associated with TNFα-upregulated genes (approximately sixfold, +/−10 kb of TSSs, $p <$

**Fig. 3 Conserved RELA-bound regions that also preserve their chromatin interaction mode across species show enhanced functional features. a** Schematic of conserved RELA peaks from a human genome perspective. Conserved RELA peaks with and without mode preservation are depicted. **b** Table indicating: (1) the number and the fold enrichment (FE) of conserved RELA peaks within each binding mode in HAECs ($p$ value: *$< 7.0 \times 10^{-5}$, Chi-squared test with Yates's correction for continuity of independence and Bonferroni correction for multiple testing), (2) the percentage and fold enrichment of mode preservation at conserved RELA peaks ($p$ value: *$< 4.0 \times 10^{-3}$, two-sided exact binomial test with Bonferroni correction). **c** Profile plots showing ATAC-seq and ChIP-seq (H3K27ac and RELA) signals in reads per million (RPM) mapped reads centered on conserved RELA peaks in human aortic ECs (HAECs) that have their mode of binding preserved (solid) or not-preserved (dashed). The plot lines indicate mean RPM ± SEM. **d** Density plots showing the fraction of RELA peaks harboring the canonical RELA motif (JASPAR - MA0107.1) as a function of distance from the RELA peak summits for conserved/preserved (solid) and conserved/not-preserved (dashed) binding modes. **e** Bar plots indicating the fold enrichment of conserved/preserved (p) and conserved/not-preserved (n) RELA binding modes within $+/-10$ kb of transcriptional start sites (TSS) of TNFα upregulated, TNFα downregulated, and constitutively expressed HAEC genes from RNA-seq analysis (the $p$ values were derived by comparing preserved and non-preserved peaks to the expected numbers using two-sided Fisher's exact test with Bonferroni correction for multiple testing: *$<0.04$). **f** Dot plot showing GO term enrichments (biological process) for preserved modes (GREAT analysis). Gray circles indicate the GO terms that did not pass the GREAT cutoffs (minimum region-based fold enrichment: 2; minimum term annotation count: 1; FDR $q$ value < 0.05; significant by both binomial and hypergeometric tests). The top five GO terms are ranked with binomial FDR $q$ value (highlighted in gray). Source data are provided in the Source Data file for Fig. 3b, e.

$1.0 \times 10^{-251}$; Supplementary Fig. 4a) and included well-characterized pro-inflammatory genes such as the chemokine *CCL2*, which was the top-ranked RELA SE in HAECs and MAECs (Fig. 5a). Approximately half of all human RELA SEs were conserved in two or more species ($n = 735$; Fig. 5a), compared to 30% of all RELA peaks. Three-species-conserved RELA SEs were significantly stronger than species-specific SEs (approximately twofold longer; $p < 7.0 \times 10^{-16}$ and approximately twofold higher RELA ChIP-seq read counts; $p < 10 \times 10^{-15}$; Supplementary Fig. 4b).

In general, inflammation-induced SEs were significantly enriched for conserved and pan-cell-type RELA peaks (1.7-fold $p < 5.0 \times 10^{-146}$ and 2.3-fold, $p < 5.0 \times 10^{-73}$, respectively; Fig. 5b). Moreover, two thirds of all Mode P regions resided within RELA SEs, showing approximately threefold enrichment ($p < 1.0 \times 10^{-170}$; Fig. 5c). Notably, Mode C regions were significantly enriched within SEs (1.2-fold, $p < 10 \times 10^{-53}$) and collectively make up approximately one third of all peaks within RELA SEs ($n = 4824$; Fig. 5c). This observation is consistent with the relatively high H3K27ac signal observed at the summits of Mode C RELA peaks (Fig. 2g).

SEs are often ascertained using ChIP-seq data for H3K27ac[68]. To put our RELA SEs in the context of H3K27ac SEs we called H3K27ac SEs in TNFα+ stimulated cells and compared them to our RELA SEs (Supplementary Fig. 4c). We found 1351 H3K27ac SEs of which 838 were common with RELA SEs (Supplementary Fig. 4c). The H3K27ac-RELA-common SEs were twofold enriched for three-species conserved RELA peaks ($p < 1.0 \times 10^{-177}$; Supplementary Fig. 4d) and 1.8-fold enriched near TNFα-upregulated genes ($+/-10$ kb of TSSs, $p < 1.0 \times 10^{-16}$; Supplementary Fig. 4e). In contrast, H3K27ac-specific SEs showed a 2.3-fold depletion of TNFα-upregulated genes ($+/-10$ kb of TSSs, $p < 2.0 \times 10^{-5}$; Supplementary Fig. 4e). To ask whether RELA-bound regions within RELA and H3K27ac ascertained SEs show different functional properties to RELA-bound regions outside of SEs, we looked at eRNA levels obtained from TeloHAEC ChRO-seq experiments 45 min after TNFα treatment (Supplementary Fig. 4f). Consistent with Fig. 1e, we see that the number of species where conserved orthologous RELA binding occurs corresponds to the level of ChRO-seq signal (Supplementary Fig. 4f). ChRO-seq signal at RELA-bound regions inside the H3K27ac-RELA-common SEs was higher than what was observed for RELA-bound regions outside of H3K27ac-RELA-common SEs ($p < 1.0 \times 10^{-16}$; Supplementary Fig. 4f).

Together, these results suggest that after TNFα treatment, NF-κB occupancy at relatively nucleosome-occluded regions is common, and that SEs are enriched for conserved orthologous and pan-cell type NF-κB binding.

**Top-ranked RELA peaks are conserved and sensitive to anti-inflammatory agents.** To investigate how NF-κB binding strength relates to function, we defined a group of "top-ranked" RELA peaks quantitatively by their ChIP-seq signal, which is a simpler way to identify regions with the functional properties of Mode P regions. We took all post-TNFα RELA peaks and ranked them by their ChIP-seq signal pre- and post-TNFα treatment and imposed an inflection point-based cutoff as is used for SE analyses. This returned 958 and 4412 top-ranked RELA peaks pre- and post-TNFα in HAECs, respectively (Fig. 5d). As expected, virtually all Mode P regions were found within the top-ranked RELA peaks pre-TNFα. Over 50% of these top-ranked RELA peaks in HAECs were conserved with either MAECs or BAECs (Fig. 5d) and they were approximately threefold enriched near NF-κB target genes ($p < 1.0 \times 10^{-14}$, Fig. 5e) and associated with GO terms related to "immune response" and "inflammatory response" (GREAT Binomial FDR < $1.0 \times 10^{-26}$, Supplementary Dataset 3). Similar results were observed for MAECs and BAECs top-ranked RELA regions, which represented ~7–8% of all RELA peaks and were conserved across species (>40%) (Supplementary Fig. 4g). Importantly, these regions were also top-ranked post-TNFα attesting to their high inducibility relative to other RELA peaks. Other NF-κB modes featured within the top-ranked post-TNFα regions including Mode OA ($n = 531$, 12%) and Mode C regions ($n = 187$, 4%).

We next assessed whether the top-ranked RELA regions are differentially targeted by therapeutic or anti-inflammatory factors. We first analyzed the recruitment of the transcriptional co-activator BRD4 at RELA peaks within SEs. BRD4 is an epigenetic reader that is recruited to RELA binding regions to induce pro-inflammatory gene expression in ECs[16]. The recruitment of BRD4 to RELA binding regions is diminished by administration of JQ1, a potent bromodomain inhibitor[16]. We found a greater reduction in mean BRD4 ChIP-seq signal in response to JQ1 treatment at the top-ranked RELA peaks (45%) compared to all RELA peaks (29%) (>1.6-fold difference, $p < 1.0 \times 10^{-26}$; Supplementary Fig. 4h), highlighting the potential therapeutic relevance of these binding regions.

Another example illustrating the functional relevance of the top-ranked RELA peaks can be seen during corticosteroid treatment, where the glucocorticoid receptor (GR) interacts with RELA binding regions in chromatin and antagonizes RELA function, thus contributing to its anti-inflammatory effects[69]. We determined if the anti-inflammatory GR is differentially recruited to the top-ranked RELA peaks. To this end, we reanalyzed GR ChIP-seq data performed in HeLa cells that were treated with the synthetic corticosteroid triamcinolone acetonide (TA) with or without TNFα exposure[69]. We found that the top-ranked

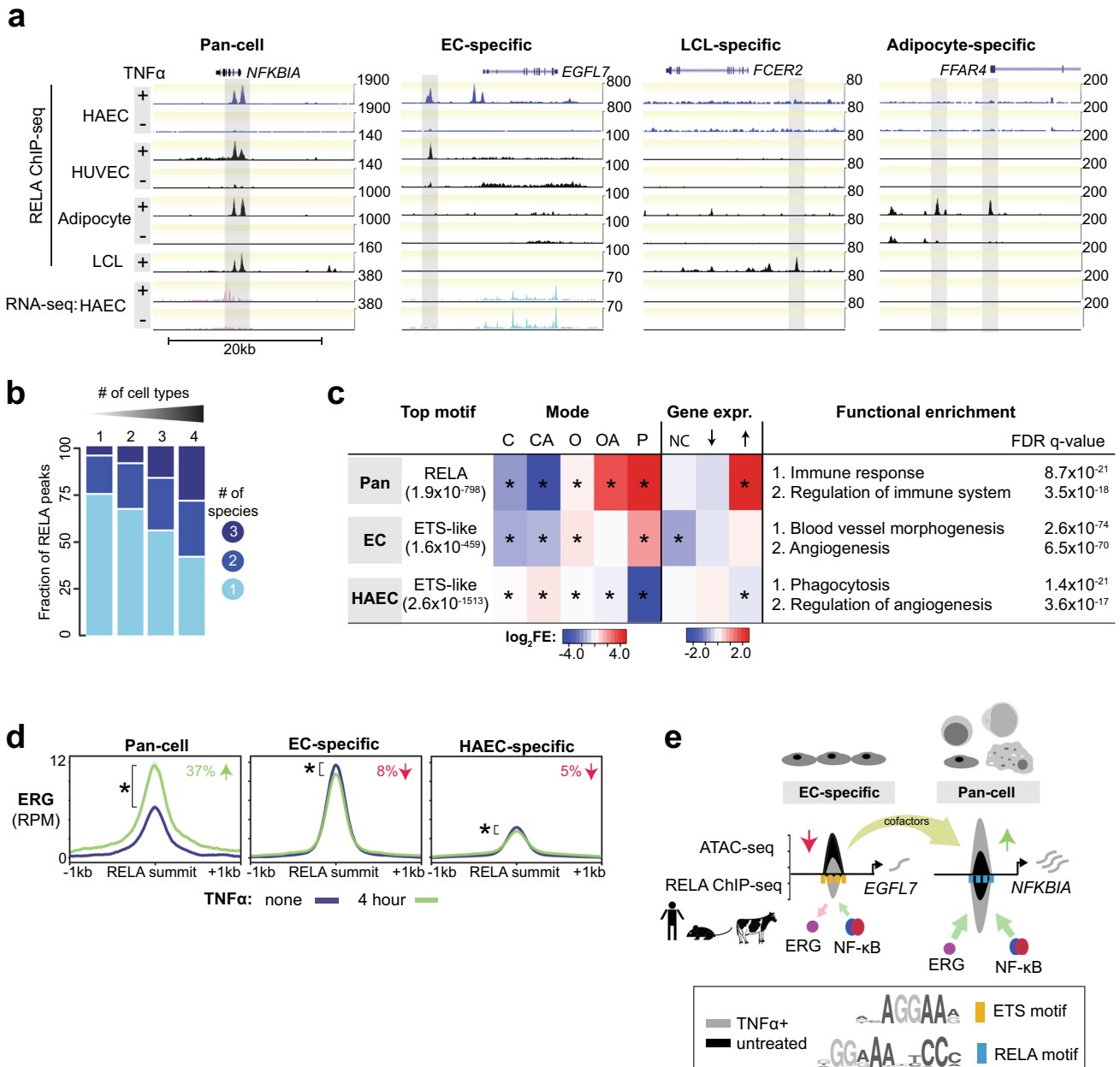

**Fig. 4 Conserved NF-κB binding is shared between cell types near common immune response genes. a** Representative genomic regions showing the pan-cell (four-cell-type-shared: between human aortic ECs [HAECs], human umbilical vein ECs [HUVECs], lymphoblastoid cell lines [LCLs], and adipocytes), EC-specific (shared only between HAECs and HUVECs), LCL-specific, and adipocyte-specific RELA peaks. **b** Stacked bar charts showing the fractions of four-, three-, and two-cell-type-shared and HAEC-specific RELA peaks (one-cell type) that are three-species-conserved, two-species-conserved, or human-specific. **c** Table showing the top-scoring de novo motifs (MEME-ChIP e values, Supplementary Dataset 4), fold enrichments of RELA binding modes (p value: *<1.0 × 10$^{-6}$, Chi-squared test with Yates's correction for continuity of independence and Bonferroni correction for multiple testing), fold enrichments near TNFα target genes (HAEC RNA-seq, ±10 kb of transcription start sites, p value: * <1.0 × 10$^{-5}$, two-sided Fisher's exact test with Bonferroni correction), and the top GO terms (GREAT FDR q values, Biological Process Ontology, Supplementary Dataset 3) for the pan-cell (four-cell-type-shared), EC-specific, and HAEC-specific RELA peaks. ↑ = upregulated, ↓ = downregulated, NC no change (i.e., constitutively expressed). **d** Profile plots comparing ERG binding signal at pan-cell, EC-specific, and HAEC-specific RELA peaks in HAECs. ERG ChIP-seq signal was generated using published raw data from ref. [15]. The plot lines indicate mean RPM ± SEM. Arrows and percentages indicate change in signal after TNFα stimulation. The p values were calculated using two-sided Welch Two Sample t-test (p value: *<0.05). **e**, Model of NF-κB-ERG dynamics and cofactor squelching showing redistribution of cofactors from EC-specific Mode O regions to pan-cell Mode P regions. Source data are provided in the Source Data file for Fig. 4c, d.

RELA-bound regions had significantly higher GR occupancy after TA treatment when compared to all RELA peaks (3.2-fold for the top-ranked pre-TNFα RELA peaks and 1.7-fold for the top-ranked post-TNFα peaks, ChIP-seq reads, $p < 8.0 \times 10^{-7}$; Fig. 5f). If we consider RELA-bound regions that are both top-ranked (pre-TNFα) and conserved, we observe a 4.4-fold enrichment of GR occupancy after TA treatment ($p < 4.0 \times 10^{-5}$). These results suggest that binding conservation together with NF-κB signal strength prior to TNFα stimulation is a simple and meaningful criterion for identifying functional NF-κB bound regions.

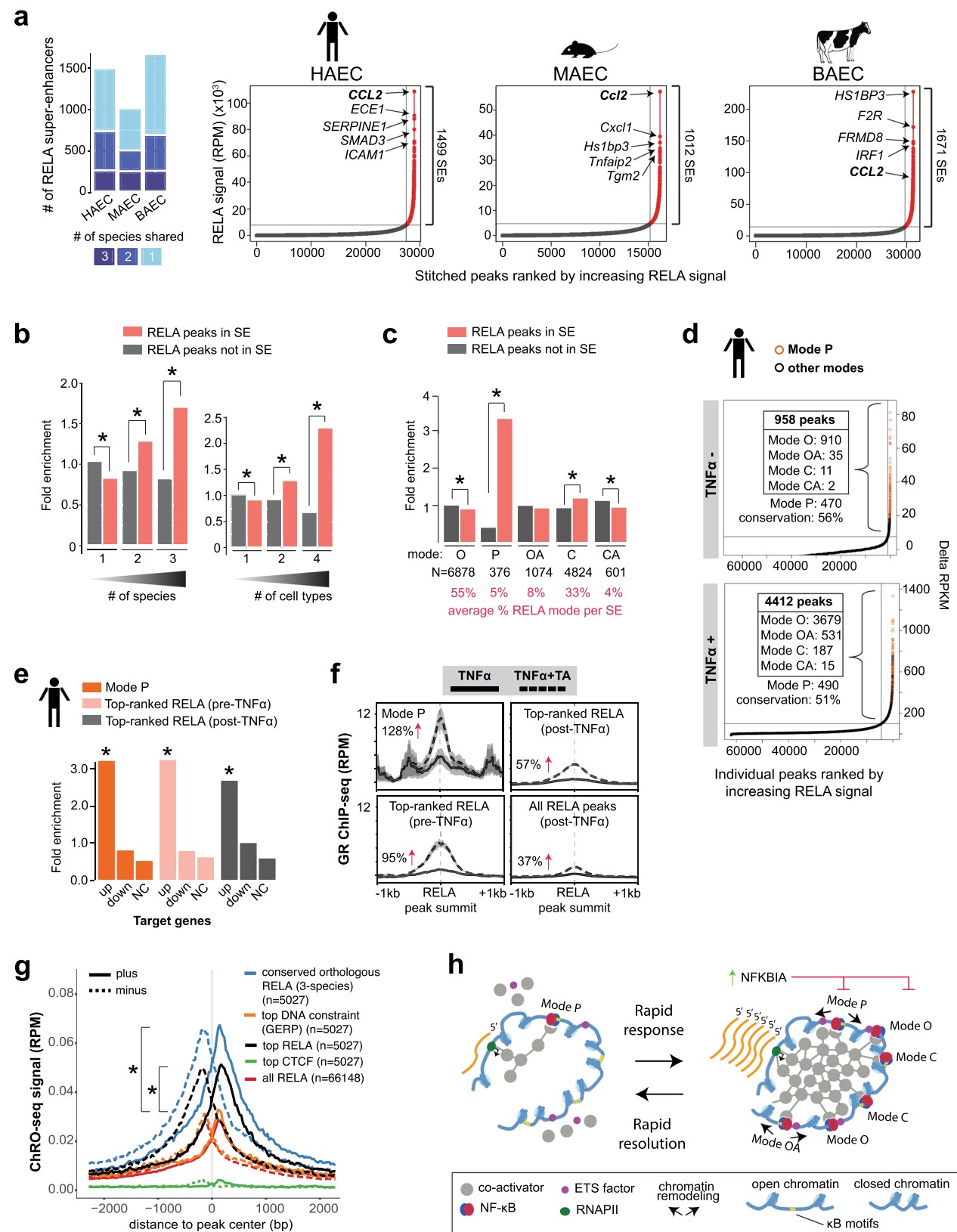

**Conserved and top-ranked RELA binding correspond to enhancer activity**. Using ChRO-Seq as a functional readout for enhancer activity, we compared the functional relevance of our three-species conserved RELA sites ($n = 5027$; see also Fig. 1) to an equivalent number of top-ranked RELA peaks and peaks with highest DNA constraint (average GERP score). Comparisons to

top-ranked CTCF peaks and all RELA peaks were also included as criteria that we would not expect to correspond with eRNA production measured by ChRO-seq. This analysis shows that the strongest ChRO-seq signal is found at three-species conserved RELA-bound regions (2.3-fold higher than all RELA peaks, $p < 2.2 \times 10^{-16}$), followed by top-ranked RELA peaks (1.6-fold higher

**Fig. 5 RELA signal strength and clustering in super-enhancers associates with conservation, pan-tissue activity, and pro-inflammatory functions. a** Stacked bars (left) showing fraction of RELA super-enhancers (SEs) that are conserved in TNFα-stimulated human, mouse, and bovine aortic ECs (HAEC, MAEC, BAEC) and plots (right) showing RELA SEs called by ROSE (red dots). **b** Fold enrichments of conserved (left) and cell-type-shared (right) RELA peaks within SEs (pink) and non-SEs (gray) in TNFα-stimulated HAECs (p value: *<$2.0 \times 10^{-35}$, Chi-squared test with Yates's correction for continuity of independence, Bonferroni correction). **c** Fold enrichments of RELA binding modes within SEs (pink) and non-SEs (gray) (p value: *<$1.0 \times 10^{-03}$, Chi-squared test with Yates's correction for continuity of independence, Bonferroni correction). Average percentage of RELA mode per SE is given. **d** HAEC RELA peaks ranked by normalized signal before and after TNFα stimulation. Mode P peaks are in orange. **e** Bar plots indicating fold enrichments of Mode P regions and the ranking of RELA peaks before (958 peaks) and after TNFα stimulation (4412 peaks) near TNFα target genes (+/−10 kb of TSS, HAEC RNA-seq; p value: *<$1.0 \times 10^{-8}$, two-sided Fisher's exact test with Bonferroni correction). Up, upregulated; down, downregulated and NC, no change (i.e., constitutively expressed) **f** Glucocorticoid receptor (GR) occupancy signal at Mode P and the top-ranked RELA peaks compared to average signals at all RELA peaks in HeLa cells. GR ChIP-seq signal was from published raw data[69]. Plot depicts mean RPM ± SEM. TA, triamcinolone acetonide. **g** Profile plot showing ChRO-seq signals (mean RPM) centered on the three-species conserved RELA peaks (n = 5027), and an equivalent number of top-ranked RELA peaks, the peaks with highest DNA constraint (average GERP score), and CTCF peaks. Average ChRO-seq signal for all RELA peaks is shown as a baseline reference. (p value: *<$2.2 \times 10^{-16}$, two-sided Mann–Whitney U test with Bonferroni correction). **h** Model of how NF-κB modes could function within the context of a super-enhancer taking into account co-activator squelching and phase transition dynamics. Connections between components represent potential physical interactions. Source data are provided in the Source Data file for Fig. 5b, c, e.

---

than all RELA peaks, $p < 2.2 \times 10^{-16}$) (Fig. 5g). Both of these enrichments were greater than what was found by taking the RELA peaks with the highest DNA constraint (top GERP vs. all: 1.1-fold, $p = 0.305$; Fig. 5g).

When considering these results in the context of clustered NF-κB binding found within SEs, we can build upon the existing models of inflammatory gene regulation. Existing models demonstrate the importance of clusters of NF-κB binding to accessible chromatin (with and without canonical NF-κB motifs). We add to this model by suggesting that a minority of NF-κB bound regions within a SE may play a disproportionate role in the activation of inflammatory gene expression (e.g. conserved Mode P regions) and that nucleosome-occluded Mode C and OA regions, also play a role in recruiting NF-κB at high concentrations of NF-κB (i.e., immediately following inflammatory stimulus) through binding to multiple canonical NF-κB motifs (Fig. 5h).

**Testing the function of conserved orthologous NF-κB binding within super-enhancers.** Individual SE components can work in additive, hierarchical, or redundant manners to regulate gene expression[66,70–73]. To test the function of Mode P regions within the context of EC SEs, we first focussed on the top-ranked human SE (chr17: 32,562,086-32,586,537), which we also found to be the top SE in our mouse RELA data. The RELA-bound regions at the CCL2 locus have been previously shown to interact with the CCL2 promoter in IMR90 cells, and these interactions are stable pre- and post-TNFα treatment[74]. CCL2 encodes monocyte chemoattractant protein-1, a member of the CC chemokine family, which is known for its crucial role in regulating monocyte chemoattraction. Genome-wide association studies (GWASs) have linked genetic variation at the CCL2 locus to coronary artery disease and other inflammation-related phenotypes and diseases[75,76]. The CCL2 SE contains three conserved RELA-bound regions (RELA peaks #1, #2, and #6) that are Mode P in HAEC and four conserved regions (RELA peaks #3, #4, #5, and #7) that are Mode O in HAECs, located upstream of CCL2 (Fig. 6a). The conserved RELA peak #6 (post-TNFα state, chr17: 32,579,059-32,580,529) was notably preserved as Mode P in all three species (Fig. 6a) and ranked as the peak with the 4th strongest RELA ChIP-seq signal post-TNFα treatment. This region also harbors a GWAS variant (rs1024611; −2578 A/G polymorphism site) that has been associated with atherosclerosis and other inflammation-related diseases[77–79] (Fig. 6a).

Using 4C-seq in HAECs, we found that the CCL2 promoter shows the strongest interactions within the SE and lesser interactions with regions near other CCL genes in that locus

(Supplementary Fig. 5a). 4C-seq in TeloHAECs recapitulated the CCL2 promoter interactions seen in HAECs (Supplementary Fig. 5a). Consistent with previous observations in IMR90 cells[74], we did not see significant changes in CCL2 promoter contacts in response to TNFα treatment in HAECs (Supplementary Fig. 5a). Messenger RNA copy-number quantification revealed CCL2 to be by far the most abundant transcript in this locus (>1000-fold higher at 3-h TNFα, Supplementary Fig. 5c).

Deletions of the individual conserved Mode P and Mode O RELA peaks in TeloHAEC clones revealed their distinct effects on CCL2 regulation (Supplementary Fig. 5d–f). Deletions of the RELA peaks #1 (Mode P), #2 (Mode P), and #3 (Mode O) resulted in 47% ($p = 0.0011$), 48% ($p < 0.0007$), and 38% ($p = 0.0009$) decrease in CCL2 expression under basal conditions, respectively (Fig. 6b and Supplementary Fig. 5e). In contrast, deleting the RELA peak #4 (Mode O) increased the expression of CCL2 by 215% ($p = 0.0023$), indicating a potential repressive action (Fig. 6b and Supplementary Fig. 5e). Strikingly, deleting the RELA peak #6 (Mode P) (Supplementary Fig. 5d) reduced CCL2 transcription by 95% ($p < 0.0001$) under basal conditions (Fig. 6b and Supplementary Fig. 5e). Similar effects were observed under TNFα stimulation conditions—peak #1: 14% decrease ($p = 0.002$); peak #2: 27% decrease ($p = 0.0042$); peak #3: 9% decrease ($p = 0.0602$); peak #4: 46% increase ($p = 0.0253$); and peak #6: 90% decrease ($p < 0.0001$) (Fig. 6b and Supplementary Fig. 5e). Deleting the RELA peak #6 (Supplementary Fig. 5d, e) still had a stronger effect than the combined ~8.5-kb deletion encompassing RELA peaks 1–4 (basal conditions (49%, $p = 0.0003$) and TNFα-stimulated conditions (60%, $p = 0.0113$; Fig. 6b). The effect of the RELA peak #6 deletion on reduced CCL2 expression was also significant at 6-h and 24-hr post-TNFα stimulation (Supplementary Fig. 5f). These results suggest the primacy of the region encompassing RELA peak #6 in CCL2 regulation.

To investigate the role that NF-κB plays in CCL2 expression within the context of RELA peak #6, we identified three full-length RELA motifs that were conserved in all three species (Fig. 6c). Using CRISPR/Cas9-based homologous recombination, we replaced the endogenous enhancer with a mutated version where all three conserved RELA motifs were scrambled (Fig. 6c). We observed a dosage-sensitive decrease in CCL2 expression with a 55% reduction ($p = 0.019$) under TNFα-stimulated conditions when the mutation was homozygous (Fig. 6c). Several non-conserved RELA motifs and multiple AP-1 motifs were still present, which may explain the less potent effect of mutation compared to deleting the entire region. Many other TFs such as IRF1, PKNOX1, and PBX2 have also been shown to bind to RELA peak #6 at the −2578 A/G polymorphism site (rs1024611)

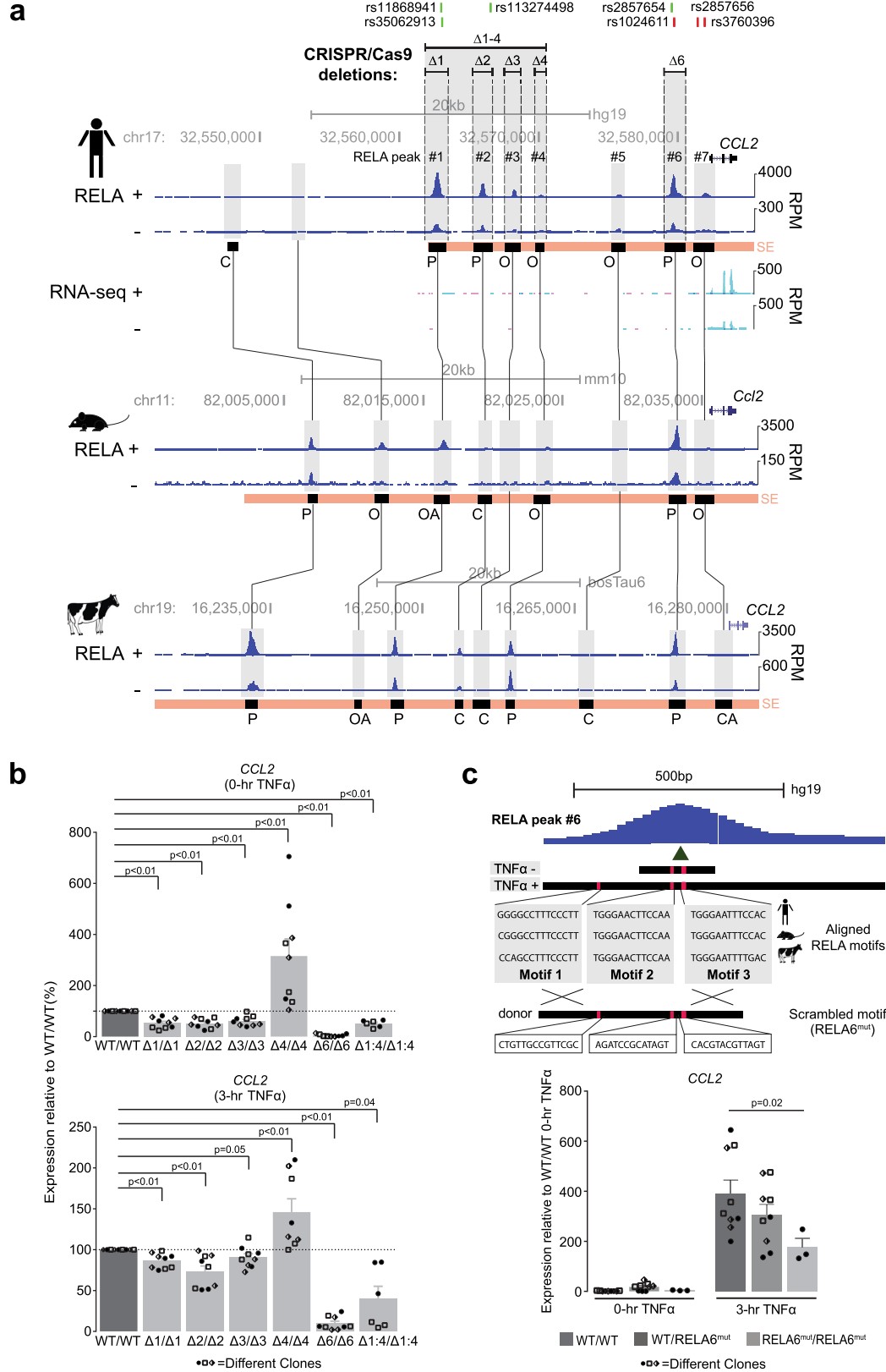

and are implicated in regulation of *CCL2* expression[80,81]. Therefore, a cumulative effect of multiple TFs at this enhancer is likely to contribute to the regulation of *CCL2* expression[77].

The dissection of the *CCL2* SE confirmed that conserved Mode P peaks within SEs may play a primary role in enhancer activity, and identification of these regions could help to discern the

functional regulatory elements from the vast pool of weakly active and redundant RELA peaks. Our earlier results indicated that many ($n = 226$) Mode P peaks were distal intergenic (Supplementary Fig. 2j) suggesting the involvement of long-range interactions in regulation of pro-inflammatory gene expression. To test this, we deleted a three-species conserved and preserved

**Fig. 6 CRISPR/Cas9-mediated genomic deletions of *CCL2* super-enhancer components reveal principal roles of conserved RELA prebound regions in gene expression. a** Evolution of the individual RELA super-enhancer (SE) components at the *CCL2* locus. The orthologous sequences in human, mouse, and cow ECs are connected with black vertical lines and highlighted in gray. RNA-seq tracks in human aortic ECs (HAEC) are shown. The SE regions are shown with pink bars, RELA peaks are indicated with black bars and modes of binding are indicated with their corresponding letters. The genomic regions deleted with CRISPR/Cas9 in telomerase-immortalized aortic ECs (TeloHAECs) are shown with horizontal lines at the top. The human phenotype-associated common genome-wide association study (GWAS) variants (green bars) and disease-linked human gene mutation database (HGMD) variants (red bars) are shown at the top. **b** Bar charts showing *CCL2* expression (RT-qPCR, ΔΔCT) before and after TNFα induction (3 h, 10 ng/mL) in TeloHAEC clones that are homozygous for the RELA binding region deletions shown in **a**. The data are plotted relative to wild-type (WT/WT) TeloHAEC clones. Each data point shape (circle, square, rhombus) represents data from each of the three tested deletion clones (*p* values were derived using a two-sided ratio *t*-test). Error bars represent SEM. **c** Schematic (top) depicting the replacement of three conserved NF-κB motifs within the conserved RELA peak #6 of the *CCL2* SE with a donor sequence harboring the scrambled motifs using CRISPR/Cas9-mediated homologous recombination (HR). Bar charts (bottom) comparing *CCL2* expression (RT-qPCR, ΔΔCT) between wild-type TeloHAECs (WT/WT, dark gray), heterozygous (WT/RELA6^mut, gray) and homozygous (RELA6^mut/RELA6^mut, light gray) TeloHAEC clones that harbor the scrambled motifs. Each data point shape (circle, square, rhombus) represents data from each of the three tested deletion clones. The data are plotted relative to the unstimulated wild-type (WT/WT 0 h) TeloHAECs (*p* values were derived using one-way ANOVA with Bonferroni correction). Error bars represent SEM. Source data are provided in the Source Data file for Fig. 6b, c.

Mode P peak (chr5: 57,537,111-57,537,790; summit at chr5: 57,537,426) within a distal intergenic SE (chr5: 57,535,517-57,542,662) that was ~218 kb away from the TNFα-upregulated gene Polo Like Kinase 2 (*PLK2*) (Supplementary Fig. 6a). Deleting this conserved Mode P peak (chr5: 57,537,426) resulted in a 28.9% (*p* = 0.0026) decrease in *PLK2* expression under TNFα stimulated conditions (Supplementary Fig. 6a). Next, we tested another three-species conserved and preserved Mode P peak (chr19: 13,949,622-13,951,172; summit at chr19: 13,950,068) within the SE (chr19: 13,941,724-13,977,078) located near the TNFα-upregulated gene, Zinc Finger SWIM-Type Containing 4 (*ZSWIM4*) (Supplementary Fig. 6a). Contrary to our expectations, deleting this conserved Mode P region (chr19: 13,950,068) did not affect *ZSWIM4* expression, but did significantly regulate the expression of Coiled-Coil and C2 Domain Containing 1A (*CC2D1A*), a gene which is located ~65 kb away (67.3% decrease, *p* = 0.0004, Supplementary Fig. 6b, c). Altogether, our results indicate that conserved NF-κB bound regions that tend to be occupied by NF-κB in the absence of stimulation (Mode P or top-ranked pre-TNFα RELA peaks) likely constitute a core set of NF-κB elements that are utilized in many cell types to control proximal and distal TNFα-induced gene expression.

**Conserved NF-κB peaks are enriched for noncoding inflammatory disease mutations.** Mutations affecting the protein coding regions of NF-κB signaling pathway components have been implicated in the pathogenesis of many complex diseases[2,3,82]. However, the majority of genetic variants associated with common diseases fall within noncoding DNA[83]. Previous studies have shown that conserved TF peaks are enriched for noncoding disease mutations in biologically relevant pathways[40]. This prompted us to test how conservation and other NF-κB properties (i.e., pan-cell-type and top-ranked RELA-bound regions) correspond to noncoding disease mutations and polymorphisms. We overlapped our dataset with the Human Gene Mutation Database (HGMD; 2019 version) and found 699 noncoding disease mutations within 318 RELA peaks (Supplementary Dataset 7). The RELA peaks harboring disease mutations were enriched within three-species conserved RELA peaks (approximately threefold, *p* < 1.0 × 10^{-28}), within the four-cell-type-shared set of RELA peaks (approximately sixfold, *p* < 1.0 × 10^{-23}), within the Mode P peaks (approximately fourfold, *p* < 8.0 × 10^{-7}), within the top-ranked RELA peaks (approximately threefold, *p* < 9.0 × 10^{-3}), and within SEs (~1.2-fold, *p* < 8.0 × 10^{-4}) (Fig. 7a). In particular, the noncoding mutations (*n* = 299) overlapping the three-species conserved RELA peaks (*n* = 78) were associated with 74 genes in pathways related to angiogenesis and the immune response (Fig. 7b), and

were linked to disease conditions including coronary artery disease and rheumatoid arthritis (Supplementary Dataset 7). Similarly, we observed 178 noncoding mutations within 58 of the top-ranked RELA peaks that were linked to 56 genes involved in cell chemotaxis, cytokine production, and inflammation (Supplementary Fig. 7), and conditions such as inflammatory bowel disease and systemic lupus erythematosus (Supplementary Dataset 7). While each of the categories significantly enriched for RELA peaks harboring noncoding HGMD polymorphisms or disease mutations a combination of the three criteria (top-ranked, three or more cell types, and three-species; *n* = 570) reveal 16-fold enrichment (*p* < 1.0 × 10^{-148}, *n* = 46) of RELA peaks harboring noncoding variants that were associated with acute inflammatory pathways (Supplementary Dataset 7). These results demonstrate both the overlapping nature and functional relevance of our NF-κB categories. Overall, our results identify RELA peaks that are disease-sensitive hotspots susceptible to a range of acute and chronic pro-inflammatory conditions across cell types.

**Genetic variants with pleiotropic associations coincide with conserved, pan-cell, and top-ranked NF-κB peaks.** We next asked if our functional categories of RELA binding (i.e., conserved, pan-cell-type, and strongest binding) significantly intersect with the established risk loci for particular diseases and traits. To this end, we used the Regulatory Element Locus Intersection (RELI) approach[84], which compares the observed versus expected number of intersections between the genomic coordinates of genomic features (e.g., RELA ChIP-seq peaks) and genetic variant associations identified through GWAS[85]. When restricting RELA peaks using our major functional categories (i.e., conservation, sharing across cell-types, or top-ranking RELA peaks), we observed significant enrichments for 125 GWAS reports (*p* < 0.01, fold enrichment ≥ 2, and number of overlapping SNPs > 1; Fig. 7c). These 125 GWAS reports could be broadly classify into 18 phenotypes (Supplementary Dataset 8). The most prevalent phenotype involved cell counts for a wide variety of blood cell types (*n* = 29), inflammatory-related disease (*n* = 22), cancer phenotypes (*n* = 10), and cardiovascular disease (*n* = 8). Overall, there were 3072 SNPs within 2142 RELA peaks, 44% of which were conserved.

We found abundant RELA peaks containing SNPs associated with inflammatory phenotypes (Fig. 7c and Supplementary Dataset 8). For example, the essential *CCL2* enhancer #6 peak contains rs1024611, which was among the SNPs reported by a highly powered study examining pleiotropic genetic variation in five chronic inflammatory diseases[86] and enriched for top-ranked pre-TNFα treatment RELA peaks (sixfold, *p* < 1.0 × 10^{-19},

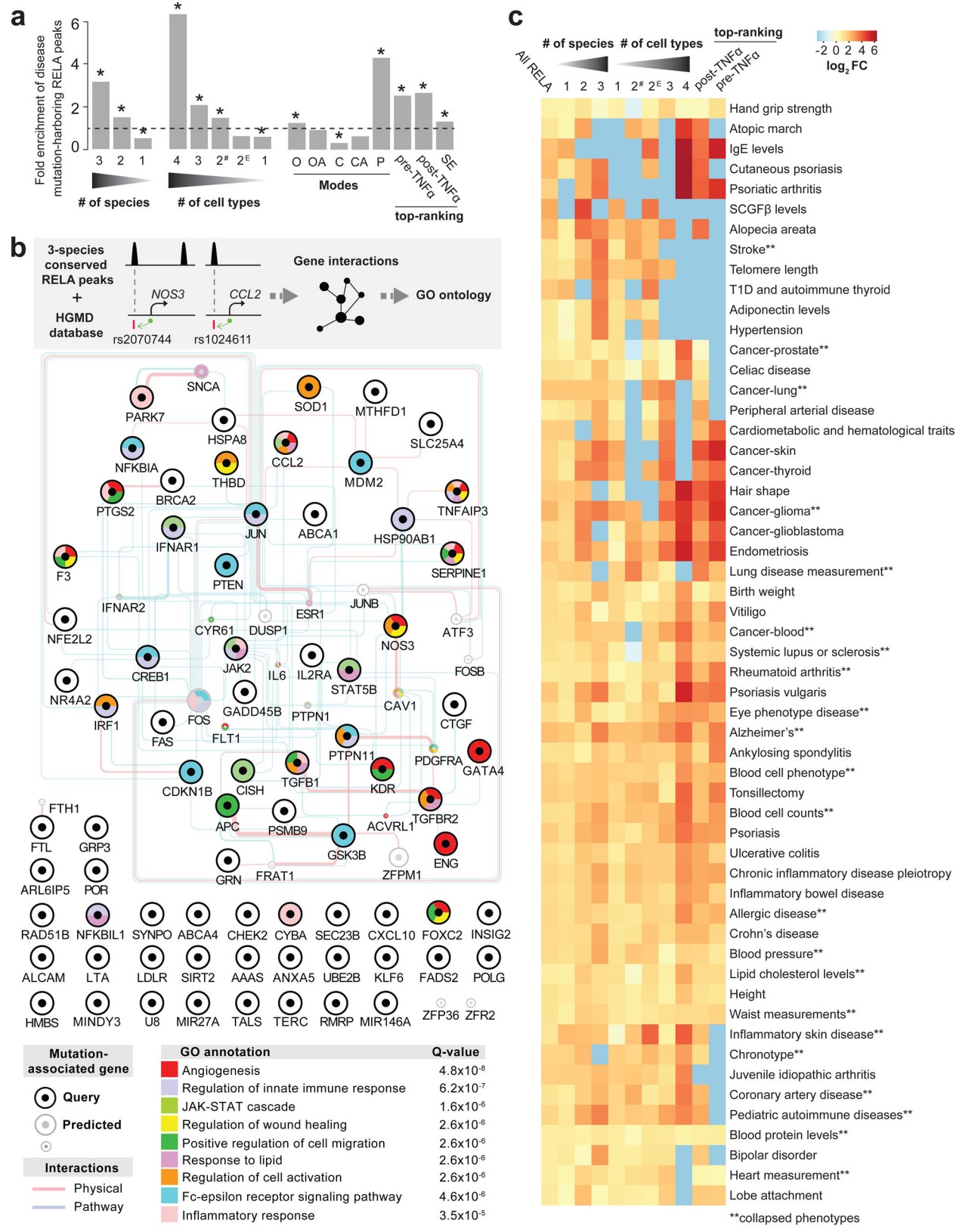

Supplementary Dataset 8). Another exemplary, functionally assayed pleiotropic SNP is rs17293632[87], which is located within intron 1 of *SMAD3*[87]. This SNP falls within a three-species conserved, top-ranked pre- and post-TNFα RELA peak, and has been associated with coronary artery disease[88], allergic disease[89], pleiotropic chronic inflammatory disease[86], IBD (14 GWAS

studies), pediatric autoimmune disease, thyroid cancer (5 studies), and ulcerative colitis (12 studies) (Supplementary Dataset 8).

Multiple cancer phenotypes were significantly enriched for conserved, pan-cell type, and top-ranked pre-TNFα RELA peak categories (Fig. 7c and Supplementary Dataset 8). For example, 12 SNPs overlapping 8 RELA peaks were associated with glioma[90],

**Fig. 7 Noncoding disease mutations associate with conserved and prebound regions in pathways and diseases related to inflammation. a** Bar charts showing fold enrichment of RELA peaks harboring noncoding disease variants (Human Gene Mutation Database [HMGD]) within different types of RELA peaks (+/−10 kb of transcriptional start sites (TSS). $2^{\#}$ = 2-cell-type-shared non-EC-specific RELA peaks, $2^{E}$ = EC-specific RELA peaks, SE = super-enhancer. p value: *<$9.0 \times 10^{-3}$, Chi-squared test with Yates's correction for continuity of independence, Bonferroni correction for multiple testing. Source data are provided in the Source Data file. **b** Network showing physical and pathway interactions (GeneMANIA) of genes that are linked to the noncoding disease mutations (HMGD) residing within the three-species conserved RELA peaks. The top-scoring nonredundant GO terms are shown (colors). **c** Enrichment of genome-wide association study (GWAS) variants for conserved, cell-type specific and top-ranked RELA peak classes. Results were filtered such that only phenotypes with significant results (adjusted $p < 0.01$, fold change ≥ 2) in at least one RELA dataset were included in the final matrix, which was visualized as a heatmap of enrichment ($\log_2$-fold change between observed and expected intersection counts is shown). Note that for certain identical or highly related phenotypes, we only show the results from the study reporting the highest number of single nucleotide polymorphisms (SNPs) (see "Methods" and Supplementary Dataset 8 for the complete list and data). This heatmap matrix was hierarchically clustered on the phenotype axis using average distance and the Canberra metric.

which was 34-fold enriched ($p < 1.0 \times 10^{-12}$) by the top-ranked pre-TNFα RELA category (Supplementary Dataset 8). Three of these glioma-associated SNPs were pleiotropic: *RTEL1* (rs1291209: allergic disease asthma hay fever or eczema), *CDKN2B* (rs6475604: coronary heart disease, glaucoma, vertical cup disc ratio), and *PHLDA1* (rs1565765: atrial fibrillation) (Supplementary Dataset 8).

We also observed phenotypes that were not enriched by the top-ranked pre-TNFα peak category but were still significantly enriched by the three-species conserved category (Fig. 7c). These phenotypes included stroke, hypertension, adiponectin levels, telomere length, alopecia areata and type I diabetes, and autoimmune thyroid disease phenotypes (Fig. 7c). For example, a unique enrichment for stroke phenotypes[91] was observed for the three-species category (13-fold, $p < 1.0 \times 10^{-6}$). This enrichment was due to SNPs at three loci: rs10776752, an EC-specific peak in the *WNT2A* intron 1; rs10786772, a HAEC-specific peak found at the *NEURL/SH3PXD2A* locus, as well as rs1537373, a HAEC/adipocyte peak also near *CDKN2B*, which has been functionally assayed and implicated as a pancreatic cancer susceptibility SNP[92] (Supplementary Dataset 8).

Overall these results suggest that classifying RELA binding by evolutionary conservation, binding mode and intensity, and cell-type specificity (i.e., pan-cell-type or cell-type specific) can identify regions of the genome where genetic variation impacts diverse phenotypes related to inflammation and NF-κB biology.

## Discussion

Since its discovery over 30 years ago, the gene-regulatory mechanisms behind the paradigmatic rapid NF-κB response have been intensively studied. Humans and mice lacking core components of the NF-κB signaling pathway demonstrate its complex role in diverse disease phenotypes[2]. Mechanistic and structural studies of NF-κB-mediated gene regulation have elucidated core and essential features of NF-κB–DNA interactions. Nonetheless, a remaining challenge that prevents a more detailed understanding of NF-κB is determining which of the more than 100,000-documented NF-κB–chromatin interactions are functional. In this study we addressed this question by performing a comparative epigenomic analysis of the NF-κB response in primary ECs isolated from three mammalian species treated with the pro-inflammatory cytokine TNFα. We identified thousands of conserved orthologous NF-κB binding events, many of which have also preserved the chromatin context in which they bind and overlap genomic regions already implicated in inflammatory diseases and phenotypes.

One of the most striking and conserved modes of NF-κB–chromatin binding occurred in regions that showed relatively high levels of RELA occupancy under basal conditions (Mode P and more broadly, RELA peak region ranked by RELA ChIP-seq signal pre-TNFα treatment). These ~1000 top-ranked NF-κB

peak regions are characterized by multiple canonical RELA motifs in the context of highly accessible chromatin. The efficient recruitment of NF-κB to these bound regions under basal conditions, their appearance in proximity to TNFα-induced gene expression changes and strong binding signals across species and cell types suggest that a small number of conserved NF-κB binding regions have a disproportionate impact on pro-inflammatory gene regulation. However, given the large number of Mode O peaks (>35,000) occurring over a quantitative continuum of binding, and at varying proximities to target genes, it is important to acknowledge that many crucial responses to TNFα will be mediated by Mode O regions that are not captured by simply ranking RELA signal under unstimulated conditions.

In agreement with previous studies[15,18,26,30,49,93], we readily identified two prevalent known modes of NF-κB binding: (1) to constitutively accessible chromatin that is pre-established by lineage determining factors (Mode O); and (2) to late accessible chromatin regions (Mode OA). Although a third mode had been postulated—binding of NF-κB to inaccessible chromatin (Mode C)[32,34,35]—its functional relevance has remained unclear. Our comparative epigenomics analyses suggest that this mode is an integral part of the NF-κB response, with almost 2500 conserved peaks where NF-κB was bound to nucleosome-occluded DNA in at least one other species (Mode C). Supporting their functional relevance, and implicating them as players in the proposed cofactor squelching and phase separation models of acute inflammation[16,20,67], we found that one third of NF-κB peaks within SEs occurred in regions of high nucleosome occupancy (Mode C). While these elements await further functional testing, some Mode C peaks ($n = 38$) overlap noncoding disease-linked variants (HGMD), including the loci of the TNFα-induced genes *TNFAIP2* and *TNFAIP3*. Our working model of the potential function of these Mode C peaks is that they are a reservoir of "vacant," NF-κB binding sites that further enhance rapid TNFα-induced phase transitions (Fig. 5h)[67]. In this scenario, NF-κB binding would be predicted to be suboptimal, yet favorable under the high concentrations of nuclear NF-κB that prevail during the inflammatory response. This binding should also be dependent on the relative positioning of the RELA motifs within the nucleosome dyad as has been shown in vitro for NF-κB[34] and more generally for other TFs that bind to nucleosome-occluded DNA[94].

In contrast to studies demonstrating clear pioneering activity of specific TFs (e.g., the de novo formation of GATA3-bound enhancers at closed chromatin during the mesenchymal to epithelial transition[95,96]), these Mode C regions are an example of non-pioneering TF binding to inaccessible chromatin. While our NF-κB mode classifications were often preserved between species and supported by data from other groups that used distinct experimental protocols for assessing chromatin accessibility (DNase-seq, ATAC-seq and MNase-seq), it is important to

recognize that all TF–chromatin interactions occur along a continuum of accessibility states, which exist at different frequencies within cell populations and between cell types[97]. Thus, it is likely that some of the Mode C regions with weak RELA ChIP-seq signal occurred at regions that were infrequently nucleosome depleted. While in vitro evidence for NF-κB-nucleosome interactions has been previously shown[31–34], structural studies, such as the recent cryo-electron microscopy results showing interactions between GATA3 and the nucleosome[96], will be invaluable for understanding the precise nature of Mode C regions.

The robustness, precision, specificity, and evolution of gene regulation rely on both high- and low-affinity TF binding motifs[98,99]. We found that the strongest NF-κB binding regions are the ones that possessed a myriad of functional properties: conserved orthologous binding, conserved binding mode, usage in multiple cell types, multiple canonical NF-κB motifs, proximity to target genes, residency in super-enhancers, increase in chromatin accessibility post-TNFα stimulation, overlap with genetic variations associated with inflammatory disease, and functional changes in gene expression after genetic ablation. Given our observations regarding the conservation and functional properties of pan-cell type binding, it is reasonable to expect that our observations reflect the NF-κB response in other cell types and in the presence of other stimuli. However, it is important to note that these results are in contrast to what has been observed for many developmental enhancers, which utilize weaker TF binding but optimal syntax to control spatial-temporal gene expression during development[100]. Such a phenomenon was also observed in a comparative analysis of liver master regulator TF binding (HNF4A, CEBPA, FOXA1, and ONECUT1) in the livers of five species. Unlike our observations for NF-κB, the top-ranked liver cis-regulatory modules with the highest ChIP-seq signal showed only modest enrichments for liver pathways relative to what was seen for an equivalent number of conserved orthologous cis-regulatory modules[40].

Our results are consistent with the additive model of NF-κB response (i.e., analog response) that has been previously shown to be mediated by multiple NF-κB motifs within individual enhancers and exemplified at the pro-inflammatory gene *NFKBIA*[60]. NF-κB binding via multiple canonical motifs facilitates non-cooperative binding that allows the NF-κB response to be proportional to the concentration of NF-κB in the nucleus[60]. This model is contrasted with a "digital" on/off response seen in developmental transcriptional regulation when sharp boundaries/transitions are required during embryonic development[60]. However, this model does not necessarily correspond to how multiple RELA binding regions interact within the context of inflammatory super-enhancers. Here we show that super-enhancers themselves are enriched for conserved orthologous RELA binding regions. Using the disease-associated *CCL2* super-enhancer as an example, we see both additive and epistatic effects of its constituent NF-κB-bound enhancers. Notably, deleting one human Mode P peak (enhancer #6), which was also conserved as a top-ranking (Mode P) RELA peak in mouse and cow, ablated *CCL2* induction, suggesting epistatic or "digital" on/off control of the super-enhancer. At the same time, other more distal conserved orthologous RELA peaks (enhancers #1, 2, and 3) showed an additive effect on *CCL2* gene expression. By deleting additional conserved Mode P peaks within two other super-enhancers, we found marked changes in nearby—but not necessarily the nearest —genes. This underscores the importance of understudied distal NF-κB binding regions in gene regulation.

Overall our comprehensive cross-species and cross-cell-type epigenomic analysis of the acute inflammatory response mediated by TNFα highlights robust principles of inflammatory enhancers. Importantly, the NF-κB modes we identified are relevant beyond conserved endothelial enhancers. The thousands of NF-κB peaks we charted, many of which coincide with human genetic variation associated with disease phenotypes, represent a conserved core of regulatory elements that play a principal role orchestrating the mammalian NF-κB response.

## Methods

**Cell culture**. Primary ECs isolated from aortas of human (two 21 year old Caucasian males, lot# 2139 and 1487; a 15 year old male, lot# 2102; and a 60 year old male, lot# 2366; Cell Application cat# 304-05a), mouse (two biological replicates of C57BL/6 males pooled from multiple mice, Cell Biologics cat# C57-6052, lot# A092913T2MP and B092913T2MP), and cow (two biological replicates, Cell Applications cat# B304-05, lot# 1165 and 1190) were thawed into T75 cell culture flasks and then further grown in T225 flasks at 37 °C and 5% CO₂ in Endothelial Cell Growth Media MV2 (PromoCell) supplemented with 5% fetal calf serum (PromoCell), 5-ng/ml recombinant human epidermal growth factor (PromoCell), 0.5-ng/ml recombinant human vascular endothelial growth factor 165 (Promo-Cell), 10-ng/ml recombinant human basic fibroblast growth factor (PromoCell), 20-ng/ml long R3 insulin-like growth factor-1 (PromoCell), 1-µg/ml ascorbic acid (PromoCell), and 0.2-µg/ml hydrocortisone (PromoCell). All experiments were carried out before passage 9. Telomerase-immortalized aortic ECs (TeloHAECs, ATCC CRL-4052) were cultured between passages 3 and 40 in the same way as described for the primary aortic ECs.

**TNFα stimulations**. To induce the acute pro-inflammatory response, HAECs (#1487 and #2139) were treated with 10-ng/mL recombinant human TNFα (Cell Applications, cat# RP1111-50), MAECs (#A092913T2MP and #B092913T2MP) were treated with 10-ng/mL recombinant mouse TNFα (Cell Applications, cat# RP2031-20), and BAECs (#1165 and 1190) were treated with 10-ng/mL recombinant bovine TNFα (R&D Systems, cat# 2279-BT-025) for 45 min in basal Endothelial Cell Growth Media MV2 (PromoCell) without supplements. The unstimulated control samples were treated with vehicle (i.e., an equivalent amount of water in MV2 media without supplements). To assess RELA translocation into nucleus by immunofluorescence, HAECs (#2139) were stimulated with 10-ng/mL TNF-α for 15, 30, 45 min, 1, and 3 hr. For the ChIP-seq experiments on histone modifications, the primary aortic ECs were starved (basal media without supplements) for 16 h prior to TNFα stimulation. For the experiments on TeloHAECs, 10 ng/mL of the recombinant human TNFα was used for 45 min, 3, 6, or 24 h.

**Immunofluorescence**. To visualize the nuclear translocation of activated RELA, unstimulated (0 min) and TNF-α-stimulated (15, 30, 45 min, 1, and 3 h) HAECs (#2139) were fixed in 2% paraformaldehyde (in PBS) for 10 min on cover slips. To permeabilize cell membranes, Triton-X (0.5% in PBS) was added for 5 min. The cover slips were incubated with RELA antibody (sc-372) in 5% donkey serum (1:50 dilution) followed by 1-h incubation with the donkey anti-rabbit Cy3-conjugated secondary IgG antibody (AP182C, Sigma-Aldrich) in 5% donkey serum (1:500 dilution). The cover slips were then incubated in DAPI (1 µg/mL in PBS) for 5 min and mounted on microscope slides. Imaging was performed using an epifluorescence microscope (Nikon TE2000). All procedures were performed at room temperature. The cover slips were washed in PBS (3 × 5 min) between each step.

**ChIP**. For multispecies ChIP-seq, two biological replicates (different individuals) of HAECs (#1487 and #2139), MAECs (#A092913T2MP and #B092913T2MP), and BAECs (#1165 and #1190) were crosslinked in 1% formaldehyde (FA) solution (50-mM Hepes-KOH, 100-mM NaCl, 1-mM EDTA, 0.5-mM EGTA, 1% FA) for 10 min at room temperature (Supplementary Dataset 1). For H3K4me1 and H3K27me3 ChIP-seq, HAECs from the individuals #2102 and #2366 were used as biological replicates (Supplementary Dataset 1). NF-κB binding events are challenging to capture due to hyper-dynamic interactions with chromatin[101]. To effectively capture the first wave of RELA binding events, we performed RELA ChIP-seq on disuccinimidyl glutarate (DSG)/FA dual crosslinked cells[101] >14 × 10⁶ cells were initially crosslinked in 2-mM DSG in PBS under shaking conditions for 30 min prior to FA crosslinking for 10 min at room temperature (FA + DSG) (Supplementary Dataset 1). Cells were then lysed for nuclei isolation as described in ref. [102]. Chromatin was sheared into 100–500-bp DNA fragments by sonication (Misonix Sonicator) at 27–28 W for 26 cycles (10-s ON, 1-min OFF, 28 W). We used ~1.7% of chromatin as input. ChIPs were performed at 4 °C overnight with 10 µg of antibody in a final volume of 250-µl block solution (0.5% BSA (w/v) in PBS): rabbit anti-RELA polyclonal (Santa Cruz sc-372 and Abcam ab7970), mouse anti-H3K27ac monoclonal (Millipore # 05-1334), rabbit anti-H3K4me2 polyclonal (Millipore # 07-030), mouse anti-H3K4me3 monoclonal (Millipore # 17-678), rabbit anti-H3K27me3 polyclonal (Millipore # 07-449), rabbit anti-H3K4me1 polyclonal (Abcam; ab8895), and rabbit anti-CTCF polyclonal (Millipore # 07-729). The cross-links were reversed at 65 °C for 16 h. Proteins and RNA were enzymatically digested, and DNA was purified using phenol–chloroform extraction and ethanol precipitation. The sc-372 and ab7970 epitopes were within the final 50 amino acids at the C terminus of human RELA. We used Protein BLAST (https://blast.ncbi.nlm.nih.gov) to test the conservation of the final 50 amino acids of

human RELA sequence (Q04206) with mouse (Q04207) and bovine (A1XG22) RELA sequences. The results showed 90% sequence identity (bit scores >85 and $e$ values $< 8.0 \times 10^{-30}$). For epigenomic characterization of TeloHAECs, two replicates (separately cultured cell populations) were crosslinked in 2-mM DSG followed by 1% FA (FA + DSG) (Supplementary Dataset 1), and ChIPs were performed at 4 °C overnight using antibodies against RELA (Santa Cruz Biotechnology sc-372) and H3K27ac (EMD Millipore 05-1334) as described above.

**ChIP-seq: library preparation and sequencing.** All of the ChIP DNA and 220 ng of input DNA were mixed with 3 units of T4 DNA polymerase (NEBNext® DNA Library Prep Master Mix Set for Illumina, #E6040L) to create blunt ends. To generate 5′-phosphates, 10 units of T4 PNK (NEBNext® DNA Library Prep Master Mix Set for Illumina, #E6040L) were added to the blunt ended DNA in NEBNext End Repair Reaction Buffer (NEB, #E6042A) and incubated for 30 min at room temperature. DNA was purified using DNA Clean and Concentrator (Zymogen, cat# D4014). To add dAMP to the 3′ ends, the end-repaired DNA was mixed with Klenow Fragment (3′ → 5′ exo⁻) (NEB, #E6044A) in NEBNext dA-Tailing Reaction Buffer (NEB, #E6045A) and incubated for 40 min at 37 °C. The dA-tailed DNA was then purified. Illumina sequencing adapters containing uracil hairpin loop structure and 3′ T overhangs (NEB, #E7337A) were ligated to the DNA by adding 150 units of Quick T4 DNA Ligase (NEB, #E6047A) in Quick Ligation Reaction Buffer (NEB, #E6048A) and incubation for 15 min at room temperature. Looped adapter sequences were opened by removal of uracil from hairpin structures by adding 3 units of USER enzyme (Uracil-Specific Excision Reagent) (NEB, M5505S) and incubation at 37 °C for 15 min. This made DNA accessible for PCR amplification with barcoded primers for Illumina sequencing (NEB, cat# E7335L). PCR amplifications were carried out for 16 cycles [98 °C 30 s, (98 °C 10 s, 65 °C 30 s, 72 °C 30 s) × 17 cycles, 72 °C 5 min, 4 °C hold]. The amplified and barcoded library was then selected for 200–350-bp fragments using Pippin Prep (Sage Science) and sequenced in Illumina HiSeq2500 with 100-bp single-end run to obtain ~20–25 million single-end reads per sample (Supplementary Dataset 1).

**ATAC-seq.** Approximately $50 \times 10^3$ cells were harvested from each biological replicate of unstimulated and TNF-α-stimulated (45 min, 10 ng/ml) HAECs (#1487 and #2139), MAECs (#A092913T2MP and #B092913T2MP), and BAECs (#1165 and #1190) (Supplementary Dataset 1). Cells were lysed in nonionic detergent lysis buffer (10-mM Tris-HCl, pH 7.4, 3-mM MgCl$_2$, 10-mM NaCl, 0.1% NP-40) for nuclei isolation. Nuclei were pelleted and resuspended in 50-µl transposition reaction solution of 2.5-µl transposome complex containing Tn5 transposase attached to sequencing adapters (Illumina Nextera Tagment DNA enzyme TDE1, Nextera® DNA Sample Preparation Kit cat# FC-121-1030), 25-µl Illumina Tagment DNA buffer (TD, Nextera® DNA Sample Preparation Kit cat# FC-121-1030), and 22.5 µl of nuclease free water and incubated in Thermomixer (Eppendorf, Hamburg, Germany) at 37 °C for 30 min under gentle mixing condition (300 rpm). The transposed and adapter-ligated DNA fragments were purified using the Qiagen MinElute PCR Purification Kit and PCR amplified using barcoded primers (Nextera Index Kit, FC-121-1011) for 12 cycles: [72 °C for 5 min, 98 °C for 30 s (98 °C for 10 s, 63 °C for 30 s, 72 °C for 1 min) × 12]. Agencourt AMPure XP beads (Beckman Coulter) were used for size-selection of >150-bp DNA fragments. ATAC-seq libraries were sequenced with 2 × 126-bp paired-end run in Illumina HiSeq2500 to obtain ~30 million, paired-end reads per sample (Supplementary Dataset 1).

**RNA-seq.** Total RNA was purified from two biological replicates of unstimulated and TNFα-stimulated (45 min, 10-ng/mL TNFα) HAECs (#1487 and #2139) using RNeasy Plus Mini kit (Qiagen). RNA quality was assessed with Agilent 2100 Bioanalyzer. Ribosomal RNA was depleted using RiboCop rRNA Depletion Kit (Lexogen). The ribosomal-RNA-depleted total RNA was used to prepare RNA-seq libraries using SENSE Total RNA-Seq Library Prep Kit (Lexogen). Libraries were sequenced to approximately fifty million, 100-bp paired-end reads per sample on a HiSeq2500 Rapid Run flowcell.

**RNA isolation, reverse transcription, and quantitative PCR.** RNA was isolated using Trizol Reagent (Thermo Fisher Scientific) as per manufacturer's instruction. Reverse transcription was performed with 0.5–1-µg RNA using the high-capacity cDNA reverse transcription kit (Applied Biosystems). The Roche Lightcycler 480 machine was used to perform RT-qPCR with primers designed for amplification of *CCL* gene transcripts (Supplementary Dataset 5) using LC 480 SYBR Green I Master Mix (Roche). For relative quantification, internal control primers for amplification of *TBP* and *GAPDH* were used (Supplementary Dataset 5). To measure absolute gene copy-number, qPCR products for each gene were purified and quantified. These qPCR products were used to generate a standard curve ranging from $10^3$ to $10^7$ copies. The gene copy-number in the sample of interest was then determined by fitting to the standard curve.

**CRISPR/Cas9 deletions and homologous recombination.** For the deletion experiments, gRNAs targeting the 5′ and 3′ boundaries of RELA peaks were designed using the MIT CRISPR design tool (http://crispr.mit.edu/) (Supplementary Dataset 6). To mutate the NF-κB consensus motifs, a gRNA was designed to

introduce a double-strand break for a homologous recombination of an introduced donor template (Supplementary Dataset 6). The gRNAs were purchased from Integrated DNA Technologies (IDT) as standard DNA oligomers. The homologous repair template was purchased from IDT as a gBlock Gene Fragment. The gRNA oligomers were then annealed, phosphorylated, and cloned into pSpCas9(BB)-2A-GFP (PX458) under the U6 promoter (Addgene Plasmid ID 48138). For the deletion experiments, TeloHAECs were transfected with 2.5 µg of each 5′ and 3′ gRNA-PX458 plasmids or two scramble control gRNA-PX458 plasmids using Lipofectamine 2000 (2 µl/1-µg DNA, Invitrogen) in 100-mm dishes. For homologous recombination, 2.5 µg of donor template containing three sequences of shuffled NF-κB motifs with one introducing a BstUI site was transfected with 2.5 µg of gRNA-PX458. The GFP+ cell populations, representing cells transfected with the GFP-tagged Cas9 construct, were isolated using FACS with a Becton Dickinson FACSAria sorter after 48 h (SickKids-UHN Flow Cytometry Facility). Non-transfected cells were utilized to set the sorting gates. After recovery, single cells were seeded in 96-well plates and inspected for colonies after 1–2 weeks. Upon 60% confluency, the colonies were split into replicate 96-well plates. Genomic DNA was isolated from one of the replicate plates by lysing cells in 96-well plates for 3 h at 65 °C in a lysis buffer (100-mM NaCl, 10-mM Tris-HCl pH 8, 25-mM EDTA pH 8, 0.5% SDS, 0.2 mg/mL) followed by 15-min inactivation at 95 °C. Genotyping was performed using Taq Polymerase (Invitrogen) with primers flanking the deletion sites or the motif mutation sites. Primers used for PCR reactions are listed in Supplementary Dataset 5. The PCR products were run-on gel electrophoresis, visualized using MiniBIS Pro (DNR Bio-Imaging Systems), and confirmed by Sanger sequencing (Supplementary Fig. 5d). Sanger sequencing was confirmed by aligning the human genome (hg19) using the Blat function in the UCSC Genome Browser. Given the potential of generating undesired large deletions using CRISPR/Cas9, up to three heterozygous and three homozygous clones were tested for each deletion as that would diminish the likelihood of basing our conclusions on one clone with spurious off target deletions. To genotype the NF-κB motif mutations, the PCR product was then digested with BstUI (NEB). The PCR products were run-on gel electrophoresis, visualized using MiniBIS Pro (DNR Bio-Imaging Systems), and confirmed by Sanger sequencing.

**Circularized chromosome conformation capture sequencing (4C-seq).** Primers were designed using the 4C primer designer (4Cpd) tool (https://mnlab.uchicago.edu/4Cpd/) for the enzyme combination DpnII (primary) and NlaIII (secondary). The site-specific sequences (highlighted in bold) were appended to the sequences complementary to Illumina adapters; CCL2_promoter_reading:TCCCTACACGA CGCTCTTCCGATC**AGGGCTGGTGATGATC**; CCL2_promoter_nonread: GTGACTGGAGTTCAGACGTGTGCTCTTCCGATC**AGGAATGGTGAACT TGGACAG**). $5 \times 10^3$ HAECs and TeloHAECs were fixed in 1% FA according to the standard ChIP-seq crosslinking procedure as described above. Pellets were stored at −80 °C prior to use. 4C libraries were constructed as described previously[103]. The isolated nuclei were digested with DpnII (NEB) overnight. Enzymes were heat-inactivated and the digested chromatin was ligated for 15 min at room temperature with T4 ligase (NEB). DNA was reverse-crosslinked and purified using AMPure XP beads (Agencourt) before digestion with NlaIII. Enzymes were heat-inactivated and the second ligation was performed at 16 °C overnight. DNA was purified by isopropanol precipitation at −80 °C overnight, of which 1 µg was used per primer set for PCR. The PCR products were column purified, and the barcoded sequencing adapters (NEB) were added in the second round of PCR. Final libraries were purified using AMPure XP beads and sequenced on the Illumina HiSeq2500 to a depth of 2–3 million single-end 54-bp reads.

**ChRO-Seq.** Chromatin run-on sequencing (ChRO-seq) on TeloHAECs was performed as previously described[53]. Briefly, chromatin was isolated from $5 \times 10^6$ cells in 1 mL of 1X NUN buffer (20-mM HEPES, 7.5-mM MgCl$_2$, 0.2-mM EDTA, 0.3-M NaCl, 1-M urea, 1% NP-40, 1-mM DTT, 20 units/mL SUPERase In RNase Inhibitor (Thermo Fisher Scientific, Waltham, MA, AM2694), 1X Protease Inhibitor Cocktail (Roche, 11836145001)). Samples were vortexed for 1 min, an additional 500 µL of 1x NUN buffer was added to each sample and were vortexed again for another 30 s. Samples were incubated in Thermomixer (Eppendorf, Hamburg, Germany) at 4 °C with shaking at 1500 rpm for 30 min followed by centrifugation at 12,500 × g for 30 min at 4 °C. Each sample was washed thrice with 1-mL 50-mM Tris-HCl (pH 7.5) supplemented with 40 units/mL SUPERase In RNase Inhibitor and centrifuged at 10,000 × g for 5 min at 4 °C. 100 µL of chromatin storage buffer (50-mM Tris-HCl pH 8.0, 25% glycerol, 5-mM magnesium acetate, 0.1-mM EDTA, 5-mM DTT, and 40 units/mL SUPERase In RNase Inhibitor) was added to the chromatin pellet and incubated on ice for 5 min. Samples were placed in a Bioruptor pico sonicator (Diagenode, Denville, NJ) and sonicated for ten cycles of 30-s ON and 30-s OFF. Sonication was repeated for another two times to completely solubilize the chromatin, and samples were snap frozen on liquid nitrogen and stored at −80 °C. Run-on reaction: 100 µL of solubilized chromatin was mixed with 100-µL 2X run-on reaction mix (10-mM Tris-HCl pH 8.0, 5-mM MgCl$_2$, 1-mM DTT, 300-mM KCl, 400-µM ATP, 400-µM GTP, 40-µM Biotin-11-UTP (Perkin Elmer, Waltham, MA, NEL543001EA), 40-µM Biotin-11-CTP (Perkin Elmer, Waltham, MA, NEL542001EA), 0.8 units/µL SUPERase In RNase Inhibitor, 1% sarkosyl). The run-on reaction was performed at 37 °C for 5 min and stopped by adding 500 µL Trizol LS (Thermo Fisher Scientific, 10296-010) to the reaction.

RNA samples were precipitated with ethanol with glycoblue as coprecipitant and resuspended in 20 µl of diethylpyrocarbonate treated water and heat denatured at 65 °C for 40 s, and base hydrolysis was performed with 0.2-N NaOH on ice for 8 min. Nascent RNA was pulled-down with streptavidin beads (New England Biolabs (NEB), Ipswich, MA, S1421S) as previously described[53]. RNA was extracted using Trizol (Thermo Fisher Scientific, 15596-026) and 3′ adaptor ligation was performed with T4 RNA Ligase 1 (NEB, M0204L). Second pulldown with streptavidin beads was performed followed by 5′ decapping with RNA 5′ pyrophosphohydrolase (RppH, NEB M0356S). The 5′ end of the RNA molecule was phosphorylated with T4 polynucleotide kinase (PNK, NEB M0201S), and 5′ adaptor ligation was performed with T4 RNA Ligase 1. Third streptavidin bead pulldown was performed again followed by reverse transcription using SuperScript III Reverse Transcriptase (Thermo Fisher Scientific, 18080-044). cDNA was amplified by PCR using the Q5 High-Fidelity DNA Polymerase (NEB, M0491S) to generate ChRO-seq libraries. Libraries were sequenced (3′ single end) at The Centre for Applied Genomics (Toronto, Canada) on Illumina NovaSeq 6000 (Illumina, San Diego, CA).

**ChIP-seq and ATAC-seq data: quality control, alignment, and peak calling**. The quality of raw ChIP-seq and ATAC-seq sequencing data was assessed with FastQC (v0.11.8) (http://www.bioinformatics.babraham.ac.uk/projects/fastqc/). The reads were trimmed of adapter sequences using Trimmomatic (v0.32; using recommended parameters)[104]. Burrows-Wheeler Aligner (BWA) (v0.7.8, default parameters)[105] was used to align the trimmed reads from HAECs, MAECs, and BAECs to hg19 (GRCh37), mm10 (GRCm38), and bosTau6 (University of Maryland v3.1) genome reference assemblies.

ChIP-seq peaks were called relative to the input (whole cell extract) data using MACS2 (v2.1.1)[106] with FDR cutoff $q \leq 0.01$. The broad option was used for calling histone peaks. ChIP-seq data were assessed with quality control metrics of the ENCODE consortium[107] for PCR bottleneck coefficient, normalized strand coefficient, nonredundant fraction, and relative strand coefficient (Supplementary Dataset 1). The reproducibility of RELA ChIP-seq peaks between the biological replicates was assessed using the irreproducible discovery rate statistic[108] (Supplementary Dataset 1). The reads from two biological replicates were then pooled, aligned, and used for downstream analyses. To identify RELA super-enhancers, we used the ROSE tool[109,110] with default parameters (ROSE_main v0.1 available at: younglab.wi.mit.edu/super_enhancer_code.html) on the RELA ChIP-seq datasets. For the inter-tissue and epigenetic comparative analyses, raw ChIP-seq data from HUVEC[16], LCL[14], HeLa[69], Adipocyte[20], and HAEC[15] were downloaded from GEO database under accession numbers GSE54000, GSE19486, GSE24518, GSE64233, and GSE89970, respectively, and processed as described above.

The quality metrics for the aligned ATAC-seq reads were assessed using ataqv (v1.0.0) (https://github.com/ParkerLab/ataqv) (Supplementary Dataset 1). The ATAC-seq reads that mapped to[57] the mitochondrial chromosome were removed, and peaks were called on reads pooled from two biological replicates using MACS2 broad option with FDR cutoff $q \leq 0.01$ (with the exception of BAEC TNFα sample in which case only #1190 replicate was used as the second replicate #1165 did not generate enough peaks; Supplementary Dataset 1). The raw ATAC-seq data from the 4-h TNFα-stimulated HAECs[15] were downloaded from GEO database (GSE89970) and processed as described above. To call nucleosomes, we used NucleoATAC (v0.3.4) with default parameters[55]. Processed data for HUVEC DNase-seq[56], HUVEC MNase-seq[18], Adipocyte DNase-seq[20], and raw data for T-cell p50 ChIP-seq[57] and T-cell ATAC-seq[57] were downloaded from GEO database under accession numbers GSE26328, GSE53343, GSE64233, GSE126505, and GSE118189, respectively.

**Cross-species comparative analyses**. To find conserved peaks, the orthologous sequences were retrieved from 13-way eutherian mammals EPO MSA available in the Ensembl Compara multispecies database (Ensembl 70)[111,112]. Conserved peaks were defined as the RELA or ATAC-seq peaks in a species genome that overlap RELA or ATAC-seq peaks in the orthologous sequences of the other species genomes by at least one base pair. The peaks that had no overlaps in the orthologous regions in the other species or the peaks that did not align to the genomes in the other species were defined as species-specific. Similarly, a conserved RELA SE was defined as an SE called in one species that overlaps an SE in the orthologous sequences of the other species genomes by at least one base pair.

**RNA-seq data: quality control, alignment, and differential analysis**. Raw RNA-seq reads were processed with Trimmomatic (v0.32; using recommended parameters)[104] to remove adaptor sequences and preserve high quality reads. The trimmed reads were then aligned to the human genome reference assembly (GRCh37/hg19) using STAR (v2.5.1b; default parameters)[113]. Quality control metrics for raw and aligned RNA-seq reads were analyzed using MultiQC (v1.3)[114]. Reads with low alignment quality (MAPQ value < 3) were filtered out. We used featureCounts (http://subread.sourceforge.net/, v1.5.0)[115] to count intronic and exonic reads separately for all genes based on the annotation file obtained from GENCODE version 19 (Ensembl 74) and filtered the "detectable" genes based on expression values (counts per million (CPM) > 1) and the number of samples they were observed in ≥2. The pipeline used to call the acute TNFα responsive genes was

similar to ref. [52] for intron-based analysis part, except we additionally included exonic counts. To detect the significant acute TNFα responsive genes at 45 min, we used edgeR (v3.18.1)[116] with double cutoffs (FDR < 0.1 and |log_2 fold change| > 0.6) and then combined the exon- and intron- identified genes. The rest were defined as constitutive genes if $\log_2\text{CPM}$ was >5 and variance was <100. To identify the significant TNFα responsive genes at 4-h TNFα induction of HAECs from previously published gene expression data[15], we downloaded the publicly available raw RNA-seq data from GEO database (GSE89970) and processed only the exonic reads.

**ChRO-seq data analysis**. We used proseq2.0 (https://github.com/Danko-Lab/proseq2.0) to process ChRO-seq data from two replicates of TeloHAEC cell lines. The default parameters for single-end reads were used. We counted a merged ChRO-seq signal ($n = 2$) over target peak regions using homer (v4.11)[93,117]. For Figs. 1e and 2e, we obtained distal RELA peaks using command "getDistalPeaks.pl [peak.bed] hg19>[peak_distal.bed]" and then calculated the mean signal using "annotatePeaks.pl [.bed] hg19 -size 4000 -hist 25 -d path_to_tnf_-treated_teloHAEC_chroseq -pc 3>[signal.txt]". For Fig. 5g, we used only the second command to calculate signal from all input peak regions. For the statistical test in Fig. 5g, we counted both strands over regions of test using bedtools coverage[118] from TNFα data ($n = 2$). An example code is: bedtools coverage -a <bedfile> -b "telo_tnf_merged.bam" -counts> "chroseq_counts.txt." We then tested specifically whether (1) three-species conserved RELA peaks have higher signal than top GERP-ranked RELA peaks and (2) whether top-ranked RELA peaks (ranked by ChIP-seq signal) have higher signal than top GERP-ranked RELA peaks using the Mann–Whitney $U$ test.

**Enrichment calculations**. ChIP-seq and ATAC-seq peaks were overlapped with bedtools v2.23.0[118]. To classify different modes of RELA binding, the overlapped peaks were visualized using UpSetR package (v1.4.0)[119]. The enrichments of conserved RELA and ATAC-seq peaks within different types of RELA peaks (X type) were calculated as follows: $\frac{\#\text{ of Conserved X type peaks}}{\#\text{ of Total X type peaks}} \Big/ \frac{\#\text{ of Total conserved peaks}}{\#\text{ of Total peaks}}$. For inter-tissue analyses, we calculated the fold enrichments within the tissue-shared or tissue-specific RELA peaks with the following formula: $\frac{\#\text{ of Tissue-shared(specific)X type peaks}}{\#\text{ of Total tissue-shared(specific)peaks}} \Big/ \frac{\#\text{ of Total X type peaks}}{\#\text{ of Total peaks}}$.

To calculate the expected frequencies of mode preservations of RELA peaks that are conserved between human and at least one other species we used:

$$P(A) = P(A \cap B) + P(A \cap C) + P(A \cap D)$$

$$= P(A1|B) \cdot P(B) + (A2|B) \cdot P(B) + (A3|B) \cdot P(B) + P(A2|C) \cdot P(C) + P(A3|D) \cdot P(D)$$

$$= (1)P(\text{preserved in all 3 species}|\text{conserved in all 3 species}) \times P(\text{conserved in all 3 species})$$

$$+ (2)P(\text{preserved in H\&M only}|\text{conserved in all 3 species}) \times P(\text{conserved in all 3 species})$$

$$+ (3)P(\text{preserved in H\&C only}|\text{conserved in all 3 species}) \times P(\text{conserved in all 3 species})$$

$$+ (4)P(\text{preserved in H\&M only}|\text{conserved in H\&M only}) \times P(\text{conserved in H\&M only})$$

$$+ (5)P(\text{preserve in H\&C only}|\text{conserved in H\&C only}) \times P(\text{conserved in H\&C only})$$

where;

A = mode preservation of the conserved human (H) RELA site in either mouse (M) or cow (C)

A1 = mode preservation of the conserved human (H) RELA site in both mouse (M) or cow (C)

A2 = mode preservation of the conserved human (H) RELA site in mouse (M) only

A3 = mode preservation of the conserved human (H) RELA site in cow (C) only

B = conservation of RELA binding across all three species (H & M & C)

C = conservation of RELA binding between H and M only (H & M)

D = conservation of RELA binding between H and C only (H & C)

$$(1) = \frac{\#\text{ Mode X among H\&M\&C conserved in H}}{\#\text{ H\&M\&C conserved sites in H}} \times \frac{\#\text{ Mode X among H\&M\&C conserved in M}}{\#\text{ H\&M\&C conserved sites in M}}$$
$$\times \frac{\#\text{ Mode X among H\&M\&C conserved in C}}{\#\text{ H\&M\&C conserved sites in C}} \times \frac{\#\text{ H\&M\&C conserved sites in H}}{\text{Total }\#\text{ conserved sites in H}}$$

$$(2) = \frac{\#\text{ Mode X among H\&M\&C conserved in H}}{\#\text{ H\&M\&C conserved sites in H}} \times \frac{\#\text{ Mode X among H\&M\&C conserved in M}}{\#\text{ H\&M\&C conserved sites in M}}$$
$$\times \frac{\#\text{ not Mode X among H\&M\&C conserved in C}}{\#\text{ H\&M\&C conserved sites in C}} \times \frac{\#\text{ H\&M\&C conserved sites in H}}{\text{Total }\#\text{ conserved sites in H}}$$

$$(3) = \frac{\#\text{ Mode X among H\&M\&C conserved in H}}{\#\text{ H\&M\&C conserved sites in H}} \times \frac{\#\text{ not Mode X among H\&M\&C conserved in M}}{\#\text{ H\&M\&C conserved sites in M}}$$
$$\times \frac{\#\text{ Mode X among H\&M\&C conserved in C}}{\#\text{ H\&M\&C conserved sites in C}} \times \frac{\#\text{ H\&M\&C conserved sites in H}}{\text{Total }\#\text{ conserved sites in H}}$$

$$(4) = \frac{\#\text{ Mode X among H\&M only conserved in H}}{\#\text{ H\&M only conserved sites in H}} \times \frac{\#\text{ Mode X among H\&M only conserved in M}}{\#\text{ H\&M only conserved sites in M}}$$
$$\times \frac{\#\text{ H\&M only conserved sites in H}}{\text{Total }\#\text{ conserved sites in H}}$$

$$(5) = \frac{\#\text{ Mode X among H\&C only conserved in H}}{\#\text{ H\&C only conserved sites in H}} \times \frac{\#\text{ Mode X among H\&C only conserved in C}}{\#\text{ H\&C only conserved sites in C}}$$
$$\times \frac{\#\text{ H\&C only conserved sites in H}}{\text{Total }\#\text{ conserved sites in H}}$$

To calculate the enrichments of different types of RELA peaks within the RELA SEs we used: $\frac{\text{# of X type peaks in SEs}}{\text{# of Total peaks in SEs}} / \frac{\text{# of Total X type peaks}}{\text{# of Total peaks}}$. The enrichments of SEs harboring a specific mode of RELA binding (Mode X) within the tissue-shared or tissue-specific RELA SEs we used the following formula: $\frac{\text{# of Tissue−shared(specific)X mode harbouring SEs}}{\text{# of Total Tissue − shared(specific)SEs}} / \frac{\text{# of Total X mode harbouring SEs}}{\text{# of Total SEs}}$. For the enrichments of the disease mutations, the RELA peaks were overlapped with SNPs from the publicly available version of the Human Gene Mutation Database (HGMD® Professional 2019.1[120]) and the enrichments were calculated as follows: $\frac{\text{# of X type peaks wit SNP}}{\text{# of Total peaks with SNP}} / \frac{\text{# of Total X type peaks}}{\text{# of Total peaks}}$. The $p$ values were calculated using the Chi-squared contingency table tests with Yates's correction for continuity. The Bonferroni correction was used to correct for multiple testing. For enrichments of RELA peaks near the RNA-seq identified target genes, the peaks were assigned to genes within +/−10-kb window of the annotated TSS. Enrichments of conserved and species-specific RELA peaks near target genes were calculated over peaks that are +/−10 kb of all genes to eliminate the bias near TSS: $\frac{\text{# of X type peaks +/− 10 kb of target gene TSS}}{\text{# of Total X type peaks +/− 10 kb of TSS of all genes}} / \frac{\text{# of Total peaks +/− 10 kb of target gene TSS}}{\text{# of Total peaks +/− 10 kb of TSS of all genes}}$ while the enrichments for all the other types of peaks were calculated using: $\frac{\text{# of X type peaks +/− 10 kb of target gene TSS}}{\text{# of Total X type peaks}} / \frac{\text{# of Total peaks +/− 10 kb of target gene TSS}}{\text{# of Total peaks}}$. The $p$ values were derived with the two-tailed Fisher's exact test, and the Bonferroni correction was used to correct for multiple testing.

**GO functions, gene interactions, genomic distances, DNA constraint, TF binding profiles, and TF motif analyses.** The GREAT v3.0 API was used to analyze gene associations of RELA peaks with parameters set to 5-kb upstream and 1-kb downstream from the gene TSSs to define basal gene-regulatory domain (assigned to extend up to 1 Mb in both directions to the nearest gene's basal gene-regulatory domain)[121]. Functional annotations of genes from GO Biological Process were used to get significant gene and function enrichments with GREAT (binomial FDR $q$ value is ≤0.05, fold enrichment ≥2 over the genome). Gene interactions and networks were analyzed using the GeneMANIA prediction server (v3.5.1) (http://genemania.org)[122] and plotted using Cytoscape (v3.6.1)[123]. R package ChIPseeker (v1.12.1)[124] was used to annotate the genomic features using TxDb.Hsapiens.UCSC.hg19.knownGene, TxDb.Mmusculus.UCSC.mm10.known-Gene, and UCSC bosTau6 ensGene annotation databases for human, mouse, and cow, respectively. To normalize aligned reads as reads per kilobase of transcript per million mapped reads and generate heatmaps and profiles for ChIP-seq and ATAC-seq we used deepTools2 (v3.0.0)[125] and to visualize we used the UCSC genome browser[126,127]. To generate heatmaps for RELA peak enrichments we used Heatmapper[128]. To estimate DNA constraint, we used GERP scores from the precomputed elements and base-wise RS scores available for human assembly hg19 at UCSC (GERP++ tracks data). To plot MNase-seq nucleosome profiles, we used deepTools2 bamCoverage tool with the –MNase argument, which considers only 130–200-bp fragments and avoids dinucleosomes or other artifacts. To scan RELA peaks for the RELA motifs, we used RSAT matrix-scan (Markov order: 1, weight score ≥ 1; v 1.214)[129] with the matrix profiles of RELA (MA0107.1) from the JASPAR database (http://jaspar.genereg.net). We used R package "beanplot" (v1.2)[130] to plot the number of RELA motifs per peak. De novo motif discovery was performed using MEME-ChIP (MEME Suite v5.1.0) (http://meme-suite.org/tools/meme-chip) (1-order model background, motif width: 6–20 bp, other parameters were set as default)[131].

**4C-seq data analysis.** Primers were designed using 4Cpd (http://mnlab.uchicago.edu/4Cpd/). 4C datasets were processed using the computational pipeline previously described[103]. Briefly, demultiplexed reads were obtained in fastq format and filtered for reads beginning with the 4C reading primer sequence (edit distance of 2). Filtered reads were processed using 4Cseqpipe (v0.7) to generate contact profiles for a 200-kb region (chr17: 32500000–32700000) with the parameters set to: -nearcis –read_length 75 –stat_type median –trend_resolution 2000.

**Comparison of RELA binding with GWAS results.** We examined 11 RELA filtered peak datasets of interest using the RELI algorithm[84], which gauges the significance of the overlap of each dataset with disease-associated genetic variants obtained from the GWAS catalog (v1.0.2, downloaded in Sep. 2018). Results were filtered such that only phenotypes with significant results (adjusted $p$ value < 0.01, fold change ≥ 2) in at least one RELA dataset were included in the final matrix, which was visualized as a heatmap of enrichment (fold change between observed and expected intersection counts) (Fig. 7). For identical or highly related phenotypes returned by RELI, we only show the results from the study involving the highest number of SNPs (see Supplementary Dataset 8 for full list).

**Reporting summary.** Further information on research design is available in the Nature Research Reporting Summary linked to this article.

## Data availability

ChIP-seq, ATAC-seq, RNA-seq, 4C-seq, and ChRO-seq data generated in this study have been submitted to the ArrayExpress database under accession numbers: "E-MTAB-7889," "E-MTAB-7878," "E-MTAB-7896," "E-MTAB-8272," and "E-MTAB-9425," respectively. The publicly available raw ChIP-seq data for HUVEC[16], LCL[14], HeLa[69], Adipocyte[20], and HAEC[15] were downloaded from GEO database under accession numbers: "GSE54000," "GSE19486," "GSE24518," "GSE64233," and "GSE89970." The publicly available raw ATAC-seq and RNA-seq data for the 4-h TNFα-stimulated HAECs[15] were downloaded from GEO database under accession number: "GSE89970." The publicly available processed data for HUVEC DNase-seq[56], HUVEC MNase-seq[18], Adipocyte DNase-seq[20] and raw data for T-cell p50 ChIP-seq[57] and T-cell ATAC-seq[57] were downloaded from GEO database under accession numbers: "GSE26328," "GSE53343," "GSE64233," "GSE126505," and "GSE118189," respectively. All other relevant data supporting the key findings of this study are available within the article and its Supplementary Information files or from the corresponding author upon reasonable request. A reporting summary for this Article is available as a Supplementary Information file. Source data are provided with this paper.

## Code availability

Source code for peak enrichment, differential analysis, and motif density analysis is available on Wilson lab GitHub repository (https://github.com/wilsonlabgroup/comparativeRELA) and https://doi.org/10.5281/zenodo.4281310. RELI source code is available on the Weirauch lab GitHub repository (https://github.com/WeirauchLab) and https://zenodo.org/record/4266978#.X77dphNKjUI.

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

## Acknowledgements

We would like to thank: The Donnelly Sequencing Centre and The Centre for Applied Genomics (TCAG) for assistance with next-generation sequencing; Liz Li (TCAG Statistical Analysis Facility) for statistical consultation; Bhooma Thiruvahindrapuram (TCAG Bioinformatics Facility) for HGMD overlaps; Tristin Liu and Bing Ren for training in the 4C-seq method and analysis; Charles Danko for generous advice setting up the ChRO-seq assays, and Helen Pickersgill (Life Science Editors) for editorial assistance. Funding for this project was provided by the Canadian Institutes of Health Research (CIHR) (201603PJT-364832) to M.D.W. and J.E.F and a Medicine by Design to M.D.W. and J.E.F (which receives funding from the Canada First Research Excellence Fund). J.E.F. and M.D.W. were the recipients of Early Researcher Awards from the Ontario Ministry of Research and Innovation and Tier 2 Canada Research Chairs from CIHR. N.K. received a Canada Graduate Scholarship from the Natural Sciences and Engineering Research Council of Canada (NSERC) and an Ontario Graduate Scholarship. L.A. was supported by a NSERC CGS-M. M.D.W and K.R were partially supported by a grant from the National Institutes of Health (NIH) through the grant NHGRI R01-HG010045-01. K.R. is the recipient of a post-doctoral fellowship from the Ted Rogers Centre for Heart Research. M.T.W. was partially supported by grants from the National Institutes of Health (R01 NS099068 and R01 AR073228) and the Cincinnati Children's Hospital Research Fund (CCRF Endowed Scholar, CCHMC CpG Pilot study award, and CCHMC GAP Award). M.K. received an Undergraduate Student Research Award from NSERC. A.A. received a Youth Abroad Scholarship from the Ministry of Education of Azerbaijan. AMR was supported by Programa de Apoyo a Proyectos de Investigación e Innovación Tecnológica – Universidad Nacional Autónoma de México (PAPIIT-UNAM) grant [IA201119] and a CONACYT FORDECYT-PRONACES grant [11311].

## Author contributions

A.A. designed and conducted experiments, performed computational analyses, analyzed data, and wrote the manuscript. N.K. designed and conducted experiments, analyzed data, and wrote the manuscript. L.W., L.A., M.L., A.M.-R., and M.K. conducted experiments and analyzed data. X.C. performed disease-relevant computational analyses. M.T.W. supervised disease-relevant computational analyses and provided critical feedback on the manuscript. J.E.F. and M.D.W. designed experiments, analyzed data, wrote the manuscript, acquired funding, and supervised the project. All authors edited and approved the manuscript.

## Competing interests

The authors declare no competing interests.
