## [Peer Review File · Nature Communications]

Reviewers' comments:

Reviewer #1 (Remarks to the Author):

In general I found the work in Alizada et al to be very interesting and in scope for Nature Communications and would have chosen to read the manuscript had I seen it published. In particular, I liked the cross species approach. Obviously, the intention is to use conservation to identify a likely functional core of the NF- κ B binding events and compare its characteristics relative to the non-conserved binding events through induction and with other genomically stratified binding events in the three genomes. They further follow up their observations with 4C analysis and genome editing of some key paradigm loci

This is obviously an interesting and useful set of data and analyses for the inflammation field but from my own standpoint there was also many interesting take homes for the gene regulation field. In general the data is of reasonable quality, with the cow being the weakest though probably serviceable in this respect. I also found the analysis comprehensive, to the point that it made the paper very "data dense" but I guess that is to be expected. I was interested in their stratification of the binding events into different modes, suggesting there are prebound events, events that seem to "biggy back" on existing open chromatin, more predictable open events in response to the translocation of the transcription factor and more contentiously binding to non-open chromatin areas.

While I very much enjoyed the work I have some comments that should be addressed.

1. We, and many others, have found that in genome editing we find a high proportion of larger deletions that are difficult to detect using just local PCR based approaches. This can obviously really mess up the interpretation of gene editing in and around clustered regulatory elements. It is not very clear how the authors have taken this into account, even down to number of clones generated and edited or long range PCRs. This leaves me with some doubt about the interpretation at the moment so it would be useful if they could explain in the methods why this would not be a concern.
2. I expect they are correct that the prebound regions are high affinity binding sites, but the other possibility is this is due to cross reactivity of the antibody to other related proteins. I am not an expert of these particular TFs so it would be worth explaining why this might not be the case to a more general readership, even if it was just to extend their analysis of proteins with similar amino acid sequences over the epitope region.
3. I feel the binding of the TF to non-open chromatin sites to still be highly contentious. Obviously, the ability for TFs to functionally bind outside of open chromatin would be a major problem to how we, at present, annotate the non-coding genome for a whole range of applications so I think this needs to be strongly supported. When looking at the data tracks in UCSC for the ccl locus it seemed rather sporadic as to why some sites were classed as C, O and OA when actually looking at the data. It looks a lot like evolutionary drift was acting on these regulatory elements and that weak binding was associated with weak chromatin, which would be sporadically "peak called". i.e. that on a per cell basis these regions were only occasionally bound by the transcription factor and also infrequently nucleosome depleted. Therefore it is still possible that the TF only binds in an open chromatin context, but weak binding with weak ATAC merely make it look like it is binding closed chromatin. While I am perfectly willing to be convinced if this is indeed correct, it seems to me that all of their analysis in this respect could be explained by this mechanism.

Reviewer #2 (Remarks to the Author):

In their manuscript, "Conserved regulatory logic at accessible and inaccessible chromatin defines the acute inflammatory response in mammals", Alizada et al. analyze the DNA binding response of NF- κ B and the associated epigenome status changes in TNF-treated aortic endothelial cells from three different species and compare the profiles to published data from several cell lines.

The authors find that 30% of NF- κ B binding occurs at orthologous regions across species, and that at some regions NF- κ B binds even before TNF stimulation and binding most strongly increases relative to the remainder of binding sites. The authors report that binding of p65 at 56% of these "super" sites" is preserved and that these regions overlap sequence variants associated with inflammatory and cardiovascular phenotypes. To assess function of these sites, the authors delete enhancer-like regions and knock-in NF- κ B motif mutations at the CCL2 locus, leading to 95% and 55% reductions in CCL2 expression, respectively. From this data, reanalysis of glucocorticoid receptor data in HeLa cells relative to p65 peaks and deletions at two other loci that however yielded unanticipated results by affecting more distal genes, they conclude that these "super sites" are relevant for inflammatory gene regulation in general.

The experimental parts of the study appear well-controlled and characterized, and the fact that the authors only analyze IDR-reproducible peak data from two individuals will offset any individual-specific peak differences associated with interindividual genomic differences. The analysis and integration of ChIP-seq and ATAC-seq to define modes of binding and resembles similar work done for GATA3 (PMID: 26922637), the integration of species comparisons with ChIP-seq and chromatin accessibility is novel. However, I struggle identifying the promised novel mechanistic insights, and the significance of the claims that their approach identifies "functional" NF- κ B sites. This is in part due to the fact that the authors generally do not test functionality of NF- κ B motifs themselves save for region# 6 of the CCL2 locus, and here only by simultaneously mutating three NF- κ B sites, without resolving the contribution of the surrounding sequence or the neighboring disease-associated variant. In my opinion, simply claiming that sites must be functional and important for inflammatory responses because their TF binding is conserved, without providing additional insight into the nature of what is conserved on the sequence level does not elevate this study over what has previously been published. Similarly, the claims related to NF- κ B binding to nucleosome-occupied sites, derived from ATAC-seq data that is dependent on accessibility to Tn5 transposase at the transcription factor binding locations to generate linker region-dependent internucleosomal signal for flanking nucleosomes are not further substantiated by e.g. MNase-based mapping of nucleosomes, and alternative explanations such as a lack of nucleosome remodeling due to, e.g. a lack of enhancer transcription or binding of additional transcription factors at these locations are not explored.

1. The use of the term "binding sites" to describe transcription factor-bound regions is confusing, especially when discussing sequence conservation. Transcription factor binding sites are typically defined as the DNA sequence motif where a transcription factor binds, and the conservation of that site relates to the lack of DNA differences between the sequences present in genomes from different organisms. In the current manuscript, "binding sites" contain multiple motifs, i.e. represent regions under the ChIP-seq peaks, the size of which is dependent fragment size and the number of bound motifs within the region. To avoid confusing the reader, I would strongly suggest using "regions" or "peak regions" instead of "binding sites" throughout the manuscript, unless binding of a protein to its cognate recognition sequence is meant.

2. Increased transcription factor binding, gene transcription activity and responsiveness to activation and inhibition, cell type-specificity and increased sequence conservation are general features of super-enhancers (PMID: 27965291). What remains to be shown is whether the frequency of "functional" individual sites within super-enhancers is higher than in regular enhancers, or whether the functionality of individual sites within super-enhancers depends on their being part of the enhancer ensemble that make up the super-enhancer. In line with this, the authors find that GR binding is most increased at conserved RELA-"super" sites" by analyzing published agonist-induced GR binding data in HeLa cells and conclude that these RELA sites are preferentially repressed by GR and thus their approach serves to identify "functional" NF- κ B sites. However, it is not clear whether this enhanced binding is simply a result of generally increased accessibility of the loci (which would not be novel) or whether this is specific to inflammatory and anti-inflammatory transcription factors and thus a function of these sites. In other words, how do other transcription factors behave in these regions, are they equally binding more frequently, which would suggest an overall increase accessibility of these sites, or is their binding not enriched? Another related question is how the NF- κ B-defined super-enhancers relate to H3K27ac-defined super-enhancers, what are the overlapping and unique sets of each, how well are they conserved, does NF- κ B binding to each differ, and how are associated genes affected by TNF

stimulation?

3. What is the reason for the increased binding of NF-κB to mode P sites over mode O and OA sites? For example, do the log odds scores/motif strengths of the NF-κB motifs in each class of peaks differ? Are motifs for other factors enriched, and where, relative to NF-κB motifs? To elucidate the molecular mechanisms underlying their findings related to chromatin accessibility it would be useful to map nucleosome positions by MNase-seq. This would provide answers as to how motifs are positioned relative to nucleosomes in each class and where there is an indication for nucleosome remodeling at either class of sites. The total RNA-seq data could provide additional information about the differences between modes in terms of eRNAs produced at any of these sites, and whether their production correlates with H3K27ac and/or (RNA polymerase II transcription-mediated) chromatin accessibility, which could be an alternative explanation for transposase-accessible versus inaccessible chromatin.

4. If I understand correctly, "conservation" in the manuscript relates to preservation of binding of p65 across species, or to the chromatin accessibility as measured by ATAC-seq rather than describing the sequence conservation of the underlying DNA sequence within orthologous regions. The preservation of p65 binding across species could be the result of actual sequence conservation of the entire region (especially when close to essential genes), the stochastic mutation of nucleotides not involved in transcription factor binding of either p65 or cooperating transcription factors, or selection against mutations in these binding sites due to their essential function. Since the manuscript does not distinguish between these possibilities and does not report the mutation frequencies within "conserved" regions of preserved transcription factor binding, it is difficult to gauge whether this conservation is serendipitous or due to functional sequence selection. The discussion of sequence conservation would benefit if binding events were put into the context of sequence conservation in general, i.e. what are the mutation frequencies at the different types of sites as defined by NF-κB ChIP-seq and ATAC-seq?

5. Related to the claim that NF-κB sites are functional when the region they are in is conserved, mutating the NF-κB motifs in CCL2-associated enhancer #6 only incurs ~55% reduction in CCL2 inducibility, compared to a 95% reduction when deleting the enhancer in its entirety. This suggests that sequences other than NF-κB motifs substantially contribute to TNF response of this enhancer. It would be important to know what the other functional sequence determinants of this and similar enhancers are, and to characterize the conservation of elements within in order to truly begin to unravel the conserved regulatory logic that underlies their function. To support that their approach can contribute to understanding the regulatory logic and meaning of disease-associated sequence variants, it would be best to test several sites that encompass GWAS SNPs by mutating them similar to their approach for the CCL2 locus. At a minimum, they should test the relevance of the rs1024611 alleles for CCL2 induction by TNF to make this claim.

6. What is the distribution of overlaps between NF-κB peaks in different species? How much would a more stringent definition of overlap (potentially modeled after the NF-κB motif distribution), e.g. ½ peak width instead of 1 bp overlap, affect the author's conclusions?

Miscellaneous

P2, L21-25: Sentence is an assortment of paraphrases that lack a final statement to conclude the "In response to..." introduction.

P3, L12-14: Sentence makes no sense, especially in conjunction with the previous sentence. There seems to be a part missing.

P3, L19: Reference 27 is not an example of a study that analyzed modes of NF-κB binding. A better reference would be Heinz, Romanoski et al, Nature 2013 (PMID: 24121437).

P4, L23: variations -> variants

P4, L31: transposon -> transposase

P7, L4: "nucleosome occupied" -> "nucleosome-occupied"

P7, L5: "NF-κB-binding" -> "NF-κB binding"

P7, L20: "RELA bound" -> "RELA-bound"

P10, L7: ") enriched" -> ") were enriched"

P16, L6: end of sentence is missing: "...when NF-κB."...?

P19, L2 "disease associated" -> "disease-associated"

P19, L3 "NF-κB bound" -> "NF-κB-bound"
P39, L19: Burrow-Wheeler -> Burrows-Wheeler

Reviewer #3 (Remarks to the Author):

The manuscript by Alizada seeks to identify highly conserved NF-κB/RELA binding sites across species, and to analyze the role of conservation and chromatin state in NF-κB-dependent binding and gene expression. The paper is well written and presents excellent integration of multiple genomic datasets across species and modalities. The paper introduces the idea of NF-κB Super Enhancers (SEs) based on ChIP-seq signal intensity and demonstrates that SEs are enriched for functional sites. This work is carefully performed and clearly presented, and will be of interest to researchers interested in NF-κB-dependent gene expression. However, it remains unclear how this work extends beyond the many NF-κB ChIP-seq/genomic studies that have analyzed NF-κB binding, binding strength, chromatin state, in relation to gene expression. While the Super Enhancer (SE) designation in terms of NF-κB binding appear new, it seems more a proxy for high ChIP-seq signal. In which case, the insight is that strong NF-κB binding and sequence conservation is enriched in functional sites, which has been said in different ways in many NF-κB (or other transcription factor) cistromes analysis papers.

The authors appear to focus attention on the P mode binding where NF-κB signal is identified prior to TNFα stimulation. As the authors show, the P sites are among the most highly bound (i.e., strongest ChIP signal) sites in the genome. Therefore, another interpretation of these is that the pre-TNF binding is either due to some binding while RELA is shuttling between cytosol and nucleus (as has been discussed in the literature), or that a small fraction of the cells are activated. Either way, this would lead to a signal at these clusters of high affinity sites. In which case, P sites is a proxy for high affinity sites in open chromatin regions. It is seemingly not surprising that these sites are enriched for active sites, or at least, it is not clear how this is novel beyond the findings of other labs that high-affinity NF-κB sites and sequence conserved binding enrich at functional sites.

As noted, the paper is well written, but it is not clear where this work distinguished itself from the many previous papers on this factor in particular, and in this area more generally. While the careful cross-species comparison is helpful for evolutionary analyses, and the integration with chromatin maps is also very useful, the conclusions seem consistent with many prior papers. Therefore, while it is solid work and should definitely be published, it is not clear that it rises to the level of Nature Communications.

Minor comments:

Page 8 "This observation suggests that many of the observed RELA binding to accessible chromatin (Mode O) may reflect opportunistic or even potentially antagonistic binding to established EC-related enhancers". While it is true that the enrichment is less, the overall numbers of O (~35,000) versus P (~600) suggest that the actually O-type sites might more relevant in terms of effects on gene expression, but that this category just happens to also have many negatives or non-functional elements. It would help if this overall number versus rate concept was articulated.

We thank all three reviewers for their constructive and thoughtful comments on our manuscript. We have addressed all the comments below and in doing so believe we have an even stronger manuscript. To address the reviewers' comments we added relevant original data consisting of chromatin run-on assays as well as additional analyses of published DNase-seq and MNase-seq data to further strength our novel observations that NF- κ B engages in substantial nucleosome-occluded binding (see new or modified Figure panels: Fig1e, Fig2e, Fig5g, FigS1e, FigS2a,b,c,d,e,d,g,i and m, FigS4c,d,e,f,, S5d,e). We have included the comments in full (indicated in blue italics) and a point-by-point response below each comment.

Reviewer #1 (Remarks to the Author):

R1-Q1 *“In general I found the work in Alizada et al to be very interesting and in scope for Nature Communications and would have chosen to read the manuscript had I seen it published. In particular, I liked the cross species approach. Obviously, the intention is to use conservation to identify a likely functional core of the NF- κ B binding events and compare its characteristics relative to the non-conserved binding events through induction and with other genomically stratified binding events in the three genomes. They further follow up their observations with 4C analysis and genome editing of some key paradigm loci*

This is obviously an interesting and useful set of data and analyses for the inflammation field but from my own standpoint there was also many interesting take homes for the gene regulation field. In general the data is of reasonable quality, with the cow being the weakest though probably serviceable in this respect. I also found the analysis comprehensive, to the point that it made the paper very “data dense” but I guess that is to be expected. I was interested in their stratification of the binding events into different modes, suggesting there are prebound events, events that seem to “piggy back” on existing open chromatin, more predictable open events in response to the translocation of the transcription factor and more contentiously binding to non-open chromatin areas.”

We thank the reviewer for their encouraging comments and their interest in our study. We are grateful that they appreciated the relevance and novelty of our work to the field of gene regulation as well as for the inflammation field.

“While I very much enjoyed the work I have some comments that should be addressed.

R1-Q2 *We, and many others, have found that in genome editing we find a high proportion of larger deletions that are difficult to detect using just local PCR based approaches. This can obviously really mess up the interpretation of gene editing in and around clustered regulatory elements. It is not very clear how the authors have taken this into account, even down to number of clones generated and edited or long range PCRs. This leaves me with some doubt about the interpretation at the moment so it would be useful if they could explain in the methods why this would not be a concern.”*

This is an important question and we first apologize for the lack of clarity in our description of how CRISPR/Cas9 deletion colonies were screened as this lack of detail likely contributed to this concern. We

are aware that others have reported unpredicted off-target deletions and re-arrangements when using the CRISPR/Cas9 technology (PMID:30850590, PMID:28561021, PMID:30010673). We have now included additional information and data to the Methods, main text, and most importantly additional data from overall of our heterozygous deletion mutants (which we regrettably did not provide in the original submission).

For each deletion experiment we created and tested more than 3 homozygous and 3 heterozygous colonies. The heterozygous colonies provide additional reassurance as it allowed us to observe a dose responsive trend to the effects of the deletion. Moreover, we maintain that by performing several independent deletions, all at the same locus, further support our finding. In particular, we generated a larger deletion (8.6 kb) spanning from the 5' of RELA peak #1 to the 3' of RELA peak #4. This deletion resulted in what appears to be an additive effect on *CCL2* expression that would be expected if the RELA peaks were individually deleted in the same clone.

The most striking deletion phenotype of the GWAS SNP containing, conserved Mode P region (chr19: 13,949,622-13,951,172), now has data from 4 heterozygous clones added. The heterozygous clone data fully supports the 3 homozygous clones. To further validate the RELA peak #6 deletion (which had the largest effect on gene expression), we generated mutants where the RELA motif was shuffled using a gRNA that differed from that of the prior experiment. We reason that by seeing a strong, albeit diminished reduction in *CCL2* expression post TNFA compared to the full deletion of chr19: 13,949,622-13,951,172) is consistent with importance of this region for *CCL2* expression and provides an additional 3 heterozygous and 3 homozygous clones that behave consistently before and after TNFA stimulation in terms of *CCL2* expression.

The deletion of another conserved Mode P RELA bound region in the *ZSWIM4* loci (Supplementary Figure 6) involved measuring the expression of neighboring genes within 500 kb from the deletion. We found two genes (*CC2D1A*, ~65 kb away; and *SAMD1*, ~250 kb away) that were significantly ($p < 0.05$) reduced while no significant loss of expression were detected for any of the other genes in the locus (11 genes tested in total). This second example of the functional importance of conserved Mode P regions does not appear to have occurred through large deletions in the region.

Overall, we believe that our original and focussed study of the *CCL2* locus itself (6 independently deleted regions and 1 homologous recombination experiment where independent experiments with homozygous and heterozygous clones showed results consistent with the copy number of the region) support our conclusions. We acknowledge that additional higher throughput experiments using CRISPR/Cas9 to repress, rather than delete certain regions will be valuable and we are working on such screens now. However, given COVID19 and other delays we do not have any results to share at this time. All of our clones are available, and we will freely share these with anyone wishing to study them further.

Specific changes to the manuscript to address the concern:

- We added a new supplemental figure (supplemental Fig. 5D) where we show the PCR strategy for genotyping the enhancer 6 deletion (which consisted of PCR amplification of the expected amplicon followed by Sanger sequencing confirmation), the PCR gel electrophoresis showing expected amplicon sizes, and the Sanger sequencing results are also included there.

- We improved the Fig.6bc so that its legend indicates how data from individual clones can be distinguished from repeat TNF α stimulations of the same clones.
- Importantly we now include all the data that we collected from heterozygous clones for all *CCL2* deletion experiments (Supplemental Fig. 5e).

e

- We updated supplemental table S5 so it now has all the primers used for qPCR as well as genotyping.

R1-Q3 “2. I expect they are correct that the prebound regions are high affinity binding sites, but the other possibility is this is due to cross reactivity of the antibody to other related proteins. I am not expert of these particular TFs so it would be worth explaining why this might not be the case to a more general readership, even if it was just to extend there analysis of proteins with similar amino acid sequences over the epitope region.”

We have several lines of evidence that suggest that cross reactivity to other related proteins is not responsible for the high RELA signal at prebound regions. Our study made use of multiple primary cell lines from three species (Fig. 1a, Supplementary Fig. 1b-d) and one human cell line (Supplementary Fig. 1h), two different antibodies (Supplementary Fig. 1b), as well as involved a detailed comparison of many

of the most cited NF- κ B ChIP-seq datasets. This believe this meets and exceeds the standards in our field. Nonetheless to further address this important concern we have now done an additional analysis with a very recent high-quality NF- κ B data set that profiled the NFKB1 (p50) subunit in addition to ATAC-seq (PMID: 31570894). This dataset was also helpful for addressing concerns about Mode C regions (RELA bound to inaccessible chromatin regions).

Page 7: “We could also readily identify Mode C regions using paired RELA and DNase-seq data obtained from human adipocytes ²⁰ (Supplementary Fig. 2e) as well as ChIP-seq using a different NF- κ B subunit NFKB1 (p50) and ATAC-seq data obtained from stimulated T-cells ⁵⁷ (Supplementary Fig. 2f). Together these results indicate that NF- κ B binding at relatively nucleosome-occluded regions is a robust feature of the acute inflammatory response.”

Supplemental page 8: “**f**, NF- κ B binding modes at open and closed chromatin identified using peaks called from ChIP-seq of the p50 of NF- κ B and ATAC-seq data obtained from activated CD4+ T cells sorted from two donors (Calderon et al., 2019).”

For this rebuttal we also plotted this T-cell p50 ChIP-seq data for each of our HAEC RELA bound regions (**Rev1-Fig1**). Encouragingly we recapitulate the same NF- κ B patterns using the p50 subunit data.

Rev1-Fig1: NFKB1(p50) ChIP-seq data (RPM) in CD4+ T-cells stimulated with CD3/CD28 antibodies + IL2 plotted at the summit of RELA peaks classified by their chromatin binding mode (this study). Plot shows that in T-cells, the Mode P regions we identified by anti-RELA ChIP-seq in HAECs, are also strongly bound by the NF- κ B subunit (NFKB1/p50). This argues that the strong Mode P regions are a general property of NF- κ B and not an artifact of RELA antibodies in human, mouse, cow and immortalized human HAEC cells.

Further supporting that our RELA ChIP-seq data is specific at Mode P regions:

- Motif enrichment analysis shows that RELA is the most enriched motif at prebound regions (Figure 2d). Based on de novo motif analyses using MEME-ChIP in Table S4 (sheet "P") we see three significant PWMs at Mode P regions with RELA being the top scoring motif (Supplementary Fig. 2h):
 - #1: **RELA sites (nsites= 284 E= 1.2e-390).**
 - #2 GGAA sites (nsites= 226 E= 2.5e-079) (which is an NF- κ B half site)
 - #3 JUN sites (nsites= 92 E= 4.1e-030).
- Moreover, data from two other species (mouse and cow) and TeloHAECs show enrichments for NF- κ B motifs at Mode P regions (Supplementary Table 4).
- We have also compared the binding of EC-expressed transcription factors JUN and ERG to these highly active (Mode P/ top-ranked regions) and while we see an increase in binding after TNFA treatment it is modest in comparison to what is observed for RELA (see the R2 – Q6 response).

R1-Q4 *“3. I feel the binding of the TF to non-open chromatin sites to still be highly contentious. Obviously, the ability for TFs to functionally bind outside of open chromatin would be a major problem to how we, at present, annotate the non-coding genome for a whole range of applications so I think this needs to be strongly supported. When looking at the data tracks in UCSC for the ccl locus it seemed rather sporadic as to why some sites were classed as C, O and OA when actually looking at the data. It looks a lot like evolutionary drift was acting on these regulatory elements and that weak binding was associated with weak chromatin, which would be sporadically “peak called”. i.e that on a per cell basis these regions were only occasionally bound by the transcription factor and also infrequently nucleosome depleted. Therefore it is still possible that the TF only binds in an open chromatin context, but weak binding with weak ATAC merely make it look like it is binding closed chromatin. While I am perfectly willing to be convinced if this is indeed correct, it seems to me that all of their analysis in this respect could be explained by this mechanism.”*

We thank the reviewer for their willingness to be convinced and the clear articulation of a scenario, which would occur in any ChIP-seq data set. With this clear comment as well as those from reviewer 2, we

performed additional analyses and took advantage of relevant data sets that now allow us to state that the Mode C regions can be detected in other cell types (i.e. adipocytes, T-cell), and be observed when using DNase-seq data, and Mnase-seq data. We can also support our findings using ATAC-seq and ChIP-seq data for the other common NF- κ subunit (NFKB1/p50) in stimulated T-cells. We have added additional text and supplemental figures (Supplemental Figures 2A-G) detailing our analysis. Again, we are grateful for these suggestions and believe now that we have an even stronger case for the existence of Mode C regions.

Indeed, through the course of our study we learned that NF- κ B interactions with nucleosome occluded canonical motifs had been proposed and supported with in vitro assays although this too seems to be considered somewhat contentious in the NF- κ B field as we noted in the introduction:

Page 3: “Although still considered controversial, several lines of evidence support a third mode in which NF- κ B binds to nucleosome-occupied DNA^{31, 32, 33, 34, 35}, however its importance with respect to the NF- κ B response is unclear.”

Echoing the reviewers comment about the importance of accessible chromatin for annotating the non-coding genome, a recent pre-print article using machine learning used DNA constraint, chromatin accessibility, RNA expression, and combinatorial binding to predict ChIP-seq data (aka Virtual ChIP-seq) performed well on a wide variety of TFs, the RELA ENCODE data was among the least predictable ChIP-seq datasets (<https://www.biorxiv.org/content/10.1101/168419v4>, see bottom left of Figure 1A of that preprint).

While maintain that we have made a strong and original case for these Mode C regions by relying on: the abundance of these Mode C regions across a wide range of RELA ChIP-signal strengths, the presence of multiple canonical motifs that matched or exceeded what is observed for Mode O as well as the remarkable Mode P regions; and most importantly the fact that we could observe that thousands of these Mode C regions had orthologous regions that were also Mode C regions in other mammalian species. Nonetheless this genome-wide observation warrants additional lines of evidence (**this same issue was also raised by Reviewer 2**). We added 7 new supplemental panels to Supplemental Fig 2 and updated the main text and methods accordingly:

Supplementary Figure 2. Epigenomic properties of RELA-bound regions

a, Comparisons between ATAC-seq and RELA ChIP-seq signals (HAEC TNF α + data) at Mode O and Mode C peaks. A cubic smoothing spline function `smooth.spline` was used in R to plot ATAC-seq data at each quantile of RELA ChIP-seq data ($r=0.140$ for Mode O and $r=0.097$ for Mode C, Pearson's product-moment correlation, $p < 2.2 \times 10^{-16}$). Violin plots show differences in ATAC-seq signals between Mode O and C peaks within each quantile of RELA signal data (p value: * $p < 1.0 \times 10^{-15}$, Welch Two Sample t-test and Bonferroni correction for multiple testing) **b**, MNase-seq data from HUVECs (Diermeier et al., 2014) was used to generate nucleosome occupancy profiles around RELA peak summits for each mode before and after TNF α stimulation. **c**, CTCF peaks (ENCODE HUVEC data from Pope et al., 2014) were used as a control to assess nucleosome profiles in the HUVEC MNase-seq data. **d**, Heatmaps showing how signals from HUVEC DNase-seq data (Neph et al., 2012) recapitulate HAEC ATAC-seq data at closed and open RELA binding modes. **e**, Heatmaps showing RELA binding modes identified using adipocyte

DNase-seq and adipocyte RELA data (Schmidt et al., 2015) that are common with RELA binding modes identified using HAEC ATAC-seq and HAEC RELA data. There are 11,303 Mode C regions in Adipocytes of which 1100 are also Mode C in HAECs, 54,603 Mode O regions in Adipocytes of which 10,343 are also Mode O in HAECs, and 11,884 Mode OA regions in Adipocytes of which 527 are also Mode OA in HAECs. **f**, NF- κ B binding modes at open and closed chromatin identified using peaks called from ChIP-seq of the p50 of NF- κ B and ATAC-seq data obtained from activated CD4⁺ T cells sorted from two donors (Calderon et al., 2019). **g**, HAEC ATAC-seq, RELA, and H3K27ac signals compared between ATAC-seq+ RELA- peaks, ATAC-seq+ RELA+ and ATAC-seq- RELA- peaks.

Page 7: “Further supporting our finding that NF- κ B can bind in the absence of obvious chromatin accessibility, ATAC-seq signal comparisons between Mode C regions and Mode O regions of similar RELA ChIP-seq signal revealed that Mode C regions have lower ATAC-seq signals independent of RELA binding strength (Supplementary Fig. 2a). (Supplementary Fig. 2a).

We asked whether Mode C regions could be detected using different biochemical assays for assessing chromatin accessibility (DNase-seq) and nucleosome positioning (MNase-seq) in ECs and other tissues. Indeed, using HUVEC DNase-seq data ⁵⁶, we found that Mode C regions lacked DNase-seq signal in contrast to Mode O regions (Supplementary Fig. 2d). Similarly, looking at Mode C regions using endothelial MNase-seq data ¹⁸ revealed high nucleosome occupancy at Mode C peak summits whereas Mode O peaks showed depletion of nucleosomes at peaks summits (Supplementary Fig. 2b). We could also readily identify Mode C regions using paired RELA and DNase-seq data obtained from human adipocytes ²⁰ (Supplementary Fig. 2e) as well as ChIP-seq using a different NF- κ B subunit NFKB1 (p50) and ATAC-seq data obtained from stimulated T-cells ⁵⁷ (Supplementary Fig. 2f). Together these results indicate that NF- κ B binding at relatively nucleosome-occluded regions is a robust feature of the acute inflammatory response.”

We also further supported our observation that on average Mode C regions occur in more active H3K27ac positive regions (see Supplement Fig. 2G) and the main text:

“Mode C regions were significantly enriched near target genes (~ 2 -fold, $p < 1.0 \times 10^{-15}$, ± 10 kb of TSSs; Supplementary Fig. 2k) and had higher average signal for H3K27ac than the $\sim 4 \times 10^4$ ATAC-seq peaks that lacked RELA binding (Supplementary Fig. 2g).”

We also added more to the discussion and more explicitly acknowledged the reviewers concern:

Page 20: “In contrast to studies demonstrating clear pioneering activity of specific TFs (e.g. the *de novo* formation of GATA3-bound enhancers at closed chromatin during the mesenchymal to epithelial transition ^{95,96}), these Mode C regions are an example of non-pioneering TF binding to inaccessible

chromatin. While our NF- κ B mode classifications were often preserved between species and supported by data from other groups that used distinct experimental protocols for assessing chromatin accessibility (DNase-seq, ATAC-seq and MNase-seq), it is important to recognize that all TF-chromatin interactions, occur along a continuum of accessibility states which exist at different frequencies within cell populations and between cell types⁹⁷. Thus, it is likely that some of the Mode C regions with weak RELA ChIP-seq signal occurred at regions that were infrequently nucleosome depleted. While *in vitro* evidence for NF- κ B-nucleosome interactions has been previously shown^{31, 32, 33, 34}, structural studies, such as the recent cryo-electron microscopy results showing interactions between GATA3 and the nucleosome⁹⁶, will be invaluable for understanding the precise nature of Mode C regions.”

Reviewer #2 (Remarks to the Author):

In their manuscript, “Conserved regulatory logic at accessible and inaccessible chromatin defines the acute inflammatory response in mammals”, Alizada et al. analyze the DNA binding response of NF- κ B and the associated epigenome status changes in TNF-treated aortic endothelial cells from three different species and compare the profiles to published data from several cell lines.

The authors find that 30% of NF- κ B binding occurs at orthologous regions across species, and that at some regions NF- κ B binds even before TNF stimulation and binding most strongly increases relative to the remainder of binding sites. The authors report that binding of p65 at 56% of these “super’ sites” is preserved and that these regions overlap sequence variants associated with inflammatory and cardiovascular phenotypes. To assess function of these sites, the authors delete enhancer-like regions and knock-in NF- κ B motif mutations at the CCL2 locus, leading to 95% and 55% reductions in CCL2 expression, respectively. From this data, reanalysis of glucocorticoid receptor data in HeLa cells relative to p65 peaks and deletions at two other loci that however yielded unanticipated results by affecting more distal genes, they conclude that these “super’ sites” are relevant for inflammatory gene regulation in general.

R2-Q1) *“ The experimental parts of the study appear well-controlled and characterized, and the fact that the authors only analyze IDR-reproducible peak data from two individuals will offset any individual-specific peak differences associated with interindividual genomic differences. The analysis and integration of ChIP-seq and ATAC-seq to define modes of binding and resembles similar work done for GATA3 (PMID: 26922637), the integration of species comparisons with ChIP-seq and chromatin accessibility is novel.”*

We thank the reviewer for acknowledging our data quality and novelty of the multi-species integration. Will also thank the reviewer for pointing out the GATA3 paper (which we now cite in the discussion as

well as a very recent and excellent follow-up paper from the same group in Nat. Comm) that lends credence to the binding of TFs to nucleosomes. While we are reassured that some of our key analyses are similar (i.e. we also make use of ATAC-seq to establish nucleosome positioning), we emphasize that for NF- κ B we find that the majority of RELA binding to relatively nucleosome-occluded chromatin does not result in the opening of chromatin, which is a key definition of a pioneer factor like GATA3. We have now added additional text to the discussion that hopefully clarifies this:

Page 20: “In contrast to studies demonstrating clear pioneering activity of specific TFs (e.g. the *de novo* formation of GATA3-bound enhancers at closed chromatin during the mesenchymal to epithelial transition^{95,96}), these Mode C regions are an example of non-pioneering TF binding to inaccessible chromatin. While our NF- κ B mode classifications were often preserved between species and supported by data from other groups that used distinct experimental protocols for assessing chromatin accessibility (DNase-seq, ATAC-seq and MNase-seq), it is important to recognize that all TF-chromatin interactions, occur along a continuum of accessibility states which exist at different frequencies within cell populations and between cell types⁹⁷. Thus, it is likely that some of the Mode C regions with weak RELA ChIP-seq signal occurred at regions that were infrequently nucleosome depleted. While *in vitro* evidence for NF- κ B-nucleosome interactions has been previously shown^{31,32,33,34}, structural studies, such as the recent cryo-electron microscopy results showing interactions between GATA3 and the nucleosome⁹⁶, will be invaluable for understanding the precise nature of Mode C regions.”

R2-Q2) *“However, I struggle identifying the promised novel mechanistic insights, and the significance of the claims that their approach identifies “functional” NF- κ B sites. This is in part due to the fact that the authors generally do not test functionality of NF- κ B motifs themselves save for region# 6 of the CCL2 locus, and here only by simultaneously mutating three NF- κ B sites, without resolving the contribution of the surrounding sequence or the neighboring disease-associated variant. In my opinion, simply claiming that sites must be functional and important for inflammatory responses because their TF binding is conserved, without providing additional insight into the nature of what is conserved on the sequence level does not elevate this study over what has previously been published.”*

We realize from this question and the question below (**R2-Q5**) that our use of the word “site” was ambiguous and likely generated unnecessary confusion. We did not set out to discover NF- κ B motifs but rather investigate properties of the regions where NF- κ B binds the genome using a novel cross species experimental approach which for the first time mapped and compared NF- κ B-chromatin binding in three mammalian species. We have clarified this throughout the manuscript. Before detailing the additional experiments and changes we added to address this challenging question, we would like to highlight the following points that we, to the best of our knowledge, still believe are novel aspects of our study:

- We identified a small number of highly accessible genomic regions that are bound by RELA under basal conditions and recruit the majority of factors after TNFa stimulation. Binding of

RELA to these rare regions is highly conserved across species and these regions appear to play a disproportionate role in the acute inflammatory response in the setting of multiple different inflammatory insults and across multiple tissues. We also show that drugs that inhibit the inflammatory response (e.g. BRD4 inhibitor or corticosteroids), preferentially affect these regions. Our comprehensive analysis of GWAS results also indicates that the regions we experimentally ascertained correspond to genetic variation linked to a myriad of inflammatory-related phenotypes.

- A further novelty is the finding that RELA binds to a substantial number of conserved regions that are located in inaccessible chromatin, many of which are constituents of super-enhancers. We identified ~20,000 RELA binding events within closed chromatin. Interestingly, our analyses show that Mode C regions were enriched near TNF α target genes and within super-enhancers, and are conserved, suggestive of functionality during inflammation.
- Regarding functional validation, we have comprehensively assessed the functional relevance of five RELA binding peaks in the *CCL2* locus and have deleted additional binding regions of other Mode P regions at *PLK2* and *ZSWIM4* loci. We provide additional measures of function. These provide strong support for their functionality. We are pursuing large scale mutagenesis studies using CRISPR screens to further interrogate the ‘functional’ regions that we have identified, but this is beyond the scope of the current study. Importantly, our study provides a wealth of information that can be used by us and others to further pursue how genetic variation in these regions impact inflammatory gene expression and phenotypes.
- We performed what we believe to be the first multi-species analysis of transcription factor jointly with chromatin accessibility. Doing this for NF- κ B was ideal given we could ascertain binding before and after treatment. To make this clearer in the paper we have added this text:

Page 9: “All NF- κ B modes showed evidence of DNA constraint at the RELA ChIP-seq peak summit with Mode P peaks having the highest constraint (Supplementary Fig. 2m). However, chromatin context affects NF- κ B binding²⁵, and whether the chromatin context of NF- κ B binding is also conserved is not known. We postulated that identifying conserved NF- κ B-bound regions that also preserved their mode of binding in the other species would be a novel and relevant criteria to identify functional regulatory regions.”

However, given this comment and the important question from Reviewer 2 below (**R2-Q10**) regarding how our study compares to what we would learn from using DNA constraint alone we have added new experimental data and analysis which involved performing chromatin run on assay (ChRO-seq) before and after TNFA treatment in teloHAEC cells.

Adding ChRO-seq to Page 8 and a new main figure panel, Fig. 2e:

“To further identify distinguishing features of the RELA binding modes, we looked at epigenomic and transcriptional changes induced by TNF α treatment. Relative to all other modes, the pre-bound Mode P regions showed by far the highest normalized read counts for active H3K27ac marks (>2-fold, $p < 1.0 \times 10^{-6}$)

⁴⁴), RELA (>9-fold, $p < 2.0 \times 10^{-60}$), ATAC-seq (>2-fold, $p < 1.0 \times 10^{-48}$), and eRNA (ChRO-seq) before and after TNF α stimulation (Fig. 2e). Similar RELA and H3K27ac ChIP-seq and ATAC-seq results were observed for mouse and cow (Supplementary Fig. 2i). Mode P regions also displayed the highest enrichment near TNF α -up-regulated genes (~3-fold, $p < 1.0 \times 10^{-8}$, +/-10kb of TSSs), 40% of which (n=226) were distal intergenic (Fig. 2f, Supplementary Fig. 2j). In contrast, Mode O regions did not undergo drastic changes in H3K27ac and ATAC-seq signal post-inflammation and were enriched near expressed genes that did not change in response to TNF α stimulation (1.2-fold, $p < 5.0 \times 10^{-7}$, +/-10kb of TSSs; Fig. 2f). This observation suggests that many of the observed RELA binding events at accessible chromatin (Mode O) reflect opportunistic or even potentially antagonistic binding to established EC-related enhancers.”

Fig 2g: Profile plot showing ChRO-seq signals (mean RPM) centered on the 3-species conserved orthologous RELA peaks (n=5027), and the equivalent number of top-ranked: RELA peaks, the peaks with highest DNA constraint (average GERP score), and CTCF peaks. Average ChRO-seq signal for all RELA peaks is shown as a baseline reference. (p-value: * $< 2.2 \times 10^{-16}$, Mann–Whitney *U* test with Bonferroni correction for multiple testing).

Page 14 (new manuscript section):

“Conserved and top-ranked RELA binding correspond to enhancer activity

Using ChRO-Seq as a functional readout for enhancer activity, we compared the functional relevance of our 3-species conserved RELA sites (n=5027; see also Fig. 1) to an equivalent number of top-ranked RELA peaks and peaks with highest DNA constraint (average GERP score). Comparisons to top-ranked CTCF peaks and all RELA peaks were also included as criteria that we would not expect to correspond with eRNA production measured by ChRO-seq. This analysis shows that the strongest ChRO-seq signal are found at 3-species conserved RELA bound regions (2.3-fold higher than all RELA peaks, $p < 2.2 \times 10^{-16}$), followed by top-ranked

RELA peaks (1.6-fold higher than all RELA peaks, $p < 2.2 \times 10^{-16}$) (Fig. 5g). Both of these enrichments were greater than what was found by taking the RELA peaks with the highest DNA constraint (top GERP vs all: 1.1-fold, $p = 0.305$; Fig. 5g).

When considering these results in the context of clustered NF- κ B binding found within SEs we can build upon the existing models of inflammatory gene regulation. Existing models demonstrate the importance of clusters of NF- κ B binding to accessible chromatin (with and without canonical NF- κ B motifs). We add to this model by suggesting that a minority of NF- κ B bound regions within a SE may play a disproportionate role in the activation of inflammatory gene expression (e.g. conserved Mode P regions) and that nucleosome occluded Mode C and OA regions, also play a role in recruiting NF- κ B at high concentrations of NF- κ B (i.e. immediately following inflammatory stimulus) through binding to multiple canonical NF- κ B motifs (Fig. 5h).”

- Addressing Reviewer 1 comment R1-Q4 and Reviewer 2’s additional comment regarding investigating the conservation of RELA binding withing SEs ascertained using H3K27ac (R2-Q6) should also highlight the novelty of our work.

R2-Q4) *Similarly, the claims related to NF- κ B binding to nucleosome-occupied sites, derived from ATAC-seq data that is dependent on accessibility to Tn5 transposase at the transcription factor binding locations to generate linker region-dependent internucleosomal signal for flanking nucleosomes are not further substantiated by e.g. MNase-based mapping of nucleosomes, and alternative explanations such as a lack of nucleosome remodeling due to, e.g. a lack of enhancer transcription or binding of additional transcription factors at these locations are not explored.”*

This is a very important question and we have given our response above when addressing a similar comment by Reviewer 1 (R1-Q2).

R2-Q5) *“1. The use of the term “binding sites” to describe transcription factor-bound regions is confusing, especially when discussing sequence conservation. Transcription factor binding sites are typically defined as the DNA sequence motif where a transcription factor binds, and the conservation of that site relates to the lack of DNA differences between the sequences present in genomes from different organisms. In the current manuscript, “binding sites” contain multiple motifs, i.e. represent regions under the ChIP-seq peaks, the size of which is dependent fragment size and the number of bound motifs within the region. To avoid confusing the reader, I would strongly suggest using “regions” or “peak regions” instead of “binding sites” throughout the manuscript, unless binding of a protein to its cognate recognition sequence is meant.”*

We thank the reviewer for this suggestion to make the writing clearer. As indicated in R2-Q1, the requested changes have been made throughout the manuscript.

R2-Q6) *“2. Increased transcription factor binding, gene transcription activity and responsiveness to activation and inhibition, cell type-specificity and increased sequence conservation are general features of super-enhancers (PMID: 27965291). What remains to be shown is whether the frequency of “functional” individual sites within super-enhancers is higher than in regular enhancers, or whether the functionality of individual sites within super-enhancers depends on their being part of the enhancer ensemble that make up the super-enhancer.*

We appreciate and agree that showing whether the frequency of “functional enhancers” within super enhancers was not fully shown. In our original submission we asked about the conservation of the components of RELA super enhancers and compared these to RELA binding that is found outside of super enhancers (Fig 2b) and Supplemental Fig. 4a (comparing RELA peaks within and outside SE’s near TNFA sensitive genes). To our knowledge this multispecies comparison of conserved orthologous TF binding within super enhancers has not been done before. Indeed in Fig. 5B and Fig. S4A we show (to our knowledge the first time for any transcription factor) that conserved orthologous RELA binding is higher within super enhancers. We have further addressed this question by performing new analyses by calling super enhancers using our H3K27ac data collected before and after TNFA stimulation and looking at the eRNA levels within RELA-bound regions inside and outside SEs (Supplemental Fig 4 c,d,e,f) and our response to comment R2-Q7 below.

In line with this, the authors find that GR binding is most increased at conserved RELA-“super sites” by analyzing published agonist-induced GR binding data in HeLa cells and conclude that these RELA sites are preferentially repressed by GR and thus their approach serves to identify “functional” NF-κB sites. However, it is not clear whether this enhanced binding is simply a result of generally increased accessibility of the loci (which would not be novel) or whether this is specific to inflammatory and anti-inflammatory transcription factors and thus a function of these sites. In other words, how do other transcription factors behave in these regions, are they equally binding more frequently, which would suggest an overall increase accessibility of these sites, or is their binding not enriched?

To remove unnecessary jargon from the manuscript we have relabelled ‘super sites’ with the more descriptive ‘top-ranked’ RELA bound regions pre-TNFA treating. Our reanalysis was done using published data in HeLa cells and the authors did not profile any additional transcription factors.

To address the reviewer’s important question about whether, in general the increased binding of RELA at the top-ranked RELA regions (pre-TNFA as well as post-TNFA treatment) is specific to RELA we performed an additional analysis using our original HAEC data. Reviewer Figure 2 (below) shows that top-ranked RELA bound regions (pre-TNFA) show an increase in RELA signal (relative to the bulk of RELA sites) that far exceeds (~10 fold) the increase observed from our original unpublished JUN ChIP-seq data (Reviewer 2 Fig 1A; before and 45 minutes after TNFA treatment; ERG ChIP-seq data (from the Romanoski lab pre and 4 hours post TNFA treatment PMID). Zooming into the JUN and ERG data show do indeed indicate modest increases in TF binding occur at these top-ranked sites.

Reviewer 2 - Figure 1: (A) RELA, JUN, and ERG ChIP-seq signals (RPM) plotted at the top-ranked RELA peaks (pre-TNFa-ranked, n=958). (B) Zoomed in look at JUN and ERG ChIP-seq signals (RPM) compared between the top-ranked RELA peaks (pre-TNFa-ranked) (dashed lines) and all RELA peaks (solid lines). Note the scale

R2-Q7) “Another related question is how the NF- κ B-defined super-enhancers relate to H3K27ac-defined super-enhancers, what are the overlapping and unique sets of each, how well are they conserved, does NF- κ B binding to each differ, and how are associated genes affected by TNF stimulation?”

This is another relevant question, which will be of interest as many studies use H3K27ac to characterize SE’s. We have now included this new analysis in the paper (Supplementary Fig. 4c-f):

S4 **c**

Supplementary Fig. 4c-f : **c**, H3K27ac SEs (red dots) in TNF α -stimulated (45-min) HAECs. The x-axis shows stitched RELAs enhancers ranked by reads per million (RPM) signal (y-axis) (as derived from ROSE). The cut-offs defining the RELAs SEs are indicated with vertical and horizontal lines. Venn diagram showing overlaps between H3K27ac and RELAs SEs. **d**, Bar charts showing fold enrichments of the conserved RELAs peaks within H3K27ac-specific, H3K27ac-RELA-common, and RELAs-specific SEs in TNF α -stimulated HAECs. Fold enrichments for the 3-species conserved, 2-species conserved, human-specific RELAs peaks are shown within SEs (pink) and non-SEs (grey) (p-value: $* < 1.0 \times 10^{-5}$, Chi-squared test with Yates's correction for continuity of independence, Bonferroni correction for multiple testing). **e**, Heatmap indicating fold enrichments of SEs within +/- 10 kb of the TSS of TNF α target genes from HAEC RNA-seq analysis (p value: $* < 1.0 \times 10^{-4}$ Fisher's exact test with FDR correction for multiple testing). up=up-regulated, down=down-regulated, NC=no change (i.e. constitutively expressed). **f**, Profile plots showing ChRO-seq signals (TeloHAEC) at conserved and human-specific RELAs peaks within and outside of H3K27ac-specific, H3K27ac-RELA-common, and RELAs-specific SEs.

These new results have been added to page 12:

“SEs are often ascertained using ChIP-seq data for H3K27ac⁶⁸. To put our RELAs SEs in the context of H3K27ac SEs we called H3K27ac SEs in TNF α + stimulated cells and compared

them to our RELA SEs (Supplementary Fig. 4c). We found 1351 H3K27ac SEs of which 838 were common with RELA SEs (Supplementary Fig. 4c). The H3K27ac-RELA-common SEs were 2-fold enriched for 3-species conserved RELA peaks ($p < 1.0 \times 10^{-177}$; Supplementary Fig. 4d) and 1.8-fold enriched near TNF α -up-regulated genes (± 10 kb of TSSs, $p < 1.0 \times 10^{-16}$; Supplementary Fig. 4e). In contrast, H3K27ac-specific SEs showed a 2.3-fold depletion of TNF α -up-regulated genes (± 10 kb of TSSs, $p < 2.0 \times 10^{-5}$; Supplementary Fig. 4e). To ask whether RELA-bound regions within RELA and H3K27ac ascertained SEs show different functional properties to RELA bound-regions outside of SEs, we looked at eRNA levels obtained from TeloHAEC ChRO-seq experiments 45 minutes after TNF α treatment (Supplementary Fig. 4f). Consistent with Figure 1e we see that the number of species where conserved orthologous RELA binding occurs corresponds to the level of ChRO-seq signal (Supplementary Fig. 4f). ChRO-seq signal at RELA bound regions inside the H3K27ac-RELA-common SEs was higher than what was observed for RELA bound regions outside of H3K27ac-RELA-common SEs ($p < 1.0 \times 10^{-16}$; Supplementary Fig. 4f).

Together, these results support our observation that SEs present after TNF α treatment are enriched for conserved orthologous NF- κ B binding and that NF- κ B occupancy at relatively nucleosome occluded regions is a common feature of these SEs. ”

R2-Q8. “3. What is the reason for the increased binding of NF- κ B to mode P sites over mode O and OA sites? For example, do the log odds scores/motif strengths of the NF- κ B motifs in each class of peaks differ? Are motifs for other factors enriched, and where, relative to NF- κ B motifs?”

This is a crucial question which we addressed in Figure 2D and Fig. S3D. In the first panel in Figure 2D we show that Mode P, Mode OA and Mode C regions have a greater fraction of canonical RELA motifs than the rest of the Mode O sites. In the second panel we show that Mode C, Mode OA and Mode P sites also have a greater number of canonical motifs. Our analyses also recapitulate that RELA binding to accessible chromatin can also occur through NF- κ B half-sites (GGAA) which are highly similar to ETS transcription factor motifs Supplementary Table S4 shows that the canonical RELA PWM looks the same across the modes but is differentially enriched at P, C, OA, CA and O peaks. NF- κ B motifs are enriched in P, C and OA peaks. We hope this rebuttal clarifies this point and that by addressing the confusing language regarding binding site and NF- κ B bound regions makes it more obvious.

R2-Q9. “To elucidate the molecular mechanisms underlying their findings related to chromatin accessibility it would be useful to map nucleosome positions by MNase-seq. This would provide answers as to how motifs are positioned relative to nucleosomes in each class and where there is an indication for nucleosome remodeling at either class of sites. The total RNA-seq data could provide additional

information about the differences between modes in terms of eRNAs produced at any of these sites, and whether their production correlates with H3K27ac and/or (RNA polymerase II transcription-mediated) chromatin accessibility, which could be an alternative explanation for transposase-accessible versus inaccessible chromatin.

- As mentioned above to Rev2-Q8, (Fig. 2D) we show the number and the enrichment of canonical motifs at each of the Nf-κB modes at the summit of RELA bound regions. The striking result here is that Mode C regions show strong enrichment for the canonical RELA motif at the peak summit and that the region surround Mode C peaks have multiple canonical motifs, which would presumably help recruit NF-κB.
- We have incorporated new analyses to this excellent suggestion of looking into MNase-seq data (see respond to Reviewer 1: R1-Q4).
- The suggestion about looking at eRNAs was very helpful and we have not included new experiment that look at eRNAs at peak regions (Fig 1e):

Page 5: “We found ChRO-seq signals to be the highest at the 3-species conserved RELA peaks followed by 2-species and human specific RELA peaks (Fig. 1e). Thus, from the perspective of NF-κB target genes and NF-κB bound regions, conserved orthologous RELA peaks highlight a functionally relevant set of inflammatory enhancers.”

Fig1e, Profile plot showing ChRO-seq signals at distal RELA bound regions (mean $\text{RPM} \times 10^{-2}$; $>3\text{kb}$ from the TSS). The signals are centered on RELA peak summits in HAECs, as indicated by color.

- Of the different RELA modes. We have also generated and analysed ChRO-seq to look at eRNA activities at distal peaks ($>3\text{ kb}$ from the TSS) for each Mode (Fig 2E). Indeed, Mode P regions produce the most eRNA.

Page 8: “Relative to all other modes, the pre-bound Mode P regions showed by far the highest normalized read counts for active H3K27ac marks (>2-fold, $p < 1.0 \times 10^{-48}$), RELA (>9-fold, $p < 2.0 \times 10^{-60}$), ATAC-seq (>2-fold, $p < 1.0 \times 10^{-48}$), and eRNA (ChRO-seq) before and after TNF α stimulation (Fig. 2e).”

Fig2e, Profile plots showing ChIP-seq (H3K27ac and RELA) RPM, ATAC-seq RPM, and ChRO-seq (eRNA) RPMx10⁻² signals (>3kb from the TSS as per Fig 1e).

R2-Q10 “4. “If I understand correctly, “conservation” in the manuscript relates to preservation of binding of p65 across species, or to the chromatin accessibility as measured by ATAC-seq rather than describing the sequence conservation of the underlying DNA sequence within orthologous regions.”

Yes, this is correct.

“The preservation of p65 binding across species could be the result of actual sequence conservation of the entire region (especially when close to essential genes), the stochastic mutation of nucleotides not involved in transcription factor binding of either p65 or cooperating transcription factors, or selection against mutations in these binding sites due to their essential function. Since the manuscript does not distinguish between these possibilities and does not report the mutation frequencies within “conserved” regions of preserved transcription factor binding, it is difficult to gauge whether this conservation is serendipitous or due to functional sequence selection. The discussion of sequence conservation would benefit if binding events were put into the context of sequence conservation in general, i.e. what are the mutation frequencies at the different types of sites as defined by NF-kB ChIP-seq and ATAC-seq?”

We thank the reviewer for this suggestion. In response, we assessed DNA constraints across RELA modes using GERP. Indeed, DNA constraint at regions where we found conserved orthologous RELA binding in 3 species is higher than regions where we only require conserved orthologous binding in two species. The higher DNA constraint of Mode P regions was observed and is consistent with our observation that these Mode P regions (>50% show conserved orthologous binding in another species (Supplemental Fig. 2M). We have added a supplemental Figure (Fig S2M) and commented on this in the main text:

Page 9: “All NF- κ B modes showed evidence of DNA constraint at the RELA ChIP-seq peak summit with Mode P peaks having the highest constraint (Supplementary Fig. 2m). However, chromatin context affects NF- κ B binding²⁵, and whether the chromatin context of NF- κ B binding is also conserved is not known.”

Supplemental Fig 2m legend (page 6: “**m**, DNA constraint (GERP score) of human RELA binding modes using hg19 GERP++ track data at UCSC.”)

We also see evidence of DNA constraint for all of the Modes. We were also to address this question through a related question from Reviewer 3 (see R3-Q2 and Fig 5g where we show that conserved orthologous TF binding sites show higher eRNA production than taking the equivalent number of RELA peaks ranked by their DNA constraint.

R2-Q11 “5. Related to the claim that NF- κ B sites are functional when the region they are in is conserved, mutating the NF- κ B motifs in CCL2-associated enhancer #6 only incurs ~55% reduction in CCL2 inducibility, compared to a 95% reduction when deleting the enhancer in its entirety. This suggests that sequences other than NF- κ B motifs substantially contribute to TNF response of this enhancer. It would be important to know what the other functional sequence determinants of this and similar enhancers are, and to characterize the conservation of elements within in order to truly begin to unravel the conserved regulatory logic that underlies their function.

This is an important point and indeed other groups who have studies the function of the rs1024611 SNP which is shown to interact with other transcription factors. We address this and properly acknowledge the literature:

Page 15: “Several non-conserved RELA motifs and multiple AP-1 motifs were still present, which may explain the less potent effect of mutation compared to deleting the entire region. Many other TFs such as IRF-1 and Prep1/Pbx have also been shown to bind to RELA peak #6 at -2578 A/G polymorphism site

(rs1024611) and implicated in regulation of *CCL2* expression^{80,81}. Therefore, a cumulative effect of multiple TFs at this enhancer could regulate *CCL2* expression⁷⁷.”

R2-Q12 “To support that their approach can contribute to understanding the regulatory logic and meaning of disease-associated sequence variants, it would be best to test several sites that encompass GWAS SNPs by mutating them similar to their approach for the *CCL2* locus. At a minimum, they should test the relevance of the rs1024611 alleles for *CCL2* induction by TNF to make this claim.”

This is an excellent suggestion, but we feel that this is beyond the scope of the current manuscript. Several studies have demonstrated that the rs1024611 allele is associated with altered *CCL2* expression (see R2-Q11 above), but it would indeed be useful to make this mutation in the endogenous locus. Of note, this SNP is not expected to directly impact RELA binding as it does not affect the RELA motif. We will focus on modeling select GWAS SNPs in future studies to better understand their functional impact. We would like to emphasize that our study identified a set of regions with immune-phenotype associated genetic variation within conserved Mode P, top-ranked and pan-tissue categories which are organized for others to use to prioritize future functional studies (Supplemental Table 7).

R2-Q12 “6. What is the distribution of overlaps between NF-κB peaks in different species? How much would a more stringent definition of overlap (potentially modeled after the NF-κB motif distribution), e.g. 1/2 peak width instead of 1 bp overlap, affect the author’s conclusions?”

To answer this question, we performed additional cross-species analysis of RELA in human, mouse and cow with varying cutoffs (see Supplemental Fig. 1E). These cutoffs did not have a major effect on the number of conserved orthologous RELA binding regions we detected. We have added additional text to the results:

Supplemental 1e: Using 1, 20, 50, and 100bp RELA peak overlaps in the ENSEMBL-EPO multiple sequence alignment as the criteria for conserved orthologous RELA binding in HAECs.

Corresponding manuscript text on Page 5: “The vast majority of the conserved RELA peaks remained (~98%) when we increased the minimum RELA peak overlap within the MSA from 1 bp to 20, 50, or 100bp (Supplementary Fig. 1e).”

R2-Q13 *“Miscellaneous*

P2, L21-25: Sentence is an assortment of paraphrases that lack a final statement to conclude the “In response to...” introduction.

P3, L12-14: Sentence makes no sense, especially in conjunction with the previous sentence. There seems to be a part missing.

P3, L19: Reference 27 is not an example of a study that analyzed modes of NF-κB binding. A better reference would be Heinz, Romanoski et al, Nature 2013 (PMID: 24121437).

P4, L23: variations -> variants

P4, L31: transposon -> transposase

P7, L4: “nucleosome occupied” -> “nucleosome-occupied”

P7, L5: “NF-κB-binding“ -> “NF-κB binding“

P7, L20: “RELA bound” -> “RELA-bound”

P10, L7: “) enriched” -> “) were enriched”

P16, L6: end of sentence is missing: “...when NF-κB.”... ?

P19, L2 “disease associated” -> “disease-associated”

P19, L3 “NF-κB bound” -> “NF-κB-bound”

P39, L19: Burrow-Wheeler -> Burrows-Wheeler”

We thank the reviewer for highlighting these areas of improvement. All of the above-mentioned corrections have been made in the main text.

Reviewer #3 (Remarks to the Author):

R3-Q1 *“The manuscript by Alizada seeks to identify highly conserved NF-κB/RELA binding sites across species, and to analyze the role of conservation and chromatin state in NF-κB-dependent binding and gene expression. The paper is well written and presents excellent integration of multiple genomic datasets across species and modalities. The paper introduces the idea of NF-κB Super Enhancers (SEs) based on ChIP-seq signal intensity and demonstrates that SEs are enriched for functional sites. This work is carefully performed and clearly presented, and will be of interest to researchers interested in NF-κB-dependent gene expression. However, it remains unclear how this work extends beyond the many NF-κB ChIP-seq/genomic studies that have analyzed NF-κB binding, binding strength, chromatin state, in relation to gene expression. While the Super Enhancer (SE) designation in terms of NF-κB binding appear new, it seems more a proxy for high ChIP-seq signal. In which case, the insight is that strong NF-κB binding and sequence conservation is enriched in functional sites, which has been said in different ways in many NF-κB (or other transcription factor) cistromes analysis papers.”*

We thank the reviewer for these kind and generous comments about data integration and the use of multiple genomic datasets that we generated for multiple species. We hope that from Reviewer 1’s comments and our response to Reviewer 1 and Reviewer 2 (which involved additional experiments using ChRO-seq), as well as further analyses to better define the binding of RELA to nucleosome occluded chromatin that we have addressed the concerns regarding novelty. While we indeed build upon the extensive work and concepts established by the NF-κB and comparative genomics fields, we are not

aware of any cistrome papers that have done a genome-wide comparative analysis of NF- κ B binding using experimental data in primary cells. We respectfully maintain that our study presents several novel insights that build upon the existing foundation that has previously been laid in this field.

R3-Q2 *“The authors appear to focus attention on the P mode binding where NF- κ B signal is identified prior to TNFa stimulation. As the authors show, the P sites are among the most highly bound (i.e., strongest ChIP signal) sites in the genome. Therefore, another interpretation of these is that the pre-TNF binding is either due to some binding while RELA is shuttling between cytosol and nucleus (as has been discussed in the literature), or that a small fraction of the cells are activated. Either way, this would lead to a signal at these clusters of high affinity sites. In which case, P sites is a proxy for high affinity sites in open chromatin regions. It is seemingly not surprising that these sites are enriched for active sites, or at least, it is not clear how this is novel beyond the findings of other labs that high-affinity NF- κ B sites and sequence conserved binding enrich at functional sites. As noted, the paper is well written, but it is not clear where this work distinguished itself from the many previous papers on this factor in particular, and in this area more generally. While the careful cross-species comparison is helpful for evolutionary analyses, and the integration with chromatin maps is also very useful, the conclusions seem consistent with many prior papers. Therefore, while it is solid work and should definitely be published, it is not clear that it rises to the level of Nature Communications”*

We completely agree with the Reviewers interpretation of the Mode P sites that we can detect prior to TNFA treatment. Our findings certainly support the observation that high-affinity regions are in general more highly functional and we do cite original studies that made this clear. We are not aware of any NF- κ B papers that did a similar ranked-based analysis of NF- κ B signal prior to and after stimulation and compared this response across species. We show that these sites are highly enriched for conserved orthologous RELA binding, pan-tissue usage, as well as correspond to rare disease mutations and regions linked to inflammation-relevant phenotypes. Our analysis of the preservation of modes of NF- κ B across species also provides new evidence supporting the long-debated presence of NF- κ B to nucleosome occluded DNA is a feature of the NF- κ B response that is conserved in mammals. We also show that a simple ranking, using an approach analogous to what is done to classify super enhancers, can reveal a set of NF- κ B bound regions that possess many important functional characteristics. Reassuringly these functional characteristics regarding multiple canonical NF- κ B motifs is in line with previous studies of individual model enhancer and promoters (e.g. the work on the NFKBIA promoter region done by Giorgetti et. al Mol Cell 2010). We are not claiming this SE-like analysis is particularly sophisticated, but rather this relatively simple approach is an effective way to prioritize NF- κ B regions with enhancer function. We also highlight in our discussion the contrast with our previous work that not all TFs have the property where stronger TF binding is equated to coordinated functional gene expression:

Page 21: “Given our observations regarding the conservation and functional properties of pan-cell type binding, it is reasonable to expect that our observations reflect the NF- κ B response in other cell types and in the presence of other stimuli. However, it is important to note that these results are in contrast to what has been observed for many developmental enhancers, which utilize weaker TF binding but optimal syntax to control spatial-temporal gene expression during

development¹⁰⁰. Such a phenomenon was also observed in a comparative analysis of liver master regulator TF binding (HNF4A, CEBPA, FOXA1 and ONECUT1) in the livers of five species. Unlike our observations for NF- κ B, the top-ranked liver *cis*-regulatory modules with the highest ChIP-seq signal showed only modest enrichments for liver pathways relative to what was seen for an equivalent number of conserved orthologous *cis*-regulatory modules⁴⁰.”

We are grateful for being challenged to better articulate the novelty of our study. This challenge, in combination of the comments of the other reviewers, encourage us to perform chromatin run-on assays which allowed us to have an independent measure of enhancer function (in addition to using conserved orthologous binding, pan-tissue binding, overlap with GWAS results and signal intensity and overlap with GWAS variation and rare regulatory mutations associated with human phenotypes). These ChRO-seq experiments allowed us to perform several new analyses which further underscore the importance of conserved orthologous NF- κ B binding above what we see for the equivalent number of top-Ranked RELA bound regions (see Fig 5G and R2-Q7 response where we now show high ChRO-seq signal at conserved orthologous RELA bound regions). We think the results shown in Fig.5G demonstrate why experimental profiling of NF- κ B occupancy in multiple species is important.

We are reassured that many of our findings are consistent with the extensive body of work looking at NF- κ B-DNA interactions. We have done our best to acknowledge the extensive literature in this field and by taking a novel evolutionary approach as well as using state of the art data integration we believe we have made new contributions both to the inflammation field as well as the comparative genomics field. We have done a novel and substantial functional dissection on the important chemokine *CCL2* and its super enhancer, which has been clearly linked to several important human diseases including atherosclerosis, myocardial infarction and infection susceptibility. We hope that our collective comments to all three reviewers help convince the reviewer of the novelty of our study.

“**Minor comments:**

R3-Q4 Page 8 “*This observation suggests that many of the observed RELA binding to accessible chromatin (Mode O) may reflect opportunistic or even potentially antagonistic binding to established EC-related enhancers*”. While it is true that the enrichment is less, the overall numbers of O (~35,000) versus P (~600) suggest that the actually O-type sites might more relevant in terms of effects on gene expression, but that this category just happens to also have many negatives or non-functional elements. It would help if this overall number versus rate concept was articulated.”

We agree with the reviewer that the binding to Mode O regions is numerically much more abundant and that many of these may indeed be functional. However, as a means to identify potentially functional regulatory regions, classification as Mode O is not helpful as these do not greatly enrich for function. On the other hand, Mode P (or more simply top-ranked RELA bound regions pre-TNFA treatment) is associated with function (e.g. association with disease-associated variation). Importantly, Mode P regions

are also utilized across multiple cell types to regulate the inflammatory response. In general, Mode O regions differ greatly across cell types due to the activity of cell-type specific TFs that maintain these regions in an open state, and they do not on aggregate control the inflammatory response across tissues.

We have discussed this important caveat that the Mode O category may contain a large number of non-functional elements, which may decrease enrichment scores in the discussion:

Page 19: “However, given the large number of Mode O peaks (>35,000) occurring over a quantitative continuum of binding, and at varying proximities to target genes, it is important to acknowledge that many crucial responses to TNF α will be mediated by Mode O regions that are not captured by simply ranking RELA signal under unstimulated conditions.”

REVIEWERS' COMMENTS

Reviewer #1 (Remarks to the Author):

I was supportive of the work in general on my initial review and I must complement the authors on the completeness of their responses to my queries. I must say I find the idea of TFs binding outside of open chromatin a very interesting observation and their point as to the ineffectiveness of the ML approaches to predict accurately is well taken as well as their further substantiation and I will reflect on these points further. I still feel that this is not a completely "nailed" question in the field but I feel they have done more than enough to support their conclusions in the context of the paper. I think this manuscript will have a positive impact on the field and generate much discussion. No more comments.

Reviewer #2 (Remarks to the Author):

I enjoyed reading the revised version of the manuscript. I feel that all of the points I raised in my previous review were addressed and congratulate the authors on a very thorough and thought-provoking manuscript.

Reviewer #3 (Remarks to the Author):

I am satisfied with the additions to the manuscript. All my concerns have been addressed.